# Firn changes at Colle Gnifetti revealed with a high-resolution process-based physical model approach

Enrico Mattea[1], Horst Machguth[1], Marlene Kronenberg[1], Ward van Pelt[2], Manuela Bassi[3], and Martin Hoelzle[1]

[1]Department of Geosciences, University of Fribourg, Fribourg, Switzerland
[2]Department of Earth Sciences, Uppsala University, Uppsala, Sweden
[3]Department of Forecasting Systems, Regional Agency for Environmental Protection of Piedmont, Turin, Italy

**Correspondence:** Enrico Mattea (enrico.mattea@unifr.ch)

**Abstract.** Our changing climate is expected to affect ice core records as cold firn progressively transitions to a temperate state. Thus, there is a need to improve our understanding and to further develop quantitative process modeling, to better predict cold firn evolution under a range of climate scenarios. Here we present the application of a distributed, fully coupled energy balance model, to simulate cold firn at the high-alpine glaciated saddle of Colle Gnifetti (Swiss/Italian Alps) over the period 2003–2018. We force the model with high-resolution, long-term and extensively quality-checked meteorological data measured in closest vicinity of the firn site, at the highest automatic weather station in Europe (Capanna Margherita, 4560 m a.s.l.). The model incorporates the spatial variability of snow accumulation rates, and is calibrated using several, partly unpublished high-altitude measurements from the Monte Rosa area. The simulation reveals a very good overall agreement in the comparison with a large archive of firn temperature profiles. Our results show that surface melt over the glaciated saddle is increasing by 3–4 mm w.e. yr$^{-2}$ depending on the location (29–36 % in 16 years), although with large inter-annual variability. Analysis of modeled melt indicates the frequent occurrence of small melt events (< 4 mm w.e.), which collectively represent a significant fraction of the melt totals. Modeled firn warming rates at 20 m depth are relatively uniform above 4450 m a.s.l. (0.4–0.5 °C per decade). They become highly variable at lower elevations, with a marked dependence on surface aspect and absolute values up to 2.5 times the local rate of atmospheric warming. Our distributed simulation contributes to the understanding of the thermal regime and evolution of a prominent site for alpine ice cores, and may support the planning of future core drilling efforts. Moreover, thanks to an extensive archive of measurements available for comparison, we also highlight the possibilities of model improvement most relevant to the investigation of future scenarios, such as the fixed-depth parametrized routine of deep preferential percolation.

## 1 Introduction

Cold firn and ice – defined by negative temperatures year-round – are recognized as a valuable archive of past atmospheric conditions, accessed through ice cores (e.g., Masson-Delmotte et al., 2006; Lüthi et al., 2008; Wolff et al., 2010; Wagenbach et al., 2012). In the recrystallization and recrystallization-infiltration firn facies, meltwater infiltration is respectively absent or limited to the near-surface layer of the firn (Shumskii, 1964; Hoelzle et al., 2011). This enables preservation of the original

layering of accumulated snow, which can be dated to provide an atmospheric record including greenhouse gases, aerosols,
precipitation, and isotopic temperature proxies (e.g., Preunkert et al., 2001; Barbante et al., 2004; Thevenon et al., 2009; Konrad et al., 2013; Bohleber et al., 2018).

The longest records are found in ice cores from the polar regions. Nonetheless, cold firn is also present in the Alps above 3400–4150 m a.s.l., depending on location and aspect (Suter et al., 2001). Such alpine cold firn is located close to major, historical sources of European anthropogenic emissions, thus providing a particularly valuable record of man-made changes to atmospheric composition (Jenk et al., 2006; Thevenon et al., 2009; Legrand et al., 2013). Moreover, the Alpine region features a historically high density of meteorological observations as well as other paleoclimatic records (such as tree rings and speleothems), enabling calibration and comparison of the climatic archives (Wagenbach et al., 2012, and references therein).

Besides its importance for ice core studies, cold firn also acts as a buffer against glacier mass losses caused by a warming climate. Specifically, meltwater refreezing close to the surface does not contribute to water runoff: thus an increased input in the firn surface energy balance (SEB), with enhanced meltwater production, does not directly affect mass balance (e.g., Harper et al., 2012). As a result, rising temperatures – instead of mass losses – are the main expression of 20th-century atmospheric warming in cold firn (Haeberli and Beniston, 1998; Hoelzle et al., 2011; Gilbert and Vincent, 2013; Vincent et al., 2020).

Climate change is expected to trigger a progressive transition from cold to temperate firn, naturally advancing from the lower elevations towards the higher (Lüthi and Funk, 2001; Vincent et al., 2007; Gilbert et al., 2010). Expected consequences for sites of presently cold firn are the onset of mass loss (e.g., van Pelt and Kohler, 2015) and an irremediable degradation of the climatic archive, induced by meltwater infiltration to increasing depths (Gabrielli et al., 2010; Hoelzle et al., 2011). Moreover, a change of thermal regime could affect the stability of cold-based hanging glaciers, with potentially hazardous consequences (Gilbert et al., 2015). Thus, a better understanding of this transition will become crucial to the continued viability of ice core campaigns, as well as the mitigation of glacier hazards and the prediction of future runoff regimes in high-alpine and polar catchments. Particularly valuable will be the acquisition of quantitative modeling capabilities to estimate the timing and uncertainties of firn changes, also incorporating the regularly updated climatic scenarios.

Among alpine cold firn sites, the Colle Gnifetti saddle (CG; Fig. 1) in the Monte Rosa range (4450 m a.s.l., Swiss/Italian Alps) stands out for the dense coverage of glaciological measurements, acquired continuously over almost 50 years. The very low annual accumulation rates of the area (between 0.3 and 1.2 m w.e.) enable ice core records covering the last millennium and potentially extending into the late Pleistocene (up to 19 kyr B.P.), further back than any other glaciated site in the Alps (Jenk et al., 2009; Wagenbach et al., 2012; Bohleber et al., 2018). Important climatological results from CG ice cores include a 1200-year time series of air temperature, reconstructed from mineral dust proxies (Bohleber et al., 2018), as well as a glacio-chemical record of anthropogenic alterations to atmospheric composition, starting before industrialization and extending to the recent emission reductions of some pollutant species (Schwikowski, 2004; Barbante et al., 2004; More et al., 2017; Gabrieli et al., 2011).

Parallel to ice core investigations, borehole measurements starting in 1976 (Haeberli and Funk, 1991) provide a detailed picture of the thermal conditions at the CG saddle. Firn temperatures below the depth of annual fluctuations (about 18 m) were measured in the range of -13.5 °C to -10 °C by Suter and Hoelzle (2002), with a clear dependence on aspect. Haeberli and

Funk (1991) found almost steady-state temperature conditions in an englacial profile from 1983, noting only a weak influence of meltwater refreezing. Lüthi and Funk (2001) reported temperature profiles from 1995 having striking bends at a depth of about 30 m, and interpreted them as the first published clues to a non-steady firn warming situation. Subsequently, Hoelzle et al. (2011) found evidence of accelerated englacial warming exceeding the air temperature increase, highlighting an enhanced role of meltwater percolation.

Several studies have previously applied simple firn models at CG, to reproduce englacial temperatures from idealized boundary conditions (Haeberli and Funk, 1991; Lüthi, 2000; Lüthi and Funk, 2001). Suter (2002) presented a distributed model study of the area, simulating one year of SEB at daily resolution, and formulated one-dimensional firn temperature predictions according to a simple parametric model. Buri (2013) used the coupled model GeoTOP to simulate energy balance and sub-surface temperatures at several locations in the CG area.

A major challenge for firn models at CG is the complex boundary condition of surface accumulation: the low accumulation rates are due to an extreme wind scouring of the snow surface, favored by the west-east saddle orientation. This process is countered by solar radiation through melt consolidation. Thus, snow accumulation has a strong spatial gradient according to terrain aspect, and is biased towards summer compared to climatological precipitation. A significant interannual variability is also observed, including years with no residual accumulation; wind erosion sometimes even affects the previous year's layer (Alean et al., 1983; Lüthi, 2000; Wagenbach et al., 2012). Surface accumulation directly controls the cold content and initial stratigraphy of the firn, as well as the rate of heat advection (e.g., Kuipers Munneke et al., 2014). However, the complex accumulation patterns at CG have not been addressed in past modeling studies, even within models coupling the energy balance to the sub-surface. The profile calculations by Suter et al. (2001) only considered heat transfer by conduction, hence neglecting vertical advection by surface accumulation. Buri (2013) simulated individual borehole locations in the CG area, relying for all of them on weather data (including precipitation) collected at the Corvatsch automatic weather station (AWS), almost 160 km away and over 1 km lower in altitude. More recently, Licciulli (2018) established a flow model independent of surface accumulation, and obtained distributed accumulation from the model on the assumption of steady state conditions.

Moreover, surface melt – increasingly occurring at high altitudes in a warming climate – is expected to play a central role in the transition from cold to temperate firn. Gilbert et al. (2014b) found on Mont Blanc that melt events strongly affect the overall energy balance of a cold firn pack in summer, because the emission of long-wave radiation from the surface becomes limited as surface temperatures reach 0 °C. The resulting energy excess is then released within the firn pack by meltwater refreezing at depth. To this day, little knowledge exists on the atmospheric conditions leading to melt events in cold firn areas, or the dynamics of such melt events. Suter et al. (2001) modeled CG firn temperatures with a 1 day time-step, too coarse compared to the time scales of melt and infiltration processes. Buri (2013), while using an hourly time-step, did not examine melt amounts and dynamics. Licciulli (2018), focusing on ice flow, modeled the firn at CG with a 1 year time-step, assuming no meltwater infiltration in the firn.

Physical models of high-alpine cold firn were also notably applied to the Col du Dôme site (CdD) in the Mont Blanc range (French Alps). At 4250 m a.s.l., CdD has a thermal regime similar to that of CG (Suter and Hoelzle, 2002); accumulation is generally higher and less affected by wind erosion (e.g., Preunkert et al., 2000). At CdD, Gilbert et al. (2014b) present a coupled

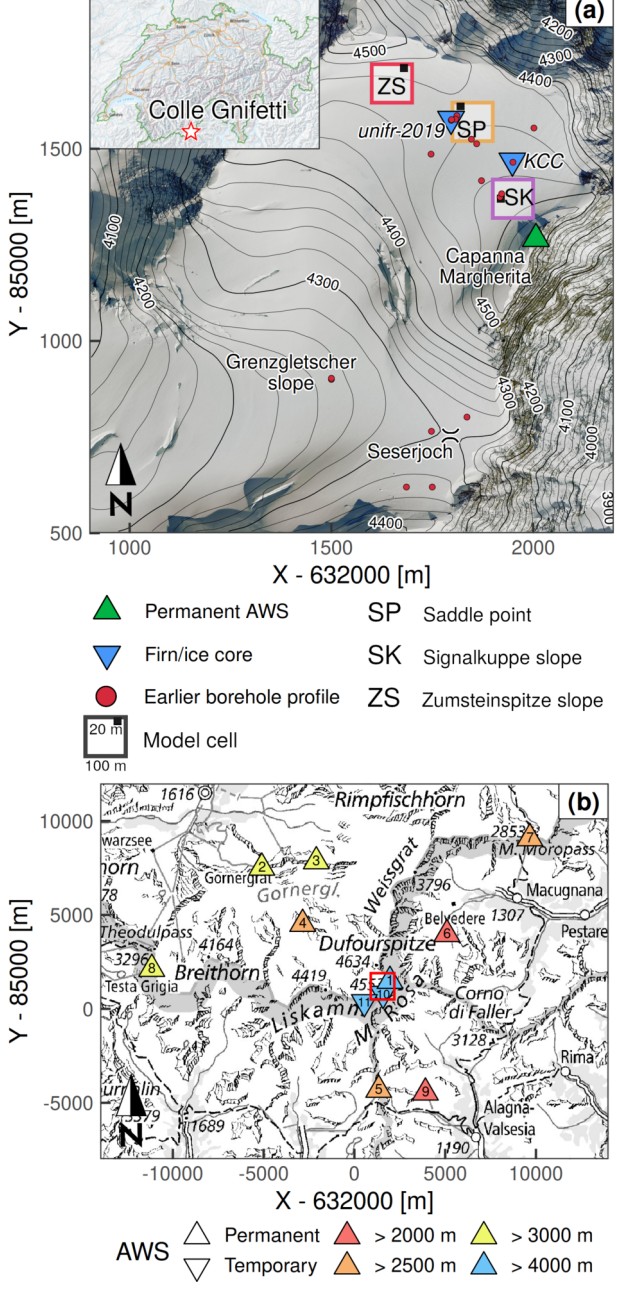

**Figure 1. (a)** Location map of the Colle Gnifetti area. Labeled model cells across the saddle correspond to the point series in Fig. 6–10. Several borehole locations were measured more than once (also see Fig. 5). **(b)** Overview map of the Monte Rosa range with AWS locations (see Table 2 for more information). The red square shows the extent of Fig. 1a. In both panels X and Y are metric CH1903/LV03 coordinates. Orthophoto and topographic map source: Federal Office of Topography swisstopo.

surface energy balance and sub-surface heat transport model, including physical representation of meltwater percolation and refreezing, but only at a single point location. Gilbert et al. (2014a) use a simplified degree-day based approach to distribute near-surface temperatures, to force a 3D glacier thermo-mechanical model.

Here, we present the application of a distributed, physically-based coupled model to simulate firn evolution at CG, over a 16 year period on a high-resolution gridded domain. After validation against a large archive of firn temperature measurements, the model is used to investigate the conditions and dynamics of high-alpine melt events. For the first time, model forcing is based on the extensively processed hourly time series of the Capanna Margherita (CM) AWS (Martorina et al., 2003), which benefits from an exceptional location at 4560 m a.s.l. and within 400 m of the CG saddle point.

The data used within the study are described in Sect. 2, the coupled model is described in Sect. 3, the results are presented in Sect. 4 and discussed in Sect. 5, while the last section provides the conclusions and some perspectives for future cold firn research.

## 2  Data

### 2.1  Meteorological time series

Meteorological forcing of the coupled model consists of an hourly time series, describing air temperature, atmospheric pressure, wind speed, relative humidity, fractional cloud cover and precipitation (Fig. 2). The series was primarily assembled using data from the CM AWS, located on the Signalkuppe summit at 4560 m a.s.l. (Fig. 1a). The station records air temperature, barometric pressure, wind speed and direction, and global radiation (Table 1). The measured parameters are available as hourly instantaneous values.

**Table 1.** Sensors installed at the CM AWS.

| Parameter | Sensor | Notes |
| --- | --- | --- |
| Air temperature | CAE TU20 thermometer | Rated accuracy 0.2 °C |
| | | Rated ambient radiation influence < 0.8 °C |
| | | Not artificially ventilated |
| Barometric pressure | CAE BA20 barometer | Rated accuracy 0.5 hPa |
| Global radiation | CAE HE20/K pyranometer | Rated daily accuracy 5 % |
| | | Wavelength band 305–2800 nm |
| Wind speed | CAE VV20 cup taco-anemometer | Rated accuracy 0.07 m s$^{-1}$ or 1 % |
| Wind direction | CAE DV20 gonio-anemometer | Rated accuracy 2.8° |

The consistent availability (> 95 %) of long-term data (since mid-2002) makes the series a valuable data-set for high-alpine research. Still, an extensive evaluation and processing of the data is essential due to the extreme measuring conditions

(Martorina et al., 2003). In particular, freezing of the anemometer – as well as snow and ice accumulation interfering with

the pyranometer – are relatively common occurrences. Therefore, we used the hourly data from eight other high-altitude AWS located in the region (Fig. 1b, id 2–9 in Table 2), to perform quality checks, to fill gaps in the CM time series, and to provide data for parameters not measured at CM. Our selection of which stations to include was determined by the availability of different parameters measured at each site, by the effort to provide an unbiased geographic coverage in all directions from the CM AWS, and by the station tendency towards concurrent failures in challenging conditions, such as winter storms.

All AWS series were pre-processed to remove clock errors, detected with cross-correlation analysis. Then each series was entirely reconstructed from the data of the best correlated others using quantile mapping (Cannon et al., 2015; Feigenwinter et al., 2018), to provide a reference for robust outlier detection. High-resolution reanalysis series of the parameters measured at CM were also collected as an additional basis for comparison (COSMO-REA2 and COSMO-REA6: Wahl et al., 2017; Bollmeyer et al., 2015; Frank et al., 2018). Due to the large volume of data involved, an automated pre-filtering routine was

implemented, based on objective criteria (absolute values, rates of change, comparison with reconstructed series and reanalysis) to mark single values as potential outliers, which were then manually checked. The CM AWS was always processed last in order to be compared to the highest-quality data. After quality check, all gaps in the hourly CM series (Table 3) were filled with the corresponding reconstructed values. The close match of the reconstructed series (Table 4) despite the large elevation differences involved (Table 2) confirms the benefit of the extensive processing described.

The CM AWS does not record series of relative humidity (RH) or cloudiness, which are required as model input. Thus RH was entirely derived from the other quality-checked time series; cloudiness was reconstructed from the incoming SW radiation at CM when available, and otherwise computed from the incoming LW radiation and observed cloud cover, respectively at Stockhorn and Plateau Rosa. Lapse rates of temperature and pressure – used to extend the AWS record to the gridded model domain – were computed from the difference between CM and the other stations. Despite the large elevation differences (Table

2), a robust computation of lapse rates was possible thanks to the availability of multiple stations within a close distance; the spread of the individual rates was usually one order of magnitude smaller than their absolute values. Moreover, the close proximity of the CM AWS to the domain minimizes the amount of extrapolation needed, as does the narrow elevation range of the domain itself. The other variables (wind speed, cloud cover and RH) were held constant over the model domain, as no measurements were available to estimate their spatial variability. Further details on the series processing can be found in

Mattea (2020); Figure 2 shows the final time series of all meteorological parameters assigned to CM and used as model input.

To determine some of the model calibration parameters (Table 5), we also re-analysed data from two additional weather and energy-balance stations (10 and 11 in Table 2), temporarily installed above 4000 m a.s.l. within the ALPCLIM project (Auer et al., 2001). The Seserjoch station operated from September 1998 to October 2000 for the detailed SEB investigations of Suter et al. (2004); the Colle del Lys station (Rossi et al., 2000a,b) was active between 1996 and 2000.

**2.2 Sub-surface data**

For model validation (Sect. 4.1) we used archived firn temperature profiles (Fig. 1a), measured between 2003 and 2018. Besides the CG saddle, we included two profiles from the Grenzgletscher slopes and four from Seserjoch. In total, we considered 25

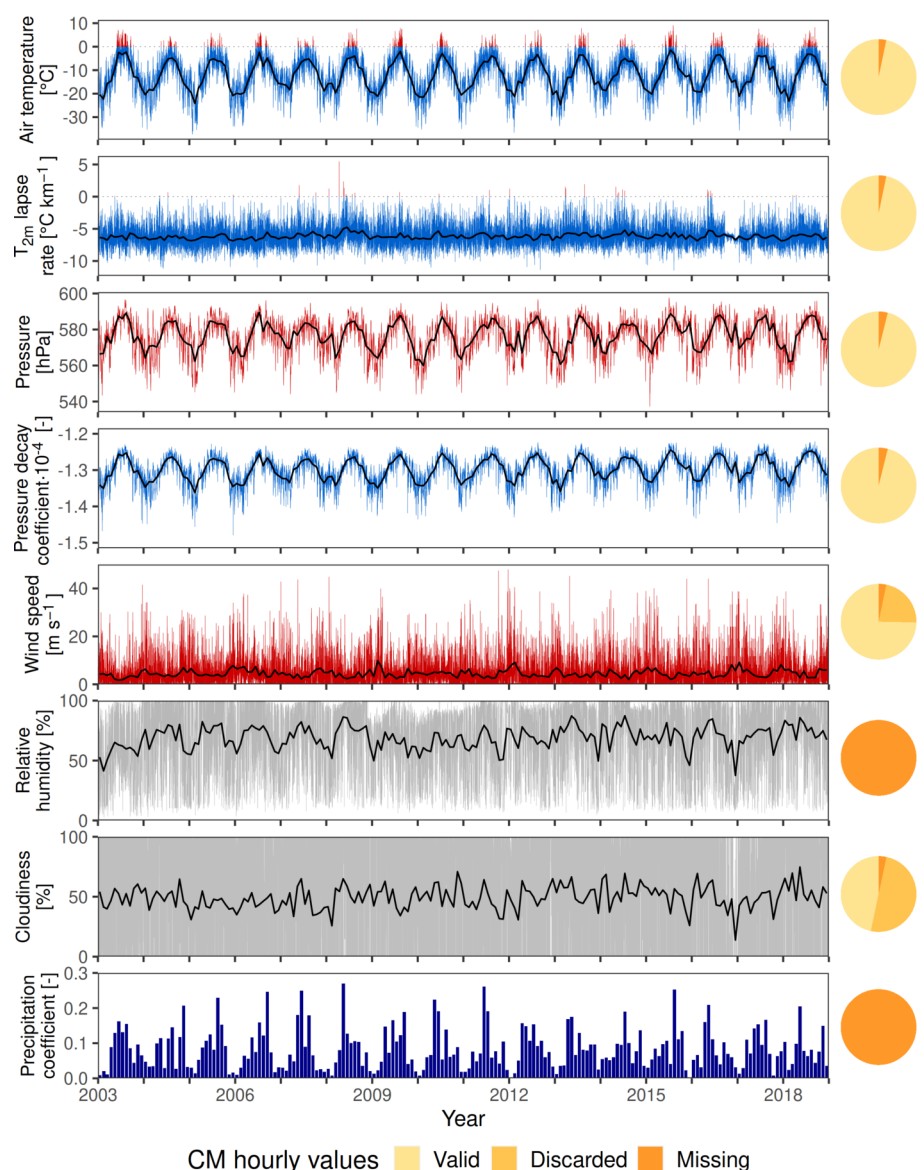

**Figure 2.** Quality-checked and gap-filled hourly weather series used as model input (reference elevation 4560 m a.s.l.). Black lines represent monthly means. For readability, only monthly means are shown for the precipitation down-scaling coefficient (Sect. 3.3). The reduced variability in temperature lapse rates in late 2016 is due to a 4 months data gap, where the series was entirely reconstructed from lower altitude stations. Pie charts show the contribution of the original CM AWS series to the model input. For cloudiness, the contribution is based on values reconstructed from SW radiation measurements. "Discarded" includes values rejected during quality check as well as night-time SW radiation measurements; "missing" includes data gaps and parameters not recorded at CM. All such values were replaced with the reconstructed series.

**Table 2.** AWS series included in this study. Numbering corresponds to Fig. 1b. X, Y and Z are metric CH1903/LV03 coordinates. $d_{SP}$ is the horizontal distance to the CG saddle point (labeled SP in Fig. 1a). Stations 10 and 11 were deployed temporarily for energy-balance studies; permanent stations 3, 4 and 6 were installed between 2006 and 2010, while the others were already operational before the CM AWS. Further AWS information is reported in Mattea (2020).

| Id | Name | X [m] | Y [m] | Z [m a.s.l.] | Variables[a] | $d_{SP}$ [m] | Reference |
|----|------|-------|-------|--------------|----------|----------|-----------|
| 1 | Capanna Margherita | 634007 | 86266 | 4560 | T, P, W, S | 380 | ARPA Piemonte (2020) |
| 2 | Gornergrat | 626900 | 92512 | 3129 | T, P, H, W, S, D | 9100 | MeteoSwiss (2020) |
| 3 | Stockhorn | 629900 | 92850 | 3415 | T, H, N, W, S, L | 7400 | Gruber et al. (2004) |
| 4 | Monte Rosa Plattje | 629149 | 89520 | 2885 | T, P, H, W, S | 5500 | MeteoSwiss (2020) |
| 5 | Passo dei Salati | 633339 | 80689 | 2970 | T, H, W | 5900 | Visit Monte Rosa (2020) |
| 6 | Macugnaga Rifugio Zamboni | 637094 | 88977 | 2075 | T, P, H, N, S, R | 4200 | ARPA Piemonte (2020) |
| 7 | Passo del Moro | 641664 | 94075 | 2820 | T, H, R, W, S | 11000 | ARPA Piemonte (2020) |
| 8 | Plateau Rosa | 620840 | 87123 | 3488 | T, P, H, W, C | 12900 | MeteoAM (2020) |
| 9 | Bocchetta delle Pisse | 635910 | 80543 | 2410 | T, N, R, W, S | 6400 | ARPA Piemonte (2020) |
| 10 | Seserjoch | 633727 | 85785 | 4292 | T, H, N, W, S, L, G, F | 820 | Suter et al. (2004) |
| 11 | Colle del Lys | 632665 | 85360 | 4236 | T, H, N, S, W | 1700 | Rossi et al. (2000a,b) |

[a] T: air temperature. P: atmospheric pressure. W: wind speed and direction. S: SW radiation. L: LW radiation. H: relative humidity. D: sunshine duration. N: snow height. C: cloud cover. R: precipitation. G: surface temperature. F: firn temperature. Not all measured variables were used in the study.

**Table 3.** Summary of the quality checks performed the CM weather series. Wind direction (see Table 1) is not included as not relevant for EBFM modeling.

| | Air temperature | Barometric pressure | Wind speed | Global radiation |
|---|---|---|---|---|
| Available values before quality check | 135689 | 134791 | 135668 | 135618 |
| Missing values before quality check | 4567 | 5465 | 4588 | 4638 |
| Available values after quality check | 135643 | 134773 | 104769 | 126970 |
| Rejection rate | 0.03 % | 0.01 % | 22.8 % | 6.4 % |
| Number of gaps after quality check | 94 | 147 | 2061 | 3286 |
| Median gap duration [h] | 9 | 2 | 4 | 1 |
| Number of gaps > 24 h | 21 | 22 | 274 | 44 |

temperature profiles from 18 boreholes, where some locations have been measured more than once. Detailed description of the profiles is reported in Hoelzle et al. (2011) and GLAMOS (2017). Earlier information on the deep temperatures (Haeberli and Funk, 1991) was also used as reference during model set-up (Sect. 3.4).

Moreover, we compiled point measurements of annual accumulation (Fig 3a), derived from layer thicknesses in GPR profiles (Konrad et al., 2013) and from the stake network of Suter and Hoelzle (2002), as well as from archived ice core measurements

**Table 4.** Summary statistics of the CM weather series processing. The mean and standard deviation (SD) are computed on the final hourly series used as model input. The root mean square error (RMSE) and mean signed deviation (BIAS) refer to the comparison between the reconstructed series and the quality-checked hourly measurements (Sect. 2.1).

| Variable | Mean | SD | RMSE | BIAS |
|---|---|---|---|---|
| Air temperature [°C] | -12.03 | 7.27 | 1.82 | 0.02 |
| Barometric pressure [hPa] | 576.70 | 9.59 | 1.26 | 0.00 |
| Wind speed [m s$^{-1}$] | 4.47 | 5.14 | 5.77 | -0.08 |
| Global radiation [W m$^{-2}$] | 194 | 290 | 114 | -2 |

(densities and annual dating) acquired at CG between 1982 and 2019. In total, 14 core profiles were available; only one (core *Zumsteinkern*, from 1991) is located on the high-accumulation, south-facing Zumsteinspitze slope (Fig. 3a). Description of the individual cores can be found in Licciulli (2018) and Lier (2018).

Finally, we hand-drilled a 5.5 m core (*unifr-2019*, Fig. 1a) near the CG saddle point on June 25, 2019, analyzing density and stratigraphy in the field. Core description is presented in Appendix A.

## 2.3 Topography

The model grid was based on a 20 m digital elevation model (DEM). Due to the Swiss-Italian border crossing the simulation domain, the DEM was produced by merging the stereo-photogrammetric SwissAlti3D dataset (acquired in 2015 with a vertical accuracy of 1–3 m) on the Swiss side, and the ICE LiDAR digital terrain model (acquired in 2011 with a vertical accuracy of 0.3–0.6 m) on the Italian side (Regione Piemonte, 2011). Elevation mismatch along the border (RMS error of 3.2 m) was corrected via smoothing, to avoid unwanted biases in slope and aspect.

## 3 The coupled energy-balance and firn model (EBFM)

The coupled model used in this work was introduced by van Pelt et al. (2012) to simulate mass balance of Nordenskiöldbreen (Svalbard). Driven by a meteorological time series, the model computes energy fluxes on the snow surface: short-wave ($SW$) and long-wave ($LW$) radiation, sensible ($Q_{SH}$) and latent ($Q_{LH}$) turbulent fluxes, heat advection from rainfall ($Q_{rain}$) and heat conduction into the snow or ice ($Q_g$). Then the SEB (Eq. 1) is solved for surface temperature and melt amounts: these, together with the lower boundary condition of geothermal heat flux $Q_{ground}$, drive the sub-surface evolution.

$$Q_{melt} = SW_{net} + LW_{net} + Q_{SH} + Q_{LH} + Q_{rain} + Q_g \tag{1}$$

Simulation of surface processes is developed along the lines of Klok and Oerlemans (2002), while the multi-layer sub-surface snow model is based on the SOMARS approach (Simulation Of glacier surface Mass balance And Related sub-surface processes, Greuell and Konzelmann, 1994). In this work, we used the model version described by van Pelt et al. (2019), with

a parametrized water percolation routine simulating preferential flow (Marchenko et al., 2017), and an updated scheme for albedo decay based on Bougamont et al. (2005). This model participated in the firn meltwater Retention Model Intercomparison Project (RetMIP) under the designation "UppsalaUniDeepPerc" (Vandecrux et al., 2020).

In the following we highlight the main EBFM routines and their respective adaptations to the CG setting. The model was originally developed over large, polythermal Arctic glaciers: set-up for a high-alpine cold firn saddle was possible thanks to a rich archive of energy-balance measurements from the Monte Rosa area.

## 3.1 Radiative fluxes

The model computes incoming SW radiation as

$$SW_{in}(x,y) = TOA_{shaded}(x,y)\ t_{rg}\ t_w\ t_a\ t_{cl} \tag{2}$$

$$t_{cl}(n) = 1 - an - bn^2. \tag{3}$$

Here, $TOA_{shaded}(x,y)$ is the unattenuated top-of-atmosphere radiation (W m$^{-2}$), corrected for topographic shading and angle of incidence on the surface. Coefficients $t_{rg}$, $t_w$, $t_a$ and $t_{cl}$ are the gaseous, water vapor, aerosol and cloud transmissivities, $n$ is the fractional cloud cover, and $a$ and $b$ are calibration parameters. Values for $a$ and $b$ (Table 5) were derived from Greuell et al. (1997) as calibrated in high-alpine terrain, unlike the EBFM defaults which were tuned from measurements in the Arctic (van Pelt et al., 2012). The parameter values of Greuell et al. (1997) increase the dependence of $t_{cl}$ on the cloud fraction, allowing a stronger decrease of the incoming SW flux under overcast conditions. Such a higher dependence was confirmed by an analysis of the distribution of the incoming radiation flux measured at the CM AWS, supporting our parameter choices (Table 5).

Reflected SW radiation is controlled by a broadband, isotropic surface albedo $\alpha_{snow}$ (Eq. 4). Albedo evolution is modeled after Oerlemans and Knap (1998) as an exponentially decaying function of time $t$ since the last significant snowfall (Eq. 5), bounded by constant values for fresh snow ($\alpha_{fresh}$) and firn ($\alpha_{firn}$). The precipitation threshold to reset albedo is 0.1 mm w.e. h$^{-1}$ (van Pelt et al., 2019). The time-scale of albedo decay is a function of snow surface temperature (Eq. 6) to account for slower metamorphism in cold conditions (Bougamont et al., 2005; van Pelt et al., 2019):

$$SW_{out} = \alpha_{snow} SW_{in} \tag{4}$$

$$\alpha_{snow}(t) = \alpha_{firn} + (\alpha_{fresh} - \alpha_{firn})e^{-\frac{t}{t^*}} \tag{5}$$

$$t^*(T_s) = \begin{cases} t^*_{wet} & T_s = 0\,^\circ C \\ t^*_{dry} + K\,|max(T_s, T_{max,t^*})| & T_s < 0\,^\circ C, \end{cases} \tag{6}$$

where $t^*$ is the computed albedo decay time-scale (d), $T_s$ snow surface temperature (K), $t^*_{wet}$ ($t^*_{dry}$) time-scale (d) for a melting (dry) surface at 0 °C, $K$ a calibration parameter (d °C$^{-1}$) and $T_{max,t^*}$ a temperature cut-off value (K) for decay slow-down (Table 5).

Incoming LW radiation is computed with the Stefan-Boltzmann law for grey-body radiation (Eq. 7); sky emissivity is modeled after Konzelmann et al. (1994) as a function of cloud cover, air temperature and humidity (Eqs. 8 and 9):

$$LW_{in} = e\sigma T^4 \tag{7}$$

$$e = e_{cs}(1 - n^2) + e_{cl}n^2 \tag{8}$$

$$e_{cs} = 0.23 + c\left(\frac{VP}{T}\right)^{0.125}, \tag{9}$$

where $\sigma$ is the Stefan-Boltzmann constant (W m$^{-2}$ K$^{-4}$), $e_{cs}$ clear-sky emissivity, $c$ a calibration parameter (K$^{0.125}$ Pa$^{-0.125}$), $VP$ vapor pressure (Pa), $T$ air temperature (K), $e$ sky emissivity and $e_{cl}$ cloud emissivity.

We calibrated several parameters of the radiation routines (Table 5) to reflect the local conditions of the high-alpine CG site. We selected the parameters to be tuned based on their relevance for our site, the availability of local measurements, and the simplicity of comparison within the model result. Within the albedo routine we optimized parameters $\alpha_{fresh}$, $t^*_{wet}$ and $K$ (Eq. 5 and 6), which refer to situations often observed at CG (respectively, a fresh snow surface, a melting surface, and a sub-freezing surface). By contrast, we kept the default values for $\alpha_{firn}$ and $t^*_{dry}$, as they correspond to conditions which are almost never encountered at our site (respectively, a bare firn surface and a non-melting surface at 0 °C). We also used the original formulations of $t_{rg}$, $t_w$ and $t_a$ (Eq. 2): this because the modeled incoming SW flux (Eqs. 2 and 3) is derived from measured radiation at CM, through the series of reconstructed cloud cover (Sect. 2.1). Thus, the effect of these transmissivity coefficients is already taken into account in the cloud cover series. Furthermore, we optimized parameters $c$ and $e_{cl}$ in the LW radiation module (Eq. 8 and 9), since locally measured values were available from the Seserjoch station.

To perform the tuning, we applied the EBFM radiation routines individually (outside the full model runs), driving them with the meteorological and energy-balance measurements of the Seserjoch and Colle del Lys stations at 10 minute resolution. Then, we adjusted the parameter values to find the best match (in terms of bias and RMSE) between the simulated and measured series of albedo and incoming LW radiation.

## 3.2 Turbulent heat fluxes

In the EBFM, turbulent heat exchange is modeled with the glacier katabatic wind parametrization of Oerlemans and Grisogono (2002). This was developed with a focus on large valley glaciers; notably, it computes heat fluxes which are independent of the ambient wind field, since wind speeds are estimated from the katabatic flow model (Oerlemans and Grisogono, 2002). This was deemed inadequate for a high-alpine, wind-exposed saddle: thus we re-implemented the EBFM computation of turbulent heat exchange, following the bulk aerodynamic equations of Essery and Etchevers (2004). These were chosen due to their operational simplicity, allowing calculation of turbulent fluxes from a single measurement level of wind speeds. The fluxes are computed as

$$Q_{SH} = \rho_a c_{p,a} C_h V_1 (T_1 - T_s) \tag{10}$$

$$Q_{LH} = \rho_a L_{s,v} C_h V_1 (q_1 - q_s), \tag{11}$$

**Table 5.** EBFM parameters considered for calibration. Additional model parameters were kept at the default value (van Pelt et al., 2012; van Pelt and Kohler, 2015; van Pelt et al., 2019) and are listed in the supplementary material.

| Parameter | Definition | Unit | Value | Source |
|---|---|---|---|---|
| $a$ | Cloud SW transmissivity coefficient | - | 0.233 | Greuell et al. (1997), supported by CM AWS data |
| $b$ | Cloud SW transmissivity coefficient | - | 0.415 | Greuell et al. (1997), supported by CM AWS data |
| $\alpha_{fresh}$ | Fresh snow albedo | - | 0.83 | Tuned from Seserjoch and Colle del Lys radiation |
| $\alpha_{firn}$ | Firn albedo | - | 0.52 | EBFM default (van Pelt and Kohler, 2015) |
| $t_{wet}^*$ | Decay time-scale (melting snow surface) | d | 10 | Tuned from Seserjoch and Colle del Lys radiation |
| $t_{dry}^*$ | Decay time-scale (dry snow surface at 0 °C) | d | 30 | EBFM default (Bougamont et al., 2005) |
| $K$ | Increase of $t_{dry}^*$ at negative temperatures | d °C$^{-1}$ | 14 | Tuned from Seserjoch and Colle del Lys radiation |
| $T_{max,t^*}$ | Snow temperature cut-off for the $t_{dry}^*$ increase | °C | -10 | EBFM default (Bougamont et al., 2005) |
| $c$ | Constant in LW emission formula | - | 0.420 | Tuned from Seserjoch LW measurements |
| $e_{cl}$ | Clouds emissivity | - | 0.960 | Tuned from Seserjoch LW measurements |
| $z_0$ | Surface roughness length | m | 0.001 | Suter et al. (2004) from Seserjoch wind profiles |
| $Q_{ground}$ | Geothermal flux | W m$^{-2}$ | 0.040 | Lüthi and Funk (2001) from CG boreholes |
| $\rho_{fresh}$ | Fresh snow density | kg m$^{-3}$ | 350 | Klok and Oerlemans (2002) |
| $z_{lim}$ | Characteristic depth of meltwater infiltration | m | 4 | Tuned to CG 20 m firn temperatures |

where $\rho_a$ is air density (kg m$^{-3}$), $c_{p,a}$ specific heat of dry air (J kg$^{-1}$ K$^{-1}$), $V$ wind speed (m s$^{-1}$), $L_{s,v}$ latent heat of sublimation or vaporization (J kg$^{-1}$, chosen depending on the modeled surface temperature), $q$ specific humidity (kg kg$^{-1}$), and the $s$ and 1 subscripts refer respectively to the snow surface and the measurement level (2 m). Exchange coefficient $C_h$ is defined as

$$C_h = C_{hn}f_h, \tag{12}$$

where $C_{hn}$ is the value under neutral conditions and $f_h$ a correction for atmospheric stability, expressed in terms of the bulk Richardson number $Ri_B$:

$$C_{hn} = k^2 \left( \log\left(\frac{z_1}{z_0}\right)\right)^{-2} \tag{13}$$

$$f_h = \begin{cases} (1+10Ri_B)^{-1} & Ri_B \geq 0 \text{ (stable)} \\ 1 - 10Ri_B \left(1 + 10C_{hn}\frac{\sqrt{-Ri_B}}{f_z}\right)^{-1} & Ri_B < 0 \text{ (unstable)} \end{cases} \tag{14}$$

$$Ri_B = \frac{gz_1}{V_1^2}\left(\frac{T_1 - T_s}{T_1} + \frac{q_1 - q_s}{q_1 + \epsilon(1-\epsilon)^{-1}}\right) \tag{15}$$

$$f_z = \frac{1}{4}\left(\frac{z_0}{z_1}\right)^{0.5}. \tag{16}$$

Here, $k$ is the von Kármán constant, $z_1$ the measurement level (m), $z_0$ the surface roughness length (m), $g$ the gravity acceleration (m s$^{-2}$), and $\epsilon$ the ratio of molecular weights for water and dry air.

At CG, surface roughness length is a poorly constrained parameter, due to frequent scouring by extreme winds which alter the snow surface. In our simulation, we used the value computed by Suter et al. (2004) from measurements of wind profiles at Seserjoch. In Appendix B we examine the sensitivity of our simulation to this parameter (Fig. B1a/b).

### 3.3 Precipitation model

We adapted the model precipitation routine to reproduce the extreme spatial gradient of snow accumulation distinctive of the site (Sect. 1). Since the EBFM does not include a blowing snow routine, the simple model of linear precipitation rates with altitude (van Pelt et al., 2019) was replaced by a gridded precipitation time series, already corrected for snow lost to wind scouring. This was computed with a three-phase anomaly method inspired from New et al. (2000), by combining a fixed climatological grid, an annual anomaly series, and a temporal down-scaling coefficient. Specifically, for grid cell $(x,y)$, at simulation time-step $t$, in year $i$, precipitation was expressed as

$$P(x,y,t) = C(x,y)A(i)D(t),\qquad(17)$$

where $C$ is the long-term annual accumulation climatology (m w.e. yr$^{-1}$), $A$ the domain-wide annual anomaly, and $D$ the down-scaling coefficient. The main assumption of the method is that spatial patterns of relative accumulation do not change over time. The climatological grid $C$ was assembled by interpolating point values of long-term net accumulation, estimated from the snow mass of dated firn cores and from the mean layer thickness in GPR profiles (Fig. 3a). To extend data coverage and reduce the occurrence of extrapolation, stake measurements from Suter and Hoelzle (2002) were also used in the western and southern domain regions. Because single-year stake measurements are in principle not representative of the long-term means, we re-scaled their values with a conversion factor. We computed this as the mean ratio between the stake and the firn core/GPR point values, taken at the locations of overlap.

For annual anomalies $A$, the multiplicative snow mass anomaly of the *KCC* deep core (Bohleber et al., 2018) was found to be moderately anti-correlated with wind speed measured at the CM AWS (Fig. 3b). Linear fit over 9 annual data points yielded the formulation

$$A = 1.46 - 0.21\overline{V}\qquad(18)$$

where $A$ is core snow mass anomaly and $\overline{V}$ the median wind speed (m s$^{-1}$) observed at CM over the corresponding year. The computed time series of annual anomaly was then applied to the whole domain. Equation 18 represents a minimal model of the inter-annual variability of wind scouring at CG, accounting for increased erosion rates at higher wind speeds.

Finally, the down-scaling coefficients $D$ were computed by normalizing to a unit sum each year of hourly precipitation, averaged over the three closest rain gauges (Passo Monte Moro, Bocchetta delle Pisse and Rifugio Zamboni: Fig. 1b, Table 2). Equation 17 produces an hourly series of gridded precipitation (already corrected for wind erosion) which was used to force the EBFM. The model uses local air temperature to compute the fraction of precipitation falling as snow: this increases linearly from 0 to 100 % within a 2 °C interval, symmetric around a threshold $T_{s/r} = 0.6$ °C.

## 3.4 Sub-surface model

The EBFM represents the sub-surface with a Lagrangian discretization: model layers move freely along the depth axis, following the addition or removal of mass at the surface. A new layer is created whenever snowfall and riming push the topmost layer thickness beyond a fixed threshold. This approach prevents numerical diffusion and at the same time accounts for heat advection towards depth (van Pelt et al., 2014).

Layer temperature evolves according to processes of heat conduction and water refreezing (van Pelt et al., 2012):

$$\rho_f c_p(T_f)\frac{\partial T_f}{\partial t} = \frac{\partial}{\partial z}\left(\kappa(\rho_f)\frac{\partial T_f}{\partial z}\right) + FL_m \tag{19}$$

where $\rho_f$ and $T_f$ are layer density (kg m$^{-3}$) and temperature (K), $c_p$ firn heat capacity (J kg$^{-1}$ K$^{-1}$), $z$ depth (m), $\kappa$ effective conductivity (W m$^{-1}$ K$^{-1}$), $F$ refreezing rate (kg m$^{-3}$ s$^{-1}$) and $L_m$ latent heat of melting (J kg$^{-1}$). Parametrizations for $c_p$ and $\kappa$ are taken respectively from Yen (1981) and Sturm et al. (1997):

$$c_p(T_f) = 152.2 + 7.122T_f \tag{20}$$

$$\kappa(\rho_f) = 0.138 - 1.01 \cdot 10^{-3}\rho_f + 3.23 \cdot 10^{-6}\rho_f^2. \tag{21}$$

Layer density is governed by gravitational settling and water refreezing (van Pelt et al., 2012):

$$\frac{\partial \rho_f}{\partial t} = K_g(\rho_f, T_f) + F \tag{22}$$

$$K_g(\rho_f, T) = b_{acc}\, g(\rho_{ice} - \rho_f)\exp\left(-\frac{E_c}{RT_f} + \frac{E_g}{RT_{avg}}\right)C_{Lig}(b_{acc}). \tag{23}$$

Here, $K_g$ is gravitational densification (kg m$^{-3}$ s$^{-1}$) as in Arthern et al. (2010), $b_{acc}$ accumulation rate (mm yr$^{-1}$), $\rho_{ice}$ ice density (kg m$^{-3}$), $R$ universal gas constant (J mol$^{-1}$ K$^{-1}$), $E_c$ (60 kJ mol$^{-1}$) and $E_g$ (42.4 kJ mol$^{-1}$) activation energies of creep by respectively lattice diffusion and grain growth, and $T_{avg}$ year-averaged firn temperature (K). $C_{Lig}$ is a correction based on the accumulation rate, accounting for different densification regimes above and below the critical density value of 550 kg m$^{-3}$ (Ligtenberg et al., 2011).

In the EBFM the only source of water at depth is infiltration from the surface (after melt, rainfall and moisture condensation), since sub-surface melting is not simulated. The model features a parametrized routine to account for preferential percolation (Marchenko et al., 2017): as long as the near-surface layers are not impermeable, liquid water is instantly routed from the surface to a prescribed sub-surface distribution, defined by its shape along the vertical axis (constant, linear or Gaussian) and maximum depth reached ($z_{lim}$). Here we selected the Gaussian profile and we tuned $z_{lim}$ to a value of 4 m, to match the 20 m firn temperature of the SP location (Fig. 1a) as measured by Haeberli and Funk (1991). The chosen reference depth allows to minimize the impact of the annual temperature cycle and of any recent temperature trends.

Water in the sub-surface can subsequently refreeze until reaching an upper bound on layer density (the value of glacier ice) and temperature (the melting point). Excess water is partly retained by capillary and adhesive forces (irreducible water content), and partly routed to deeper layers until it is depleted; if an impermeable layer is reached, the resulting slush water drains

gradually according to the linear reservoir model (van Pelt et al., 2012; Marchenko et al., 2017). The maximum irreducible water content $\theta_{mi}$ of a layer (kg kg$^{-1}$) is computed from its porosity $n_p$ following Schneider and Jansson (2004):

$$\theta_{mi} = 0.0143 \, \exp\,(3.3n_p). \tag{24}$$

Appendix B presents a sensitivity analysis of the EBFM to parameters $z_{lim}$ and $\theta_{mi}$, among others; further details on the sub-surface model can be found in van Pelt et al. (2012) and Marchenko et al. (2017).

### 3.5 Model initialization

We initialized the model grid to steady state conditions by looping eight times over the 2004–2011 weather input. The spin-up duration (64 years) enables a complete adjustment of the whole grid to the mean surface forcing (up to 20 m depth), thus 315 avoiding transitory periods at the beginning of the actual simulation. The selected sub-period excludes the extreme melt year of 2003 and the increasing temperature warming of the 2010s. Then we performed two main model runs, with 20 m / 1 h and 100 m / 3 h spatio-temporal resolution. We introduced the coarser version to decrease the large computational volume of sub-surface investigations (Fig. 6) and to examine the impact of different spatio-temporal resolutions. On the depth axis, the model grid included 250 layers up to 10 cm thick, for an effective modeling depth of about 20 m due to layer compaction.

## 4 Results

The EBFM computes and logs a wide variety of surface and sub-surface variables. In the following, we focus on firn temperatures and melt amounts as relevant descriptors of current cold firn evolution. For these variables, an extensive archive of field measurements and model estimates exists at CG, allowing validation and comparison of our results.

### 4.1 Firn temperatures

Comparison of modeled firn temperatures to measured borehole profiles (Table 6, Fig. 4 and 5) shows that most model deviations are similar in magnitude to the spatial variability of firn temperatures. Indeed, profiles CG08-1/08 and CG08-2/08 (acquired in the flat region, on the same day and within a radius of 20 m: Fig. 4e/f) report measured temperature differences in excess of 2 °C at all depths. Model residuals show a clear spatial pattern: simulated profiles tend to be too cold on shaded, north-facing slopes, and too warm in the flat or south-facing regions of CG and Seserjoch. As such, model bias is moder-330 ately correlated with mean annual accumulation and potential incoming solar radiation (correlation coefficients of respectively 0.42 and 0.64). Moreover, model residuals appear to be maintained over time for boreholes with repeated measurements (e.g., CG05-1 and CG13-1).

The time series of modeled firn temperatures (Fig. 6) show large differences over rather small distances at the CG saddle, depending on surface aspect. Still, relative annual deviations are consistent across locations. The annual cycle on average 335 reaches an amplitude of 40 °C at the snow surface and is fully damped at a depth of 20 m.

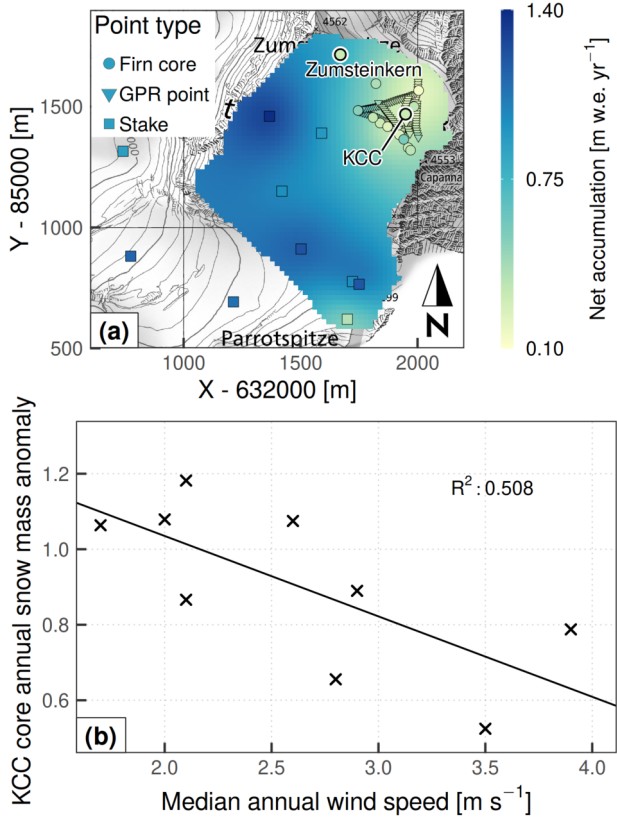

**Figure 3.** **(a)** Long-term net annual accumulation climatology at CG, serving as spatial component for the distributed accumulation model. Dense point sequences in the north-east were derived from GPR profiles. X and Y are metric CH1903/LV03 coordinates. Topographic map source: Federal Office of Topography swisstopo. **(b)** Linear fit of annual snow mass anomaly at core *KCC* versus median annual wind speed at the CM AWS.

The model simulates the frequent summer occurrence of infiltration and refreezing, reflected in sudden near-surface warming events (inset in Fig. 6). Magnitude of these events shows a high variability, with especially large heat amounts simulated in the summers of 2003, 2008, 2015 and 2017. With more meltwater refreezing, the sun-exposed ZS slope commonly shows sustained near-melting temperatures in the topmost 4 m; conversely, these conditions almost never appear at the shaded SK location. A slight positive temperature anomaly can be seen at depth after the extreme melt year of 2015, persisting over the following years despite non-record melt amounts.

The spatial distribution of 20 m firn temperatures (Fig. 7a) shows strong spatial gradients, reflecting surface elevation and aspect. Temperatures range from -17 °C at the shaded NE cliff to near-temperate conditions below 4300 m on the Grenzgletscher slope (Fig. 1a).

Modeled temperature trends are relatively uniform over the saddle area above 4400 m a.s.l., where the overall 2003-2018 warming at 20 m amounts to 0.64-0.75 °C (Fig. 7b). The steep NE cliff constitutes an exception: there, the simulation indicates

**Table 6.** Performance metrics of the modeled firn temperatures. For each case the number of considered profiles is reported in parentheses (also see Sect. 2.2). All metrics are computed as the arithmetic mean of the respective depth-averaged values of each profile. For a uniform comparison across different profiles, both measured and modeled values were linearly interpolated to a 1 cm vertical resolution before computing deviations.

| Model run | Profiles | RMSE [°C] | BIAS [°C] |
|---|---|---|---|
| 20 m / 1 h | CG only (19) | 1.3 | -0.4 |
| | All (25) | 1.4 | -0.3 |
| 100 m / 3 h | CG only (19) | 1.4 | 0.0 |
| | All (25) | 1.6 | 0.1 |

a very slight and non-uniform tendency towards decreasing temperatures. At lower elevations, temperature trends have a much higher spatial variability, ranging from 0 °C yr$^{-1}$ in the near-temperate area to 0.13 °C yr$^{-1}$ (or 2.5 times the atmospheric warming rate) on the west-facing slopes, about 200 m away.

## 4.2 Melt amounts and dynamics

Modeled mean annual melt amounts have an extreme spatial variability (Fig. 8), broadly reflecting surface elevation, slope and aspect. Values increase from less than 1 cm w.e. yr$^{-1}$ on the steepest slopes of the Signalkuppe, to 17 cm w.e. yr$^{-1}$ at the saddle point, and about 23 cm w.e. yr$^{-1}$ on the Zumsteinspitze slope. Even higher melt amounts, exceeding 30 cm w.e. yr$^{-1}$, are simulated for the lower elevation Grenzgletscher slopes (towards the western border of the domain) and for Seserjoch. Grid average is 21 cm w.e. yr$^{-1}$. Within the overall SEB, melt represents a relatively minor component (Fig. 9): the largest mean monthly contribution, in August, is well below 10 % of the total energy turnover, and in every month sublimation is a more effective energy sink than melt. Still, refreezing at depth transfers heat deep into the snow pack (Fig. 6), compared to the slower processes of diffusion and advection which proceed from the surface. In the NE domain region – where wind scouring is strongest – annual melt amounts correspond to a significant fraction of net accumulation (Fig. 3a). With regard to temporal patterns, the entire surface was found to always refreeze at night over the modeled period. Also, no melt is simulated between November and March, with only minor amounts in April and October (up to 1 % of the annual totals at ZS; Fig. 9).

Despite the large spatial heterogeneity, and an inter-annual variability exceeding 50 %, a common trend of melt increase could be detected in the annual time series (Fig. 10): the fitted slope ranges from $(3 \pm 2)$ to $(4 \pm 3)$ mm w.e. yr$^{-2}$ across the saddle. While the trend is somewhat masked by the 2003 extreme melt year at the very beginning, it becomes statistically significant over the rest of the period ($p < 0.05$ for 2004-2018).

The EBFM shows a marked tendency towards small melt amounts: frequency of modeled melt events decays exponentially with their magnitude (Fig. 11a), and a significant fraction of total melt amounts is contributed by micro-melt events under 4 mm w.e. in a single day (Fig. 11b).

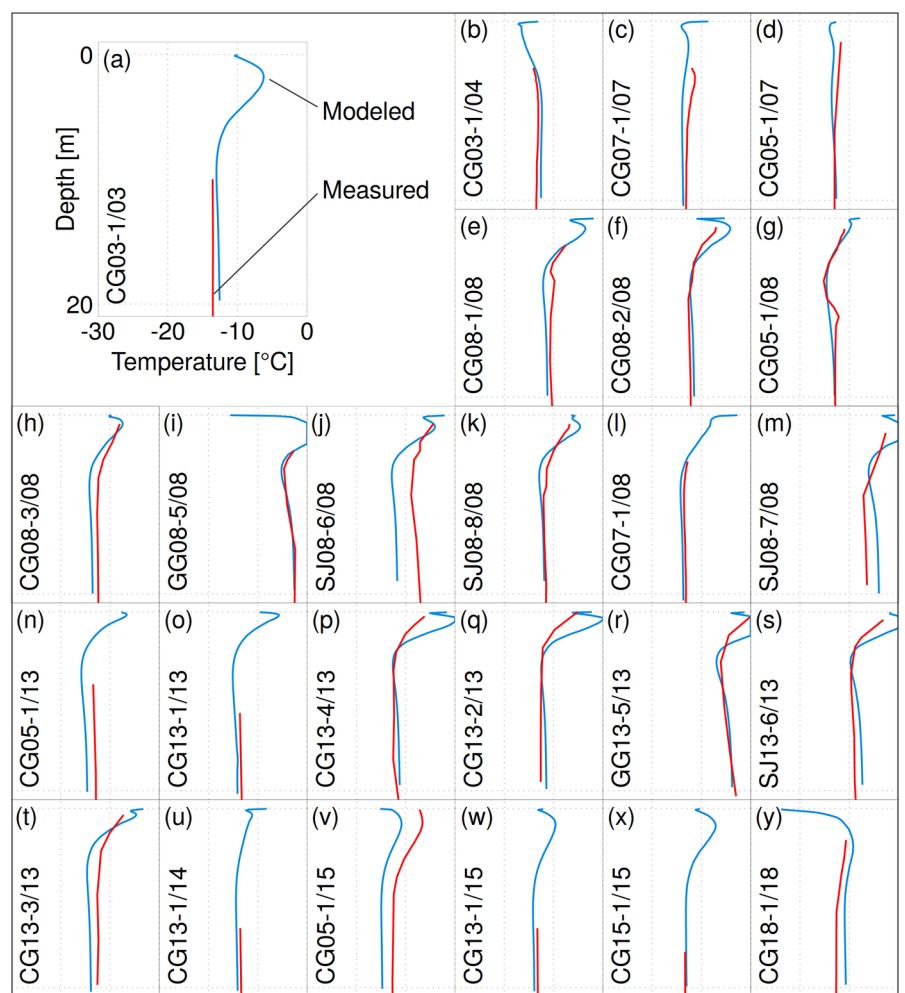

**Figure 4.** Measured and modeled borehole temperature profiles, sorted chronologically. Axes range is the same in all plots. The bottom depth of 20 m corresponds to the deepest simulated values and to the depth of zero amplitude of the annual cycle. Profile codes follow the scheme XXYY-Z/WW, with XX location code, YY borehole year, Z borehole number, and WW year of the profile measurement (Hoelzle et al., 2011; GLAMOS, 2017). Letter codes correspond to Fig. 5.

Investigation of the weather conditions leading to melt occurrence (Fig. 12) reveals the relationships between weather variables and surface melt. Air temperature provides a critical control over melt rates, unlike cloud cover which appears to have almost no effect (Fig. 12a). The majority of melt amounts happens at slightly positive air temperatures (Fig. 12b), which correspond to clear-sky conditions (cloud cover < 0.1) in more than 50 % of the cases. Still, significant melt is simulated between -5 and 0 °C: at SP, SK and ZS melt at sub-freezing air temperatures accounts for respectively 22, 34 and 17 % of the total simulated amounts. Non-zero (though minimal) melt amounts are modeled down to air temperatures of -7 °C, under clear skies and moderate to high humidity. Conversely, in very dry conditions (Fig. 12c/d) sublimation losses hinder melt even at slightly

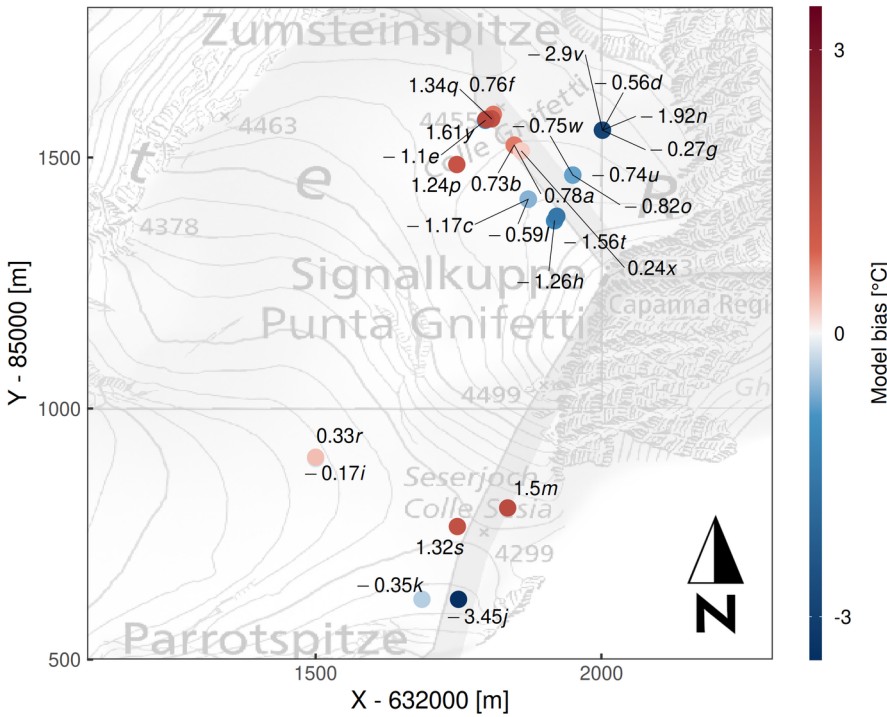

**Figure 5.** Map of depth-averaged residuals of modeled borehole temperature profiles (20 m grid, 1 h time-step). Letter codes correspond to Fig. 4. X and Y are metric CH1903/LV03 coordinates. Topographic map source: Federal Office of Topography swisstopo.

positive temperatures. Finally, wind speed appears to have a minor effect on mean melt rates. Enhanced turbulent heat losses can be seen slightly decreasing the likelihood of melt under high winds, at air temperatures between -5 and 0 °C (Fig. 12e/f).

## 5   Discussion

At CG, the repeated, long-term investigations enable interpretation of the model output against a rich literature background. The
following sections evaluate and put into context the model results for firn temperatures, meltwater infiltration and refreezing, as well as melt amounts.

### 5.1   Firn temperatures

The EBFM shows considerable potential at simulating cold firn. In addition to reproducing individual borehole profiles, the model confirms broader patterns such as the strong firn temperature gradient towards the Grenzgletscher slopes (Suter and
Hoelzle, 2002) and the depth of zero annual temperature oscillation, at about 20 m (Hoelzle et al., 2011). This last observation indicates a realistic simulation of heat conduction within the firn pack.

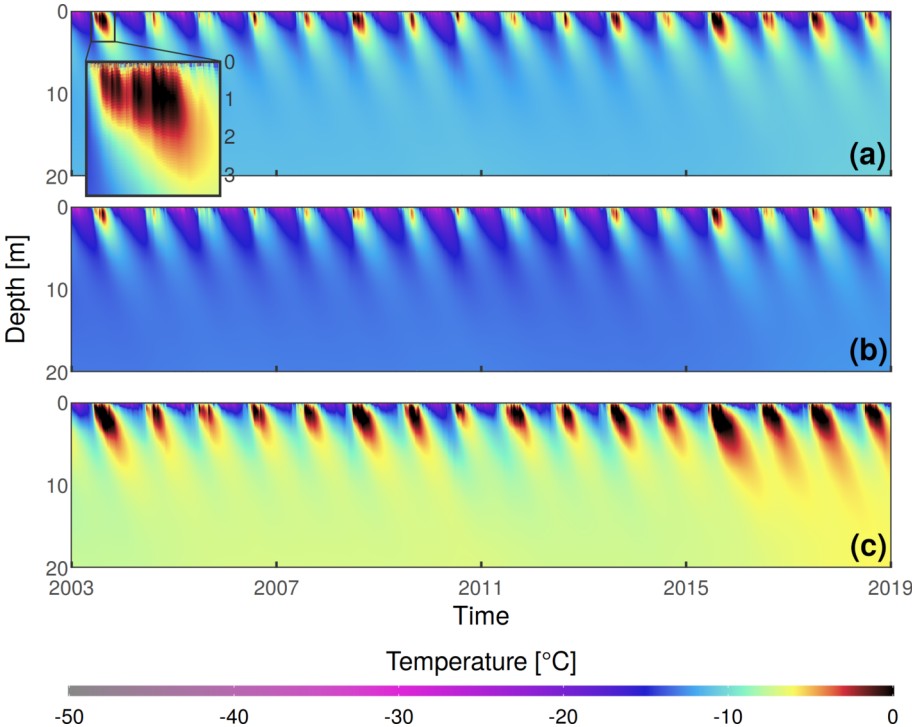

**Figure 6.** Time-depth plots of modeled firn temperatures at **(a)** SP, **(b)** SK and **(c)** ZS (map in Fig. 1a). 100 m grid, 3 h time-step. Inset shows 2003 summer temperatures down to 3.5 m depth. The annual cycle becomes smaller than 0.1 °C at approximately 15 m depth, and further decreases beyond the model quantization noise (due to the vertical layer discretization) by 20 m.

The aspect-dependent spatial pattern of temperature residuals (Fig. 5) could be affected by several factors. The locations of largest temperature under-estimation coincide with a positive bias of modeled density, by 5-30 % compared to measured core profiles below 2 m depth. This density bias leads to a higher thermal conductivity (Eq. 21), resulting in colder firn temperatures as shown in Appendix B. The density bias could be due to a lack of local calibration for the accumulation-dependent densification model, developed over Antarctic firn (Ligtenberg et al., 2011). Another possibility is a deep density increase caused by the percolation routine. Specifically, after each melt event water is distributed between the surface and the depth of $z_{lim}$ and there it refreezes, increasing density over that entire vertical extent – even in case of repeated melt-freeze cycles (which would melt a same ice surface, not increasing density). We expect this effect to be especially significant at locations where melt amounts represent a large fraction of accumulation: as such, it would be amplified by the accumulation model, which computes precipitation amounts already corrected for losses from wind scouring. Indeed, accumulation at CG results from summer precipitation events (Sect. 1), but modeled precipitation is distributed more evenly throughout the year, as it is based on weather station measurements from lower elevations (Sect. 3.3). Thus, we expect an under-estimation of the strong seasonal gradient that favours summer accumulation, so that modeled melt and refreeze can temporarily approach (or even locally exceed) the low accumulated snow amounts in summer, hence exacerbating the density bias. Simple sensitivity tests

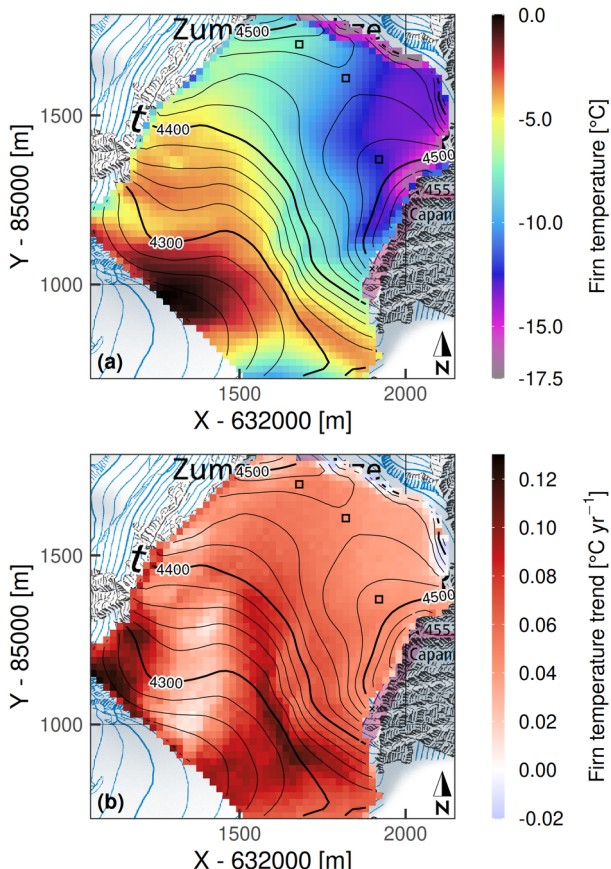

**Figure 7. (a)** Modeled 20 m firn temperatures on 31 December 2018. **(b)** Modeled 20 m firn temperature trends over 2003-2018. In both panels, marked cells correspond to the representative points of Fig. 1a. X and Y are metric CH1903/LV03 coordinates. Topographic map source: Federal Office of Topography swisstopo.

support this interpretation, by showing that the simulation of temperature profiles at low-accumulation borehole CG05-1 (Fig. 4 and 5) improves substantially with an artificial increase of precipitation amounts. Future development of the percolation model could mitigate the density bias (and thus improve the firn temperature simulation), for example introducing a dependence of percolation depth on the meltwater supply: this would prevent very small meltwater amounts from percolating to unrealistically large depths and escaping repeated melt-freeze cycles.

A deficit in summer precipitation totals in the model (with a corresponding winter excess) could also introduce a systematic deviation in the advected heat, due to a different deposition temperature of snow between summer and winter. The resulting firn temperature bias would be roughly proportional to mean accumulation (Appendix B): thus, a more extreme accumulation seasonality (with no winter precipitation) could potentially amplify the spatial pattern of modeled temperature residuals (Fig. 5). Still, complete removal of winter precipitation would be problematic for the albedo decay routine (Eq. 5). A more realistic

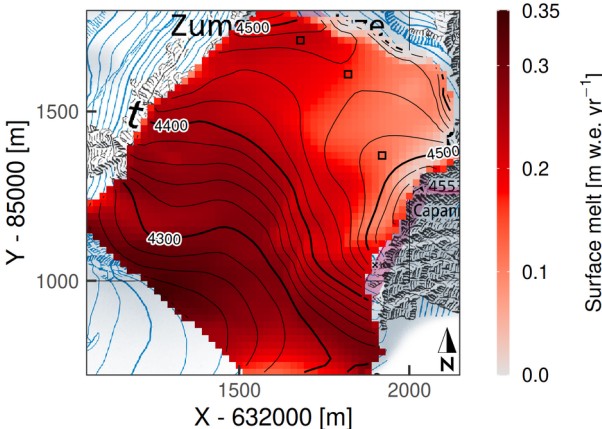

**Figure 8.** Mean (2003-2018) modeled annual melt amounts (20 m grid, 1 h time-step). Marked cells correspond to the representative points of Fig. 1a. X and Y are metric CH1903/LV03 coordinates. Topographic map source: Federal Office of Topography swisstopo.

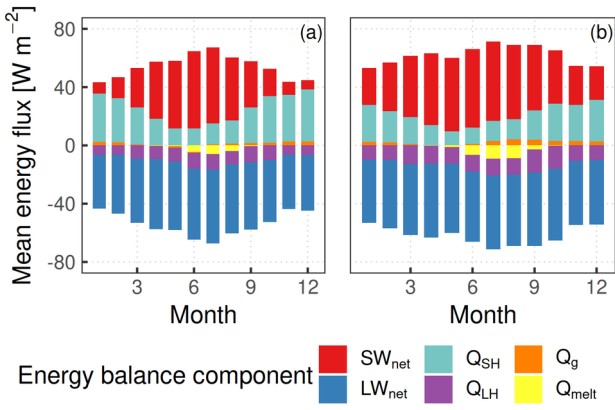

**Figure 9.** Mean (2003-2018) monthly distribution of modeled energy balance components at **(a)** SK and **(b)** ZS. 20 m grid, 1 h time-step. Distribution of fluxes at SP (not shown) is intermediate between the two.

model of wind scouring should account for the timing of snow erosion, which at CG can happen several months after deposition (e.g., Alean et al., 1983).

Some boreholes with positive model bias (Fig. 5) are simulated with too strong near-surface temperature gradients, leading to sharp positive deviations in the profile near the surface (e.g., Fig. 4f/k/p/q). Such a behavior suggests too deep refreezing is occurring in their simulation, which could again be linked to the parametrized preferential infiltration routine (Sect. 3.4). Indeed, below a depth of 0–4 m (where refreezing is occurring) the simulation appears to be unbiased at most locations. The vertical distribution of refreezing is also affected by residual saturation $\theta_{mi}$, controlling water storage after melt events and potentially affecting firn temperatures. In practice, we show in Appendix B that sensitivity to this parameter is very low in our setup, and limited to high-melt locations.

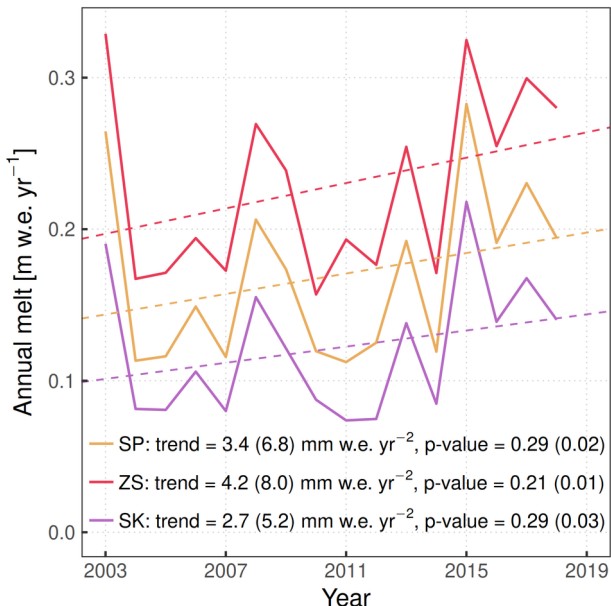

**Figure 10.** Time series of modeled annual melt amounts at three representative grid cells (solid lines; SP: saddle point, SK: Signalkuppe slope, ZS: Zumsteinspitze slope). Linear least-squares fits (each including n = 16 annual values) are shown as dashed lines. Reported annual melt trends and corresponding p-values are computed both on the whole 2003–2018 modeling period (first number), and excluding the extreme melt year of 2003 (second number, in parentheses). 20 m grid resolution, 1 h time-step.

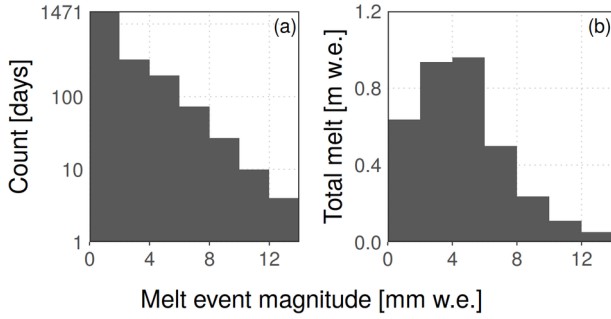

**Figure 11. (a)** Distribution of magnitudes of modeled melt events (defined as total daily melt amounts). Y-axis is logarithmic. **(b)** Cumulative melt amounts sorted by the magnitude of contributing melt events. Melt amount for each event is averaged over the whole domain of Fig. 8. 20 m grid resolution, 1 h time-step.

The spatial pattern of model residuals (Fig. 5) could in principle be affected by the lack of SW radiation reflected from the surrounding terrain in the modeled SEB. This process could induce a net energy transfer from the more sun-exposed cells towards the more shaded ones. We quantified its magnitude applying a simple Lambert reflection model (e.g., Koppal, 2014) to the modeled series of SW radiation reflected by each grid cell. We found that radiation interception can indeed introduce a

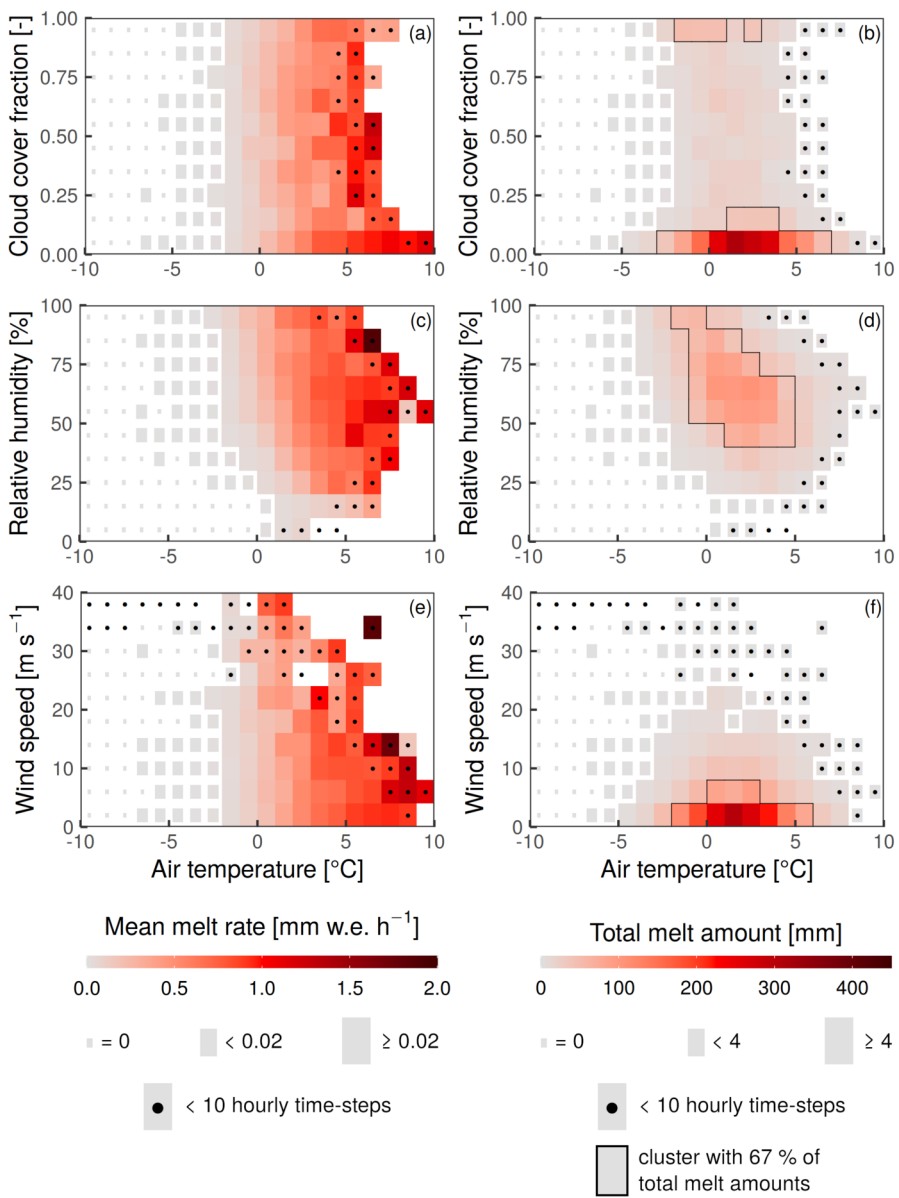

**Figure 12.** Mean melt rates (left) and cumulative melt amounts (right) modeled over 2003-2018 at the SP location (20 m grid / 1 h time-step), sorted according to the respective weather conditions: air temperature and **(a-b)** cloud cover, **(c-d)** relative humidity, **(e-f)** wind speed. Tiles of weather conditions not encountered in the input series are not drawn.

mean energy flux of up to about 3 W m$^{-2}$ in the SEB, but the net effect (taking into account mutual radiation exchange and the high surface albedo) is always smaller than $\pm 0.2$ W m$^{-2}$. As such, redistribution of reflected radiation cannot be considered a major contributor to the aspect-dependent model temperature residuals. Another effect which is not captured by our SEB

formulation is the penetration of SW radiation within the snow pack. This process – enabling sub-surface melt – challenges the EBFM assumption of water originating entirely at the snow surface, and was found to attenuate a cold bias in firn temperatures within the model of Gilbert et al. (2014b) at CdD.

The relative spatial distribution of modeled firn temperatures (Fig. 7a) is in good agreement with the interpolated result shown by Suter and Hoelzle (2002). Features such as the regular temperature gradient across the CG saddle and the near-temperate south-facing slope below 4300 m a.s.l. are reproduced well. The high spatial variability of firn temperature trends (Fig. 7b) is consistent with the borehole observations of Hoelzle et al. (2011) and the model results of Gilbert et al. (2014a) at CdD. Below 4400 m a.s.l., firn warming rates are generally high on west-facing slopes and low on south-facing ones: this could

be related to a prevalent role of solar radiation over air temperature in the firn thermal regime at south-facing locations. As warming progresses, it is possible that such pronounced spatial patterns migrate also towards the higher saddle region, where warming rates are at present more uniform. In the context of increasing firn temperatures and melt amounts, the large spatial variability of warming trends will likely have a growing importance for the localization of future ice core drilling campaigns.

       Among past modeling efforts at CG, both Lüthi and Funk (2001) and Suter (2002) formulated independent predictions for

firn temperatures evolution by 2020. As the target time frame for verification is reached, a major limitation of their modeled scenarios is found in the expected magnitude of atmospheric warming: while they assumed linear air temperature increases by respectively 0.4 and 0.45 °C between 2000 and 2020, the fitted trend over the CM AWS annual means (series in Fig. 2) amounts to $(0.05 \pm 0.03)$ °C yr$^{-1}$, corresponding to a much stronger warming of 1 °C over the period. For the CG saddle region, the two studies predict firn temperature increases at 18 m by 0.42 and 1.06 °C respectively, between 2000 and 2020. By comparison,

the EBFM simulates 0.70 °C of warming between 2003 and 2018 at SP, which can be uniformly rescaled over 2000–2020 to 0.88 °C. Such a firn warming is consistent with the results of the two studies, but lower when compared to the measured air temperature increase. A contributing factor could be the EBFM spin-up: temperatures were initialized with repeated model runs over 2004–2011, thus the initial grid at all depths is in equilibrium with the mean forcing over that period. By contrast, an adjustment time of 2–4 years is to be expected at the considered depth (Hoelzle et al., 2011). Therefore at the beginning

of the simulation the EBFM may be slightly over-estimating deep firn temperatures (Fig. 4a/b), resulting in a lower trend for 2003–2018.

       At the CdD site, Gilbert et al. (2014a) estimated firn warming by 2030 and 2050 using a thermo-mechanical coupled model, forced by three climate projections within the A1B emission scenario (Nakićenović, 2000). They found a dependence of the warming rate on advection velocities and percolation amounts. Their reported firn warming at 20 m depth is in the range 0.0-1.8

455    °C over 2010-2030 (depending on location and climate scenario), similar to our results modeled for 2000-2020 over the CG domain (Fig. 7b).

## 5.2  Meltwater infiltration

A key parameter used for model set-up is the percolation depth $z_{lim}$, which defines the maximum depth reached by infiltrated meltwater through preferential flow. In cold firn such a parameter is crucial: simulated firn temperatures (Fig. 6) indicate that

all meltwater can be expected to refreeze not far from the initial location (controlled by the parametrized vertical distribution:

Sect. 3.4). Thus $z_{lim}$ effectively determines not only the initial meltwater distribution, but also the depth of latent heat release. The effect of this parameter on firn temperatures is indeed large at all depths (Appendix B). The calibrated value of 4 m, together with the Gaussian vertical distribution, corresponds to a mean preferential percolation depth of 1.06 m. Unfortunately, in situ quantitative measurements of infiltration depths are scarce and mostly indirect. Evidence from winter snow packs and glacier accumulation areas shows that percolation and refreezing are strongly dependent on the meltwater supply amounts and rates (Marchenko et al., 2017), the temperature of the firn matrix (Koerner, 1970) and its stratigraphy (Illangasekare et al., 1990), notably affected by the previous history of infiltration and refreezing. An often observed consequence is the increase of percolation depths over the melting season (Marchenko et al., 2017). At CG, Alean et al. (1983) observed slight melting but no meltwater percolation during a warm spell in summer 1981. Suter (2002) attempted to directly track meltwater refreezing by continuously logging a temperature profile at Seserjoch in 1999, but the setup failed before the onset of summer melt. Hoelzle et al. (2011) reported evidence for increasing infiltration depths, indicating (at Seserjoch and on the Grenzgletscher slopes) a transition to a percolation regime spanning several annual firn layers. On the cold firn of CdD, Gilbert et al. (2014b) tracked sub-surface temperatures over the summer of 2012, measuring percolation depths of up to 4-5 m: these would match the value of $z_{lim}$ in our simulation. However, the stratigraphy of core *unifr-2019* (Fig. A1) revealed dry firn layers interspersed with thin ice crusts, suggesting small amounts of infiltration and refreezing, except for a thick, ice-rich layer at 4.5 m depth. Therefore in our simulation we consider $z_{lim}$ more as a tuning parameter than a realistic percolation depth.

Stratigraphy of the firn core points to a shortcoming of the subsurface model: meltwater distributed in cold firn along the parametrized vertical profile usually refreezes in place, producing a diluted density increase in the simulation. Instead, distinct ice layers are known to form at depth after preferential percolation, affecting the mechanical, hydrological and thermodynamical properties of the snow pack (Quéno et al., 2020). In the present EBFM formulation, such ice layers would not appear even with a very fine model grid. Refreezing after parametrized infiltration also distributes heat instantly over a fixed, large vertical extent, resulting in unrealistic, frequent warm pulses at depth after each melt event – no matter how small (inset in Fig. 6). These observations provide motivation for future work on a physically based percolation routine in the EBFM, accounting for the time evolution of $z_{lim}$ and its dependence on snow density and stratigraphy.

## 5.3 Melt amounts

Modeled melt amounts can be compared to the amount of refrozen ice observed in core *unifr-2019* (Fig. A1). According to the computed climatology (Fig. 3a), the core location has a mean long-term accumulation of about 50 cm w.e. yr$^{-1}$: thus the 5.5 m core should span an estimated period of about 5–6 years (with a fairly large uncertainty due to the inter-annual accumulation variability). The model predicts about 85–100 cm w.e. of melt over such a time span (Fig. 8). Instead, the core was found to contain only 31 cm of ice layers. These correspond to just 14 cm w.e. of refrozen ice after subtraction of the mean density of the ice-free sections (such a correction is fairly uncertain due to the high density variability of the profile). Some observations can be brought forward to put the apparent discrepancy into context.

First, refrozen ice amounts recorded in the core are affected by repeated cycles of melt-refreeze. Indeed, the very small amounts of meltwater produced during less intense (but rather frequent: Fig. 11) melt events can be expected to refreeze in the

very first snow centimeters. Then any subsequent melt occurring before the next snowfall would affect the same ice surface, contributing to total melt amounts but without significant increases in ice layer thickness. Such surface crusts of relatively impermeable ice have already been observed at CG (e.g., Lier, 2018). Since melt mostly happens in clustered patterns (almost only in summer and within a specific set of weather conditions: Fig. 9 and 12), contribution of repeated melt-refreeze cycles could potentially be very large.

Because ice-equivalent thicknesses of daily melt amounts are often of the same order as the size of single crystals (Fig. 11a), it is suggested that detection of some refrozen forms would require a resolution not achieved during our field analysis. This is supported by the observation in the core of sections of *icy firn*, as opposed to well-defined ice layers (see Appendix A). Investigations of such minimal melt processes are very limited in the literature. Still, Das and Alley (2005) performed hot-box experiments on the formation of thin refreeze layers in Antarctic snow, finding that a profile resolution of 1 mm was necessary to capture small-scale melt processes. Moreover, in their experiments the wetted and refrozen snow next to melt layers did not show any type of melt feature detectable in firn core stratigraphy (Das and Alley, 2005), possibly enabling some refrozen layers to remain undetected. The possibility of an overlooked vertical ice gland embedded in our core also cannot be ruled out (measured core sections were not broken up after analysis). In fact, the presence of undetected ice would be consistent with the high density variability encountered in the core profile (Fig. A1). Still, this observation could also be linked to wind compaction, and its interplay with wind erosion exposing older, denser snow.

Lier (2018) proposed an estimation of refreezing amounts at CG from the measured density anomalies over ideal dry densification profiles. At the *Sattelkern* and *Zumsteinkern* cores (respectively close to SP and to ZS) the reported refreezing rates have confidence intervals of 1–13 and 3–33 cm ice yr$^{-1}$ (Lier, 2018). The EBFM predicts approximately 19 and 25 cm ice yr$^{-1}$ of melt. On the shaded Signalkuppe flank, values span the range 0–15 cm ice yr$^{-1}$ from multiple cores; the EBFM result over the same region is between 5 and 12 cm ice yr$^{-1}$ of melt. Considering the very strong inter-annual variability of melt amounts (Fig. 10), the results are largely compatible.

The same density anomaly approach had been applied at CdD by Gilbert et al. (2014b) to estimate melt amounts of the 2011 summer season. Based on 14 firn density profiles (measured between 4230 and 4310 m a.s.l.) the authors computed melt amounts in the range 1-18 cm w.e., significantly correlated with potential incoming solar radiation. At both CG and CdD the density anomaly method shows considerable potential and consistent results, but does not account for the occurrence of repeated melt-refreeze cycles. Moreover, at CG the method involves sizable uncertainties reflecting a high variability of surface density (Lier, 2018).

The significant melt amounts modeled at sub-freezing 2 m air temperatures suggest reconsideration of degree-day models for simulating melt at high-alpine (and possibly high-latitude) locations. This is consistent with the findings of Suter et al. (2004) at Seserjoch, who observed several surface melt events but no days with positive mean temperatures over the whole 1999 summer. Indeed, laboratory experiments by Beck et al. (1988) revealed the occurrence of melt already at -4 °C with 475 W m$^{-2}$ of incoming SW radiation. At the CM AWS, values in excess of 1000 W m$^{-2}$ are a common summer occurrence (195 hours per year on average in the 2003–2018 series). This supports the plausibility of melt under even colder conditions (Fig. 12). From a theoretical perspective, Kuhn (1987) analytically explored a standard SEB equation (neglecting sub-surface

heat conduction), in relation to common weather situations on an alpine glacier. The conclusion was that melt onset can likely happen at air temperatures between -10 and +10 °C: our results appear to corroborate such a range (Fig. 12).

## 6 Conclusions and outlook

This work marks a first effort to apply a coupled, high-resolution distributed EBFM to alpine cold firn, within a multi-year simulation forced with extensively processed meteorological data, acquired at high altitude and in closest vicinity of the study 535 site. After tuning to a single measurement of deep firn temperature, we validate the model on 25 temperature profiles measured within the depth of annual temperature oscillations. In both cold and near-temperate conditions, the model achieves promising results for firn temperatures, with an average RMS error below 1.5 °C. Therefore the EBFM can be deemed suitable for further investigations of the present thermal regime as well as future temperatures evolution at cold firn sites, based on localized climate scenarios.

At CG, our results corroborate earlier observations on the spatial patterns of surface melt and deep firn temperatures. For the first time we provide a spatial estimation of distributed firn warming at the site, showing a large variability over small distances: modeled trends of deep firn temperature range from no change to 2.5 times the atmospheric warming rate. We also report a novel trend of increasing surface melt amounts, currently close to statistical significance despite high inter-annual variability and the brevity of the time series. These observations confirm the potential for accelerated changes in the thermal regime and 545 firn facies of the site (Hoelzle et al., 2011). Further developments should be closely monitored, since in the near future they could affect the suitability of CG for retrieving climate records from ice cores.

A previously unreported feature in melt dynamics is the occurrence of micro-melt events, with daily amounts below 4 mm w.e.: these remain difficult to detect, but our analysis hints at a possibly significant impact on melt totals and firn temperatures, hence on calibration and ground-truthing of model results. More field observations are needed to verify the occurrence and 550 improve the understanding of such events, and assess their potential effect on subsequent water infiltration.

Our model results point to significant melt happening at negative 2 m air temperatures, confirming earlier field observations: this would re-affirm the importance of using a full energy-balance model over a parametrized melt approach in cold conditions. Our energy-balance approach also reveals a large magnitude of the latent heat flux at the site: in every month sublimation is a more effective energy sink than melt, and latent heat losses in dry conditions prove very effective at delaying and mitigating melt 555 events. Additional field investigations of the meteorological conditions at the onset of melt would be a valuable development on this subject.

Estimations of local melt amounts and warming rates may contribute to the localization of future core drilling efforts: this provides motivation to attempt model deployment at other cold firn/ice sites and potentially on a larger scale. For this application, it is important to better constrain patterns and depths of meltwater percolation, further refining the model perco-560 lation scheme by implementing a physical infiltration routine: this would allow to overcome the limitations of a fixed-depth parametrized percolation, which presently carries a strong impact on simulated firn temperatures. The acquisition of site-

specific calibration data will be vital to support this advancement. In addition to thermal tracking, recent developments in non-destructive analysis methods (e.g., Heilig et al., 2018; Katsushima et al., 2020) could advantageously serve this purpose.

*Code and data availability.* The EBFM code used in this study is available at https://doi.org/10.5281/zenodo.4913487. Due to their large volume, modeled grids are available on request. The meteorological time series and digital elevation models should be requested from the respective providers. Full information on our processing of meteorological and topographic data (described in Mattea, 2020) is also available on simple request, including the respective software code.

## Appendix A:  Core *unifr-2019*

The 5.5 m firn core (Fig. A1) was recovered with a manually operated Kovacs Mark II corer. Firn density was measured in 20 cm sections using a digital scale, while stratigraphy was visually inspected at a resolution of 0.5 cm. Mean core density is 474 kg m$^{-3}$, with a high variability and no clear densification trend towards depth. Relatively dense snow (up to 600 kg m$^{-3}$) was encountered near the surface at around 0.5 m, with no ice layers concurrently observed. The shallowest traces of refreezing were found at a depth of about 2 m. In total, 31 cm of refrozen layers could be identified, typically less than 2 cm thick and with variable ice content. The missing 20 cm section at 3.8 m depth may have included additional ice layers. At 4.5 m several ice-rich layers mixed with icy firn were found, over a contiguous thickness of 23 cm.

## Appendix B:  Sensitivity experiments

We investigated model sensitivity to several surface and sub-surface parameters by testing single parameter perturbations, each including the respective model spin-up as in Sect. 3.5. For performance reasons we focused on firn temperature deviations at the three points marked in Fig. 1a, comparing the perturbed model output to the baseline shown throughout the paper (Fig. B1).

For surface roughness length $z_0$, we tested values of 0.1 mm and 10 mm, corresponding to the extreme ends of the range reported by Brock et al. (2006) over snow surfaces on mid-latitude glaciers. Our simulation is moderately sensitive to the value of $z_0$, with deep temperature deviations between -1.6 and +1.2 °C (Fig. B1a/b); these changes have no clear dependence on melt amounts or accumulation rates. Firn temperatures tend to decrease with $z_0$ = 10 mm, due to a strong increase in sublimation rates and a decrease in melt amounts. Conversely, the low value of $z_0$ = 0.1 mm extends the melting season from March to October, a result not supported by field evidence at CG. Overall, the observed sensitivity provides motivation to test further refinements of the EBFM turbulent fluxes routines. These could include a time dependence of roughness length, which in snow can span more than one order of magnitude over a single season (Brock et al., 2006).

The rain/snow temperature threshold $T_{s/r}$ was calibrated by van Pelt et al. (2019) against mass balance measurements in Svalbard between 1967 and 2015. The influence at CG is very small (Fig. B1c/d), due to the rarity of precipitation events at positive air temperatures: precipitation above 0 (1) °C accounts for just 1.2 (0.7) % of the total 2003-2018 amount. Such a

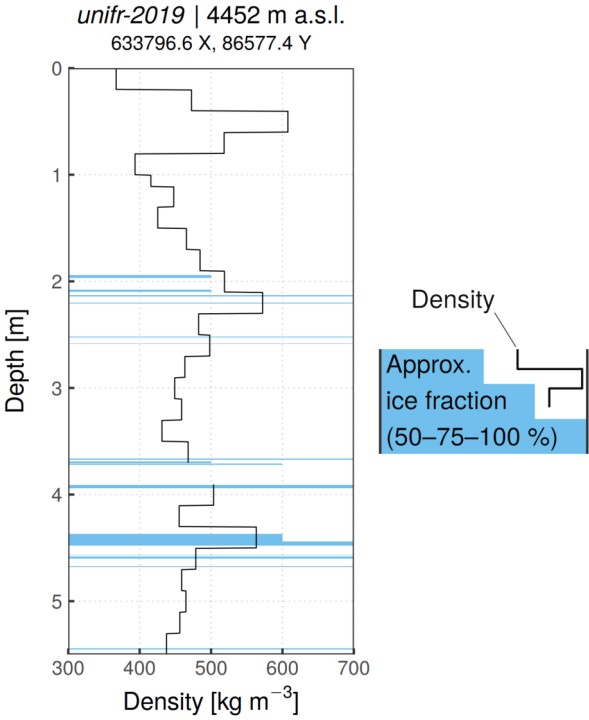

**Figure A1.** Density and stratigraphy of firn core *unifr-2019*. Values between 3.7 and 3.9 m are missing due to broken core during recovery. Drilling location is in metric CH1903/LV03 coordinates.

minor role could potentially become more significant in the future, as rainfall amounts increase amidst rising air temperatures. Thus, local calibration of parameter $T_{s/r}$ could be important for an investigation of future scenarios.

Within the sub-surface model, we tested sensitivity to residual saturation $\theta_{mi}$ and percolation depth $z_{lim}$. Residual saturation appears to have almost no effect on firn temperatures, except at the high-melt ZS location (Fig. B1e/f). This is likely due to the small meltwater amounts being distributed by the percolation routine over a considerable vertical extent, within sub-freezing snow and firn (Fig. 6): thus, all water can refreeze in place and residual saturation does not play an important role. Even at ZS, temperature deviations are within $\pm\,0.25\,^{\circ}$C after halving or doubling the parameter value. As such, $\theta_{mi}$ is not the most essential parameter for calibration in the present CG setup. Still, scenarios including higher amounts of meltwater production (Fig. 10) could be more affected by its value.

By contrast, the percolation depth parameter $z_{lim}$ provides an important control on firn temperatures, proportional to melt amounts and relatively uniform across depths (Fig. B1g/h). Compared to the tuned value of 4 m, restricting preferential percolation to the first 2 m can reduce firn temperatures by as much as 4 $^{\circ}$C at the ZS location, as a larger fraction of released latent heat can escape towards the surface (Fig. B1g).

To examine the thermal effect of summer precipitation under-estimation (Sect. 5.1), we experimented with an alternative seasonal precipitation cycle as model forcing. The complete removal of winter precipitation – to reproduce the effect of winter wind scouring – is problematic for albedo decay in the model (Eq. 5): thus, we tested an opposite change, consisting of a 50 % reduction of precipitation in May-October, redistributed over the other months to preserve the annual totals. The resulting changes in firn temperatures are strongly anti-correlated (coefficient of -0.99) with mean annual accumulation (Fig. B1i). Thus (except for albedo decay) a more pronounced accumulation seasonality in the model would likely increase firn temperatures proportionally to the mean accumulation rates. In the present setup this change could amplify the spatial pattern of firn temperature biases (Fig. 5).

Finally, we tested the recent parametrization of snow/firn thermal conductivity proposed by Calonne et al. (2019), to cover within one formula the full range of densities and temperatures found on glaciers. In the density range of interest at CG, conductivity is increased by about 20-50 % compared to the formula of Sturm et al. (1997). As a result, deep firn temperatures decrease by 1-2.5 °C, with some more differences in the seasonal cycle at shallower depths (Fig. B1j). Two factors contribute to the cooling. Melt amounts decrease by about 10 % because the higher conductivity delays the onset of melt, through a larger heat loss towards the glacier when the SEB approaches melting conditions. Also, modeled temperatures within about 2 m depth are on average colder (by up to 3-4 °C) than deep temperatures: thus, a higher conductivity shifts the deep equilibrium temperature towards colder values. The Calonne et al. (2019) parametrization of thermal conductivity will be included in an upcoming release of the EBFM.

*Author contributions.* EM performed the analysis and wrote the paper. HM and EM drilled the firn core. MH provided unpublished data and previous works. WvP supplied the EBFM model code and support to use it. MK contributed to the deployment of the model. MB provided the main meteorological data-set and extensive clarification on the CM AWS. All authors participated to the discussion of the results.

*Competing interests.* The authors declare that they have no conflict of interest.

*Acknowledgements.* We would like to thank ARPA Piemonte, PERMOS and Visit Monte Rosa for providing access to their meteorological data archives. For the meteorological series of Gornergrat and Monte Rosa Plattje, these services have been provided by MeteoSwiss, the Swiss Federal Office of Meteorology and Climatology. This work contains data/products of the Italian Air Force Weather Service. We would also like to thank Carlo Licciulli and Josef Lier (Heidelberg University) for supplying core, borehole and radar data. Swisstopo and Regione Piemonte provided the digital elevation models. Past firn temperature measurements were performed within the GLAMOS project, with funding by the Federal Office for the Environment FOEN, MeteoSwiss and the Swiss Academy of Sciences SCNAT. The legacy data collected in the Monte Rosa region was mainly organized and measured within the PhD thesis of Dr. Stephan Suter and was funded by the European Union Environment and Climate Programme under ENV4-CT97-0639 and the Swiss Government under BBW Nr. 97.0349-1, within the framework of the ALPCLIM EU project (Environmental and Climate records from high-elevation Alpine glaciers). MK, MH and

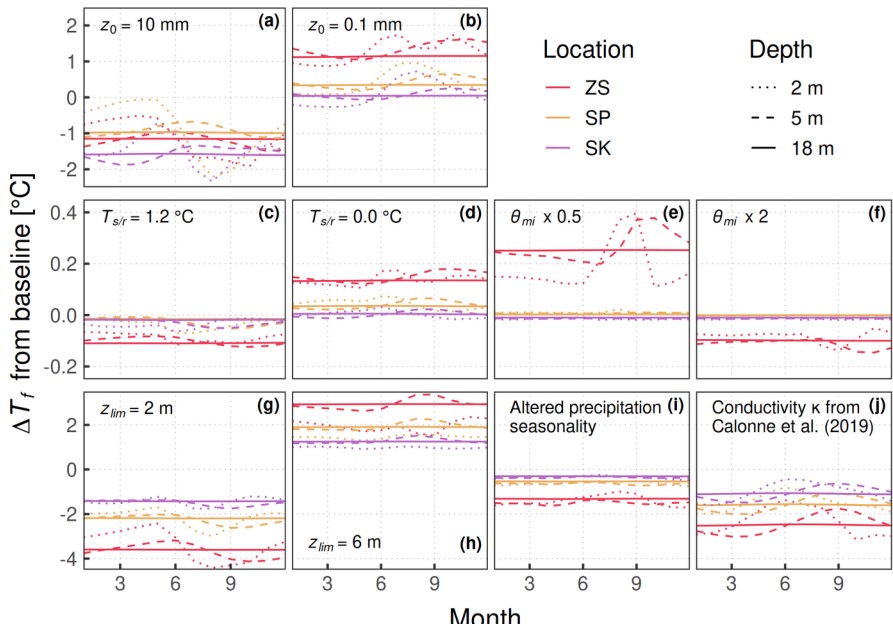

**Figure B1.** Modeled firn temperature deviations $\Delta T_f$ (2003-2018 means) by month, depth and location, after single parameter perturbations. (a)-(b): roughness length $z_0$ from 1 mm to 10 mm and to 0.1 mm. (c)-(d): rain/snow temperature threshold $T_{s/r}$ from 0.6 °C to 1.2 °C and to 0.0 °C. (e)-(f): residual saturation $\theta_{mi}$ scaled by a factor 0.5 and by a factor 2 compared to Eq. 24. (g)-(h): percolation depth parameter $z_{lim}$ from 4 m to 2 m and to 6 m. (i): precipitation amounts reduced by 50 % in May-October and redistributed over the rest of the year. (j): thermal conductivity parametrization changed from Sturm et al. (1997) to Calonne et al. (2019). The vertical scale is the same within each row. Locations are shown in Fig. 1a.

HM acknowledge support by the Swiss National Science Foundation SNSF (grant 200021_169453). WvP acknowledges funds from the Swedish National Space Agency (project 189/18). This project has received funding from the European Research Council (ERC) under the European Union's Horizon 2020 research and innovation programme (project acronym CASSANDRA, grant agreement No. 818994). We would like to thank the Editor Harry Zekollari and the reviewers Vincent Verjans, Adrien Gilbert and one anonymous, whose constructive comments and suggestions helped improving the quality of the manuscript.

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
