# Peer review of "Firn changes at Colle Gnifetti revealed with a high-resolution process-based physical model approach"

_The Cryosphere, 2020_

## Referee Comment (RC1)

Review of Mattea et al.: "*Firn changes at Colle Gnifetti revealed with a high-resolution process-based physical model approach*", by Vincent Verjans

This study uses a model-based approach to simulate changes in firn temperature at Colle Gnifetti, a region in the Swiss/Italian Alps. The authors process a meteorological dataset from a nearby weather station, corrected and complemented by using datasets from other weather stations of the region. They use an existing coupled model of Surface Energy Balance (SEB) and firn processes, which they partly re-calibrate for their site of interest. Their model simulations span the period 2003-2018 and firn temperature results are compared with 25 published measurements of firn temperature profiles. They also investigate temporal and climatic patterns in the melt rates derived from their SEB model. Based on their results, they quantify the increases in firn temperatures and surface melt amounts at Colle Gnifetti over the period of their study.

I believe that this study investigates a valuable research question and demonstrates an appropriate and relevant use of SEB and firn models in conjunction with meteorological data. Clearly, the authors have worked thoroughly on the processing of the meteorological data and on the firn simulations. They analyse in depth the important features of their results, and they make a great effort to put these into the context of previous scientific studies at their site of interest. Their conclusions are supported by their results and the manuscript is well-structured. I believe that the study will be of interest to both the firn modelling and the ice core communities. Finally, I appreciate the amount of work that went into this study. For these reasons, my review is largely positive. Nevertheless, I believe that the presentation of the calibration process is somewhat weak. Discussing the calibration more in details, in combination with a sensitivity analysis of the model parameters would bring this study to the next level. My review includes one Major and some Minor comments that I expect the authors to address in their response, and Technical comments, which are only related to the presentation of the manuscript.

Major Comment: the model calibration
I understand that the authors use the model of van Pelt et al. (2012), mostly in its original form. In Section 3 and Table 4, it is explained that many of the model parameters are calibrated using the data from Capanna Margherita (CM), the Seserjoch station and the Colle del Lys station. If I am correct, they recalibrate 6 parameters of the EBFM model ($\alpha_{fresh}, t_{wet}^{*}, K, b, e_{cl}, z_{lim}$).
Firstly, the authors do not explain their decision to re-calibrate these specific parameters, while leaving many others to their default values (see Table 4). Some parameter values are taken directly from the existing literature, without discussing the sensitivity of model results to this choice nor the particularities of their site of interest. I list here a few examples:
- $\alpha_{firn}$ is taken directly from van Pelt and Kohler (2015), which focuses on Svalbard glaciers.
- $z_0$ is taken directly from Suter et al. (2004), where this parameter value is not discussed. Given the importance of the turbulent fluxes in the SEB (see Fig. 8), I suppose that model results would be quite sensitive to this parameter value.
- According to van Pelt et al. (2012), the formulation of $t_a$ depends on the geographic location, and its parameterisation has a strong impact on results from the EBFM. Yet, the authors do not address the formulation of $t_a$, and I saw in the model code that the authors keep the same formulation and parameterisation for $t_a$ as in the Svalbard study of van Pelt et al. (2012).
- Similarly, the temperature threshold between solid and liquid precipitation is not addressed and, from the model code, I noticed that it is taken from van Pelt et al. (2019), which is another Svalbard-specific study. The air temperature is regularly above 0°C in summer at CM (Figure 2), and I thus expect this parameter to be influential.
Secondly, the calibration method for the recalibrated parameters should be better explained than the simple statement "*Tuned from*" (Table 4). Calibrating 6 model parameters implies that the number of degrees of freedom in the calibration is high. How do the authors account for potential interactions between some of the parameters? And how do they reach their final recalibrated values?
Thirdly, the parameter $z_{lim}$ is calibrated to 20 m firn temperature, and its final value is 4 m. Thus, the influence of the $z_{lim}$ parameter at 20 m depth will be outweighed by the influence of the thermal conductivity parameterisation. For this reason, I find the relevance of tuning it to 20 m temperature questionable. Why not tune it to temperatures at shallower depths (e.g. 5 or 10 m)?
Finally, Table 4 mentions that some parameters were "*verified with CM AWS data*". What does that mean? Is there any quantifiable evaluation of the verification?

As written in the introduction of this review, I believe that a more thorough sensitivity analysis to parameter values would bring the study to a next level. The application of the EBFM in many Svalbard-specific studies has been particularly successful because of the robust calibration work that was carried out in each of these studies. I understand that a full sensitivity analysis would require a considerable amount of work. Thus, at least, I recommend that the authors provide:

(1) More details about the calibration method used for the parameters that they do calibrate. This could be included in an additional section in the Appendix for example.
(2) A discussion about the limitations related to the absence of calibration for other important parameters (see above).

Minor Comments
1) The spatial contextualisation
The manuscript is well-written and mostly easy to follow. However, I had difficulties with the numerous mentions to specific locations, which are not clearly identified and not all are shown on a map. Clearly, the authors are familiar with the region of Colle Gnifetti, but they should keep in mind that most of the readers are not. For this reason, I suggest a few possibilities to improve the spatial information.

First, the authors should put all the locations referred to on the Fig. 1a map. Some of the locations that can be added are the Grenzgletscher slope, the KCC core location and the Colle de Lys station (if applicable). I also recommend expanding the legend of Fig. 1a instead of splitting the information between the legend and the caption. Finally, in the discussion of their results, the authors use terms such as CG, the saddle point, and the CG saddle to designate different things (the location of the CG grid cell or the larger area of the saddle). I believe that using CG only for the CG grid cell and using a term such as "the saddle area" would make the text less confusing.

2) Quantification
Statements in the main text often lack a quantitative support. This is an important point, especially in the Discussion section. I list some examples here:
- line 63: "*significant interannual variability*", the author could quantify the variability
- Table 3: provide also annual mean values and standard deviations, as bases of comparison for the RMSEs
- line 165: quantify the threshold required for the "*last significant snowfall*"
- Table 5: can the authors give between brackets the number of profiles considered for the "*CG only*" and "*All*" evaluations? Also, I do not understand how they compute the RMSE and Bias statistics. Do they consider all depth levels of the measured profiles? If so, how many temperature measurements are considered per core? I have the exact same questions concerning the residuals shown in Figure 5.
- line 298: "*the depth of zero annual temperature oscillation, at about 20 m*". This cannot be evaluated by the reader due to the large temperature range shown in Fig. 6, thus a quantitative metric should be given. I recommend, for example, giving the shallowest depth from which the mean interannual temperature oscillations at CG, SK and ZS are below 1°C.
- line 307: "*appears to correlate*", the author could quantify the correlation
- line 321: the authors should give the uncertainty interval on their trend. And the units should be °C yr$^{-1}$.
- line 382: "*approximately 19 and 25 cm*", why not provide the ranges of annual values through the simulation? I believe this would be more relevant because the values are compared to observed ranges.
- line 390: "*values in excess of 1000 W m$^{-2}$ are a common summer occurrence*", provide mean number of hours (or days) per year of such occurrences.
- line 406: "*significant melt happening at negative temperatures*", provide mean annual melt occurring at negative air temperature (in mm w.e. yr$^{-1}$) and/or mean fraction of the total melt occurring at negative air temperature.

3) The meteorological data processing.
The link to Mattea (2020) in the references leads to a website that cannot be accessed. As such, one cannot have the full details about the data processing method. I believe that the authors made the processing properly, but for the sake of scientific openness, all the details of the method should be available. I suggest two possible options: (1) the authors add an appendix explaining the method in details, or (2) the authors add a statement in the Data availability section guaranteeing that further details about the weather data processing is available upon request. These options are suggestions, and I believe that the editor has the final say on such issues.

4) The temperature and pressure lapse rates.
If I understand correctly, these lapse rates are calculated over a large elevation range (~2000 m according to the

elevations given in Table 2). The same lapse rates are then used over the model domain, where the elevation range is much narrower. Is it realistic to assume same lapse rate values over two ranges of elevation that are so different?

5) The extrapolation of climatic variables
The gridding of climatic forcing is not fully explained. The precipitation model is clearly detailed, and the temperature and pressure fields are adjusted via the lapse rates. However, the model must take several other climatic fields as inputs (e.g. wind speed, relative humidity). How are these calculated over the entire model domain?

6) Equation (18)
Some information about the location of the KCC core is needed (see Minor Comment 1). Is it reasonable to relate accumulation anomaly from the KCC core to wind speed at CM? Can the authors provide an intuitive, physical interpretation on why higher median wind speeds at CM should be linked to lower accumulation (maybe a link with wind scouring)?
Also, I could not find the accumulation anomaly data from Bohleber et al. (2018). How did the authors get this information? If it is through personal communication from Bohleber et al., it should be specified in the manuscript.

7) Discussion of the cold bias (lines 307-317).
The authors conclude that the cold bias of the model is due to dense, thick refrozen firn layers generated at the surface, due to the parameterised percolation, that block further infiltration and latent heat release. While this may have an impact, I think that there is a more important factor at play. If summer accumulation is underestimated, this leads to an underestimation of heat advection in the modelled firn column. Firn layers are deposited at the surface temperature of the time step. Subsequently, they are buried into the firn column, carrying this temperature signature towards greater depth. If summer snowfall events are underestimated, the amount of heat transported towards depth in this way is greatly underestimated. In my view, this could be the primary cause of the cold bias, and I would welcome the opinion of the authors about this thought.

8) Calculation of refrozen ice fraction in the *unifr-2019* core.
The authors compare their value of 31 cm of ice layers to melt amounts. Do they account for the fact that 31 cm of ice layers is not equivalent to 31 cm ice equivalent of refreezing? Meltwater refreezes in firn that has a density $>0$ kg m$^{-3}$. As such, multiplying the ice layer thickness by the ice density does not give the amount of refrozen water. Also, I note that there is no data between 3.7 and 3.9 metres depth, which seems to be an ice rich section of the core. I believe that the authors should mention this in their discussion in section 5.3.

Technical Comments
line 2
Add comma after "*Thus*".
line 26
I believe that "climatic archives" would be more appropriate than "*atmospheric archives*".
line 27
Change "*Beside*" to "Besides".
line 27
I find the wording "*mass losses brought upon glaciers by*" strange. I suggest "glacier mass losses caused by".
line 32
Why "*naturally*"?
line 35
Change "*Then*" to "Thus,".
line 42
Provide date range of the Pleistocene.
line 42
Change "*more*" to further back".
line 50
Because the study of Haeberli and Funk (1991) is quite old, specify the date range over which the steady state conditions were observed.
line 53
Change "*exposing*" to "highlighting".
line 61

Add comma after "*Thus*".

line 68

Change "*independent on*" to "independent of".

Table 1

Change "*CM*" to "Capanna Margherita".

Table 2

Is it possible to add a column $\Delta T_{CM}$ (difference in annual mean temperature with respect to CM)?

Table 3 and Table 5

Change "*RMS*" to "RMSE" (root mean square in itself is something different).

Figure 2

I suggest showing the daily mean values rather than the hourly values. I think it would give a better picture of the short-term variability, but I leave this choice to the authors.

line 121

Change "*Beside*" to "Besides".

line 148

I believe it is more relevant to refer the reader to van Pelt et al. (2012), which includes the model equations.

line 170

I believe that "$max(T, T_{max,t^*})$" should be $\left| max(T, T_{max,t^*}) \right|$ because temperature is expressed in °C.

line 172

Make sure to use $T$ only for a single variable (it is used for air temperature in the rest of the manuscript, and not snow temperature).

Table 4

The notation $b$ is used for two different parameters. And what is $Q_{ground}$ used for in the model?

line 180

Is vapor pressure the same as relative humidity? If so, I recommend sticking to the same wording as in the rest of the manuscript (e.g. Table 2 and Figure 2).

line 180

The variable $n$ was already defined.

line 180

The constant $\sigma$ should be defined as the Stefan–Boltzmann constant.

line 184

Change "*independent on*" to "independent of".

lines 217-218

"*Accumulation measured at each stake over a single year was re-scaled to a mean annual estimate by using the overlaps with firn cores and GPR points.*" This is not clear to me.

line 246

I am not sure that the use of the word "*conspicuous*" is appropriate here.

line 255

Change "*shows*" to "show".

line 264

"*Across the CG saddle*": does that mean over the entire domain (see Minor Comment 1)?

line 272

Specify that "*10%*" refers to a percentage of the sum of all the absolute energy fluxes.

lines 273-274

But the NE domain region is also where the melt amounts are lowest (see Fig. 7). Are the authors certain that it is this region where melt corresponds to the highest fraction of net accumulation?

lines 282-283

"*micro-melt events under 4 mm w.e.*": specify "under 4 mm w.e. in a single day".

lines 281-283

In my opinion, it would be interesting to investigate whether the importance of micro-melt events in the total melt amount tends to increase/decrease over time. For example, the authors could provide the trend in the ratio Total melt from melt events below 4 mm w.e. per day divided by Total melt. This is only a suggestion.

line 284

Change "*relationship*" to "relationships".

lines 285-286

"*unlike cloud cover which appears to have almost no effect*". As far as I understand, this contradicts the next sentence and Fig. 11b. It seems clear to me that low cloud cover values are associated with higher melt amounts. If the authors refer only to the melt rates, then the sentence should be clarified.

line 289

Why "*long-term*"?

Figure 8

The term "*GHF*" is not defined. And I believe that the surface fluxes at CG could also be shown.

line 310

Specify "the accumulation model (Eq. (17))".

line 313

Specify "based on weather station measurements from lower elevations".

lines 313-314

I suggest rephrasing: "Thus, we expect an underestimation of the strong seasonal gradient that favours summer accumulation."

lines 321-322

The trend estimate should have units °C yr$^{-1}$ (see Minor Comment 2).

line 325

I think that "*assumed*" should be changed to "measured".

line 334

Change "*refreezing heat release*" to "latent heat release".

line 335

Change "*correspond*" to "corresponds".

lines 339-340

I do not agree that the increase in percolation depth through the melting season is necessarily "*obvious*". For example, ice lenses could form and hinder future percolation of surface meltwater.

line 369

I recommend changing "*With daily melt amounts often close to the size of single crystals*" to "Because ice-equivalent thicknesses of daily melt amounts are often of the same order as the size of single crystals".

line 381

Is it necessary to provide yet other location names (see Minor Comment 1)?

line 386

Change "*sub-freezing temperatures*" to "sub-freezing 2m air temperatures".

line 396

"*This work marked*": maybe use present tense here.

lines 399-400

If the authors discuss the applicability of the EBFM to different scenarios, they must mention the limitations of the meltwater percolation scheme and of the highly site-specific calibration procedure.

line 405

Change "*also*" to "and".

line 406

Change "*negative temperatures*" to "negative 2m air temperatures".

line 409

I suggest changing "*on the edge of statistical significance*" to "close to statistical significance despite high inter-annual variability and the brevity of the time series".

line 410

Change "*site suitability*" to "suitability of the site".

line 423

Change "*first*" to "shallowest".

---

## Referee Comment (RC2)

[referee-annotated manuscript omitted]

---

## Referee Comment (RC3)

Review of Mattea et al., 'Firn changes at Colle Gnifetti revealed with a high-resolution process-based physical model approach'

In this study, the authors present a coupled energy balance and firn model and compare the model's output to a large dataset of firn temperature records, as well as one firn core record of refrozen melt, at Colle Gnifetti. The authors quantified the increase in firn temperature as well as surface melt totals in this location over the period of 2003-2018. Improving surface energy and firn models is an important pursuit, especially under a warming climate scenario and the uncertainties in firn meltwater retention capabilities.

This is a nicely organized and clearly written manuscript. Additionally, the figures are logically organized and easy to interpret (with a few minor suggestions for improvement below). Please find my general and line-specific comments below.

General comments:
Throughout the Discussion section, there were many mentions of imprecise comparisons and statements of significance without any quantification. The Discussion would be improved by the incorporation of values that justify statements of significant changes, variability, and appearances of correlation.

Further explanation of how the authors calculated the amount of refrozen melt was present in the unifr-2019 firn core would be helpful. How were the 31 cm of refrozen layers in the core, which certainly contained a mixture of ice and firn layers, converted to m w.e.?

The study highlights that there are still many unknowns with respect to predicting the depth of refreezing meltwater in a firn column. The authors mention that the percolation routine for the EBFM needs to account for the firn density and stratigraphy in order to improve the estimates of z_lim. Additionally, the microstructure of firn layers as well as the permeability of both undisturbed firn layers and those containing refrozen meltwater will be important for accurately estimating these depths of percolation.

Figure 1:
- The '(a)' and '(b)' labels in the two panels of the figure are hard to notice. Consider enlarging the labels or bolding the font.

- The legend symbol for the model cells is confusing because it's the same color as only one of the sites (SK). Perhaps make the legend symbol a neutral color to make it clearer that you're referring to all of the square boxes in the figure.

- It's not clear how the areas depicted in the two panels overlap. Consider adding a marker in panel (b) to designate where the CG study site is.

Figure 3:

- Which of the firn core sites in panel (a) is the KCC core site? Indicating this information would give more context to the results shown in panel (b) as well as areas of the text where KCC is mentioned.

Figure 6:
- The '(a)' and '(b)' labels in the two panels of the figure are hard to notice. Consider enlarging the labels or bolding the font.

Figure 8:
- It would be helpful to remind the reader of what the energy balance component acronyms stand for in the caption, especially 'SHF', 'LHF', and 'GHF' which are note explicitly defined in the text before this figure.

Line Specific Comments:
Line 27: 'Besides' instead of 'Beside'
Line 35: somewhat awkward transition here. What 'Then' is referring to is vague?
Lines 48-49: is this the current range of firn temperatures, or the range measured in the 1976 campaign?
Line 68: change 'on' to 'of' at the end of this line
Line 121: 'Besides' instead of 'Beside'
Lines 234-235: It's not immediately clear why the depth of 4 m in this study matches the firn temperatures of the CG saddle point at a depth of 20 m. What data was compared to determine the 4 m depth for z_lim?
Line 314: remove 'in' after 'As such,'
Line 322: should these units be $°C \ yr^{-1}$?

---

## Author Comment (AC1)

**Firn changes at Colle Gnifetti revealed with a high-resolution process-based physical model approach**

Enrico Mattea *et al.*

**Response to reviewer 1 (Vincent Verjans)**

*This study uses a model-based approach to simulate changes in firn temperature at Colle Gnifetti, a region in the Swiss/Italian Alps. The authors process a meteorological dataset from a nearby weather station, corrected and complemented by using datasets from other weather stations of the region. They use an existing coupled model of Surface Energy Balance (SEB) and firn processes, which they partly re-calibrate for their site of interest. Their model simulations span the period 2003-2018 and firn temperature results are compared with 25 published measurements of firn temperature profiles. They also investigate temporal and climatic patterns in the melt rates derived from their SEB model. Based on their results, they quantify the increases in firn temperatures and surface melt amounts at Colle Gnifetti over the period of their study.*

*I believe that this study investigates a valuable research question and demonstrates an appropriate and relevant use of SEB and firn models in conjunction with meteorological data. Clearly, the authors have worked thoroughly on the processing of the meteorological data and on the firn simulations. They analyse in depth the important features of their results, and they make a great effort to put these into the context of previous scientific studies at their site of interest. Their conclusions are supported by their results and the manuscript is well-structured. I believe that the study will be of interest to both the firn modelling and the ice core communities. Finally, I appreciate the amount of work that went into this study. For these reasons, my review is largely positive. Nevertheless, I believe that the presentation of the calibration process is somewhat weak. Discussing the calibration more in details, in combination with a sensitivity analysis of the model parameters would bring this study to the next level. My review includes one Major and some Minor comments that I expect the authors to address in their response, and Technical comments, which are only related to the presentation of the manuscript.*

We would like to thank the reviewer for the constructive and thorough review of our manuscript. Below, we provide point-by-point answers to the major and minor comments, as well as to selected technical comments. The review text is reported in *black italic,* while our responses are in blue. Figures in this response are labeled with Roman numerals to distinguish them from figures in the manuscript. Updated manuscript figures are shown at the end of the document.

*Major Comment: the model calibration*
*I understand that the authors use the model of van Pelt et al. (2012), mostly in its original form. In Section 3 and Table 4, it is explained that many of the model parameters are calibrated using the data from Capanna Margherita (CM), the Seserjoch station and the Colle del Lys station. If I am correct, they recalibrate 6 parameters of the EBFM model ($\alpha_{fresh}$, $t^*_{wet}$, $K$, $b$, $e_{cl}$, $z_{lim}$).*
*Firstly, the authors do not explain their decision to re-calibrate these specific parameters, while leaving many others to their default values (see Table 4). Some parameter values are taken directly from the existing literature, without discussing the sensitivity of model results to this choice nor the particularities of their site of interest. I list here a few examples:*
*- $\alpha_{firn}$ is taken directly from van Pelt and Kohler (2015), which focuses on Svalbard glaciers.*

We made the choice of which parameters to re-calibrate based on the availability of local high-altitude measurements, relevance of the parameter for our site, and simplicity of verification within the model result.
Of the 6 re-calibrated parameters mentioned by the reviewer, the first 5 belong to the albedo and LW radiation routines. We made the choice to re-calibrate them because the high-resolution radiation measurements from Colle del Lys and Seserjoch (both above 4000 m and within 2 km from our site) were available for a direct comparison.
From both the Seserjoch radiation measurements and our field experience, we can affirm that exposed firn is never observed at our high-alpine site, so that local measured values of firn albedo are both not available and less relevant for the calibration. For the time-scales of albedo decay, a dry snow surface at 0 °C (corresponding to parameter $t^*_{dry}$, which we kept at the default value) is an extremely rare occurrence at our site. By contrast, both melting conditions and very negative surface temperatures are far more common, allowing a robust comparison between the measurements and the albedo model (see below for the method): therefore we re-calibrated them.

*- $z_0$ is taken directly from Suter et al. (2004), where this parameter value is not discussed. Given the importance of the turbulent fluxes in the SEB (see Fig. 8), I suppose that model results would be quite sensitive to this parameter value.*

The surface roughness length $z_0$ is poorly constrained at our site, due to the frequent scouring by extreme winds which alter the snow surface. As stated in Suter *et al.* (2004), "the surface roughness length for wind [...] was determined as 0.001 m from the [wind] profile measurements"; such measurements were performed at Seserjoch, under similar conditions to those found at Colle Gnifetti. The value of 1 mm is also within the ranges found by Essery and Etchevers (2004), who introduced the formulation of turbulent fluxes used in our study. For comparison, Gilbert *et al.* (2014) reached a calibrated value of 4 mm on alpine cold firn at 4250 m a.s.l., in a similar setting in the Mont Blanc range; the review of Brock *et al.* (2006) reports values from 0.1 to 50 mm over snow and ice of mid- and low-latitude glaciers.
We have performed two new model runs to assess the sensitivity of our simulation to surface roughness length, by changing the parameter value from 1 mm to respectively 10 mm and 0.1 mm. The resulting changes in deep firn temperature (Fig. I) are within [-1.6, +1.2] °C, with an overall firn cooling when the roughness length is increased (Fig. Ia) and vice-versa (Fig. Ib). At shallower depths, temperature deviations reach slightly larger values and follow a clear annual cycle, with a strong peak in late summer: this indicates that a change in simulated melt is likely driving the observed firn temperature changes. Indeed, the corresponding distributions of energy fluxes (Fig. II-III) clearly show an inverse dependence of simulated melt amounts on the chosen value of roughness length. In the case of low $z_0$ (= 0.1 mm), some melting is simulated at ZS over as many as 8 months per year (March to October), a result which is not supported by any evidence. Given the already high melt values simulated by the model (Sect. 5.3), we believe that this analysis points to a lower bound on the roughness length value in our simulation.

[Figure]

***Figure I:*** *modelled firn temperature change by month, depth and location, after a change of roughness length from 1 mm to **(a)** 10 mm and **(b)** 0.1 mm.*

[Figure]

***Figure II:*** *mean 2003-2018 monthly distribution of modelled energy fluxes at **(a)** SK, **(b)** ZS, with a roughness length value of 10 mm.*

[Figure]

***Figure III:*** *mean 2003-2018 monthly distribution of modelled energy fluxes at **(a)** SK, **(b)** ZS, with a roughness length value of 0.1 mm.*

Better constraining the snow roughness length at the wind-scoured CG site will be an important advancement. Model development on this front could include a time dependence of this variable, which in snow can span more than one order of magnitude over a single season (Brock *et al.*, 2006). In the revised manuscript we include a discussion of the limitations of having a constant and poorly constrained roughness length value.

> *- According to van Pelt et al. (2012), the formulation of $t_a$ depends on the geographic location, and its parameterisation has a strong impact on results from the EBFM. Yet, the authors do not address the formulation of $t_a$, and I saw in the model code that the authors keep the same formulation and parameterisation for $t_a$ as in the Svalbard study of van Pelt et al. (2012).*

Calibration of $t_a$ (as described by van Pelt *et al.*, 2012) may not be applicable to our case because we used the entire SW radiation series measured at CM to reconstruct the cloudiness series, by inverting the EBFM transmissivity routine (lines 110-111 of the original manuscript). Since the aerosol transmissivity $t_a$ was already part of the equation in this reconstruction, its formulation could not be simultaneously tuned from our series. Moreover, such a tuning would not improve the SW simulation, which in our study is rather a re-computation of the measured SW values (with the model additions of gridding and topographic shading). The same applies to the gaseous and water vapour transmissivities ($t_{rg}$ and $t_w$). By contrast, we did examine the cloud transmissivity $t_{cl}$ (and finally set the parameters to Greuell *et al.*, 1997, instead of van Pelt *et al.*, 2012: see the $t_{cl}$ discussion below), because the computed cloud cover also affects the long-wave balance, which was not simply recomputed by the model but actually simulated (Eqs. 7-9).

> *- Similarly, the temperature threshold between solid and liquid precipitation is not addressed and, from the model code, I noticed that it is taken from van Pelt et al. (2019), which is another Svalbard-specific study. The air temperature is regularly above 0°C in summer at CM (Figure 2), and I thus expect this parameter to be influential.*

Air temperatures in our 16-year hourly series are above 0 (1) °C during 2.6 (1.6) % of all time-steps. Likely due to a correlation between warm air temperatures and sunny conditions, the corresponding precipitation amounts are just 1.2 (0.7) % of the 16-year totals. Thus we expected that the rain/snow temperature threshold would have a minor impact on the overall energy balance. To quantify model sensitivity to this parameter, we have performed two new model runs, with changes of the threshold from the default 0.6 °C to respectively 1.2 and 0.0 °C. The corresponding firn temperature changes (Fig. IV) are very small: below the depth of annual variation, we observe at most ±0.13 °C at the high-accumulation ZS site, and one order of magnitude less at the two drier SK and SP (saddle point – see answer to minor comment 1). At shallower depth, deviations reach up to 0.18 °C, with a small seasonal cycle. This confirms the minor role of the rain/snow threshold at our high-altitude site. The parameter could become more relevant in the future as positive air temperatures become more common within a warming atmosphere (lines 321-322).

[Figure]

***Figure IV:*** *modelled firn temperature change by month, depth and location, following a change in the rain/snow threshold temperature from 0.6 °C to **(a)** 1.2 °C, **(b)** 0.0 °C.*

In the revised manuscript, we explain our choice of re-calibrated parameters, including a summary of the new sensitivity results presented above.

*Secondly, the calibration method for the re-calibrated parameters should be better explained than the simple statement "Tuned from" (Table 4). Calibrating 6 model parameters implies that the number of degrees of freedom in the calibration is high. How do the authors account for potential interactions between some of the parameters? And how do they reach their final recalibrated values?*

For the 5 radiation parameters, we calibrated the radiation routines independently of the full model runs, using the measurements of Colle del Lys and Seserjoch. For the albedo routine, considering 10-minute measurements of albedo and snow surface temperature, we reproduced their series using the EBFM equations, tuning the $\alpha_{fresh}$, $t^*_{wet}$ and $K$ values until we reached a satisfactory match (cancelling the bias and minimizing the RMSE). For the LW routine (parameters $b$ and $e_{cl}$) we did the same over the corresponding measurements. The last parameter ($z_{lim}$) belongs to the sub-surface routine and has no effect on radiation; we tuned it until the 20 m temperatures simulated at the saddle point matched the measurements reported by Haeberli and Funk (1991). Thus, we believe that our calibration method – based on single routines rather than full model runs – is not affected by the interaction between parameters.
In the revised manuscript we provide this explanation of the calibration procedure.

*Thirdly, the parameter $z_{lim}$ is calibrated to 20 m firn temperature, and its final value is 4 m. Thus, the influence of the $z_{lim}$ parameter at 20 m depth will be outweighed by the influence of the thermal conductivity parameterisation. For this reason, I find the relevance of tuning it to 20 m temperature questionable. Why not tune it to temperatures at shallower depths (e.g. 5 or 10 m)?*

Above 20 m depth, firn temperatures are affected by the annual cycle, thus the simulation of thermal conductivity could have an even more significant impact. For example, if the penetration rate of the temperature signal were inaccurate, calibration would be distorted due to a different phase of the temperature wave. Moreover, we would expect any long-term warming trend in the firn

to proceed from the (near-)surface towards depth; tuning to a relatively large depth corresponds to using measured values most unaffected by any such trend.

The revised manuscript presents these motivations explicitly.

*Finally, Table 4 mentions that some parameters were "verified with CM AWS data". What does that mean? Is there any quantifiable evaluation of the verification?*

The two parameters in question belong to the cloud transmissivity parameterisation (Greuell *et al.*, 1997; Eq. 3 of the original manuscript). These parameters determine the fraction of clear-sky SW radiation which is transmitted by clouds, and were estimated (by both Greuell *et al.*, 1997, and van Pelt *et al.*, 2012) using a quadratic fit of cloud transmissivity versus cloud amount. In the two studies, the resulting minimum cloud transmissivity (corresponding to a cloud fraction $n = 1$) is respectively 0.352 and 0.526.

Direct measurements of cloud cover are not available at our site to perform a similar fit. Still, SW cloud transmissivities at CM can be estimated by computing the ratio of measured SW radiation to clear-sky radiation (itself computed with the EBFM parameterisations: van Pelt *et al.*, 2012, Sect. 4.1). The resulting distribution histogram (Fig. V, after filtering out time-steps with very low radiation) clearly shows that cloud transmissivities below 0.526 (the vertical line on the right in Fig. V) are a common occurrence at CM, thus the parameter values of van Pelt *et al.* (2012) would not explain the observed variability (over 33 % of the estimated transmissivities are below the allowed range of 0.526–1.0). We acknowledge that this is also true (but to a lesser extent) for the Greuell *et al.* (1997) minimum value of 0.352 (left line: about 18 % of the estimated transmissivities are lower). Still, it is reasonable to assume that cloud transmissivity does not have a sharp, well-defined minimum (due to the continuous variability of cloud optical thickness depending on cloud type: e.g. Greuell *et al.*, 1997), thus the simple quadratic parametrization will necessarily leave some unexplained variance in the measurements. We also observe (Fig. V) that the density of estimated transmissivities starts to decrease quite strongly below the 0.352 value. Considering the significant uncertainties involved in the estimation of cloud fraction (e.g. Silva and Souza-Echer, 2016), we believe that the SW measurements at CM support our choice of the Greuell *et al.* (1997) parameters. The revised manuscript includes a summary of these considerations to justify the parameter choice.

[Figure]

***Figure V:*** *distribution of the estimated cloud transmissivities $t_{cl}$ at the CM AWS (hourly series, 2003-2018). The vertical lines correspond to the lowest transmissivities allowed by the parameters of Greuell* et al. *(1997, left) and van Pelt* et al. *(2012, right).*

*As written in the introduction of this review, I believe that a more thorough sensitivity analysis to parameter values would bring the study to a next level. The application of the EBFM in many Svalbard-specific studies has been particularly successful because of the robust calibration work that was carried out in each of these studies. I understand that a full sensitivity analysis would require a considerable amount of work. Thus, at least, I recommend that the authors provide:*

> *(1) More details about the calibration method used for the parameters that they do calibrate. This could be included in an additional section in the Appendix for example.*
>
> *(2) A discussion about the limitations related to the absence of calibration for other important parameters (see above).*

In the revised manuscript we add to Sect. 3 the description of our calibration choices and methods. We also include in the Appendix several sensitivity results to surface and sub-surface model parameters, with a discussion of their relevance for calibration. We hope that the changes address the reviewer's major comment.

*Minor Comments*
*1) The spatial contextualisation*
*The manuscript is well-written and mostly easy to follow. However, I had difficulties with the numerous mentions to specific locations, which are not clearly identified and not all are shown on a map. Clearly, the authors are familiar with the region of Colle Gnifetti, but they should keep in mind that most of the readers are not. For this reason, I suggest a few possibilities to improve the spatial information.*
*First, the authors should put all the locations referred to on the Fig. 1a map. Some of the locations that can be added are the Grenzgletscher slope, the KCC core location and the Colle de Lys station (if applicable).*

In the revised manuscript we add the Grenzgletscher slope and KCC core locations to the Fig. 1a map. The Colle del Lys station is located outside the map boundaries, and including it would make the simulation area (Colle Gnifetti) too small. The location of Colle del Lys relative to the Fig. 1a extent can now be seen in Fig. 1b.

*I also recommend expanding the legend of Fig. 1a instead of splitting the information between the legend and the caption.*

Done.

*Finally, in the discussion of their results, the authors use terms such as CG, the saddle point, and the CG saddle to designate different things (the location of the CG grid cell or the larger area of the saddle). I believe that using CG only for the CG grid cell and using a term such as "the saddle area" would make the text less confusing.*

CG is the standard abbreviation used in the literature to refer to the larger area of the saddle. In order to keep consistency, we have opted to rename the saddle point grid cell to "SP" throughout the revised manuscript.

*2) Quantification*
*Statements in the main text often lack a quantitative support. This is an important point, especially in the Discussion section. I list some examples here:*
*- line 63: "significant interannual variability", the author could quantify the variability*

In the revised manuscript we add a quantitative example of the variability (wind scouring which removes all snow in certain years, even eroding the previous year's layer). A more detailed, numerical quantification (beyond the simple scope of the introduction) is also presented in Fig. 3a.

*- Table 3: provide also annual mean values and standard deviations, as bases of comparison for the RMSEs*

Done.

*- line 165: quantify the threshold required for the "last significant snowfall"*

Done.

*- Table 5: can the authors give between brackets the number of profiles considered for the "CG only" and "All" evaluations? Also, I do not understand how they compute the RMSE and Bias statistics. Do they consider all depth levels of the measured profiles? If so, how many temperature measurements are considered per core? I have the exact same questions concerning the residuals shown in Figure 5.*

In the revised manuscript we add between brackets the numbers of profiles (respectively 19 and 25). We also clarify in the Table 5 caption the method which we used to compare measured and modelled profiles (linear interpolation of both to 1 cm resolution, then depth-averaging).

*- line 298: "the depth of zero annual temperature oscillation, at about 20 m". This cannot be evaluated by the reader due to the large temperature range shown in Fig. 6, thus a quantitative metric should be given. I recommend, for example, giving the shallowest depth from which the mean interannual temperature oscillations at CG, SK and ZS are below 1°C.*

In the revised manuscript we add quantitative metrics to the caption of Fig. 6, mentioning the depth at which oscillations vanish (that is, they are not observable against model quantization noise due to the discrete layers: 20 m), as well as the depth of a threshold amplitude of 0.1 °C (which corresponds to the typical accuracy of our firn temperature measurements; this depth is 15 m).

*- line 307: "appears to correlate", the author could quantify the correlation*

Done. Correlation coefficient is 0.42, which we would describe as "a moderate correlation".

*- line 321: the authors should give the uncertainty interval on their trend. And the units should be °C yr$^{-1}$.*

Done. The new value for the air temperature trend at CM is $(0.05 \pm 0.03)$ °C yr$^{-1}$.

*- line 382: "approximately 19 and 25 cm", why not provide the ranges of annual values through the simulation? I believe this would be more relevant because the values are compared to observed ranges.*

The "observed ranges" which are reported for comparison are in fact the intervals of refrozen amounts reported by Lier (2018), as shown in Fig. VI. Rather than observed ranges, they represent reconstructed confidence intervals, obtained by comparing the core densities to the Herron-Langway dry densification model under three scenarios of the surface density parameter. Thus, they are not comparable to the mentioned *"range of annual values"* because they do not represent observed inter-annual variability, but rather confidence intervals of long-term mean values. In the revised manuscript, we mention explicitly this character of the given values.

[Figure]

**Figure 6.11:** Results of the yearly amount of refrozen melt water $\overline{M}$ for all investigated firn cores as function of $PSR$ (see section 6.1). **Top:** Maximum scenario: $\overline{M}_{max}$ with $\rho_{0,min}$ used in the H&L model. **Middle:** Mean scenario: $\overline{M}_{mean}$ with $\rho_{0,mean}$ used in the H&L model. **Bottom:** Minimum scenario: $\overline{M}_{min}$ with $\rho_{0,max}$ used in the H&L model. Note that various values of $\overline{M}_{min}$ are zero. The uncertainties of those data points are zero as well, since $\Delta\overline{M}$ is proportional to $\overline{M}$. Also note the exceptional low values of $\overline{M}$ of the KCC core in all three scenarios.

*__Figure VI:__ refrozen amounts estimated at Colle Gnifetti from density anomalies with respect to ideal profiles of dry densification. Image and inner caption from Lier (2018).*

*- line 390: "values in excess of 1000 W m$^{-2}$ are a common summer occurrence", provide mean number of hours (or days) per year of such occurrences.*

Done. The mean value is 195 hours per year.

*- line 406: "significant melt happening at negative temperatures", provide mean annual melt occurring at negative air temperature (in mm w.e. yr$^{-1}$) and/or mean fraction of the total melt occurring at negative air temperature.*

The fractional value at the three representative grid cells of Fig. 1a amounts to 17, 22 and 34 % of the total melt amounts (respectively S-facing, flat and N-facing); we are adding these values to the results (Sect. 4.2).

*3) The meteorological data processing.*
*The link to Mattea (2020) in the references leads to a website that cannot be accessed. As such, one cannot have the full details about the data processing method. I believe that the authors made the processing properly, but for the sake of scientific openness, all the details of the method should be available. I suggest two possible options: (1) the authors add an appendix explaining the method in details, or (2) the authors add a statement in the Data availability section guaranteeing that further details about the weather data processing is available upon request. These options are suggestions, and I believe that the editor has the final say on such issues.*

We agree that all the details of our methods should be made available as simply as possible. As of March 2021, the link to Mattea (2020) appears to be working normally. We will also add a statement to the *Code and data availability* section guaranteeing the availability of all details upon simple request.

*4) The temperature and pressure lapse rates.*
*If I understand correctly, these lapse rates are calculated over a large elevation range (~2000 m according to the elevations given in Table 2). The same lapse rates are then used over the model domain, where the elevation range is much narrower. Is it realistic to assume same lapse rate values over two ranges of elevation that are so different?*

The elevation range for the calculation of lapse rates was usually about 1000 m (when possible, we computed lapse rates from the difference of CM to the highest stations: Stockhorn and Plateau Rosa, at 3400 m asl). We agree that such an elevation difference is still significantly larger than the ~300 m vertical extent of the simulation domain. Still, the availability of several other high-altitude stations allowed a more robust calculation of the lapse rate (computed as mean rate with respect to more than one station), as well as a quantitative estimation of the spread of such rates at each time-step. This spread – even when including stations below 3000 m a.s.l. – was usually one order of magnitude smaller than the computed rate, confirming the robustness of the calculation. Moreover, the close vicinity (in elevation) of the CM AWS to the domain, and the narrow elevation range of the domain itself, both help to minimize the impact of inaccurate lapse rate values. The revised text will include these considerations.

*5) The extrapolation of climatic variables*
*The gridding of climatic forcing is not fully explained. The precipitation model is clearly detailed, and the temperature and pressure fields are adjusted via the lapse rates. However, the model must take several other climatic fields as inputs (e.g. wind speed, relative humidity). How are these calculated over the entire model domain?*

We calculated those variables over the grid by considering their value (Fig. 2) constant over the whole domain, due to a complete lack of measurements to estimate their spatial variability. We actually experimented with simple models of atmospheric moisture variability along mountain slopes (following Feld *et al.*, 2013), but eventually discarded them due to an excessive amount of out-of-range values (relative humidity far outside the [0,100] % interval on more than half of the time-steps). We recognize that the assumption of a uniform value in space is a substantial simplification; in mountain terrain, both the wind field and cloud cover are obviously affected by topography. The inclusion into the EBFM of spatialization algorithms for these variables will constitute an important development. In the revised text we describe the gridding of those variables explicitly.

*6) Equation (18)*
*Some information about the location of the KCC core is needed (see Minor Comment 1). Is it reasonable to relate accumulation anomaly from the KCC core to wind speed at CM? Can the authors provide an intuitive, physical interpretation on why higher median wind speeds at CM should be linked to lower accumulation (maybe a link with wind scouring)?*

We have now shown the KCC core location on the map of Fig. 1a. Wind scouring is indeed the process behind the relationship of accumulation anomaly to wind speed. This is suggested in the *Introduction* (line 60), the revised text will further clarify this point in Sect 3.3. A similar wind speed - accumulation relationship was already assumed by Suter and Hoelzle (2002, page 13) to derive a qualitative spatial "accumulation index".

*Also, I could not find the accumulation anomaly data from Bohleber et al. (2018). How did the authors get this information? If it is through personal communication from Bohleber et al., it should be specified in the manuscript.*

We computed the accumulation anomaly from the KCC core profile data (discussed in Bohleber *et al.*, 2018) which were kindly provided by Dr. Josef Lier, as mentioned in the *Acknowledgements* section.

*7) Discussion of the cold bias (lines 307-317).*
*The authors conclude that the cold bias of the model is due to dense, thick refrozen firn layers generated at the surface, due to the parameterised percolation, that block further infiltration and latent heat release. While this may have an impact, I think that there is a more important factor at play. If summer accumulation is underestimated, this leads to an underestimation of heat advection in the modelled firn column. Firn layers are deposited at the surface temperature of the time step. Subsequently, they are buried into the firn column, carrying this temperature signature towards greater depth. If summer snowfall events are underestimated, the amount of heat transported towards depth in this way is greatly underestimated. In my view, this could be the primary cause of the cold bias, and I would welcome the opinion of the authors about this thought.*

Precipitation seasonality can certainly be a source of systematic under-estimation of the energy input in our model setup. In order to quantify its magnitude, we have performed a new model run with a different seasonal distribution of precipitation. A direct verification of the mechanism would have consisted in a further reduction (up to complete removal) of the already low winter precipitation amounts (last panel of Fig. 2). Unfortunately, in our model this strongly interferes with the simulation of albedo decay, because snow albedo is only reset after a certain precipitation threshold: thus, an even drier winter would lead to unrealistically low albedo values. Therefore we have opted to attempt a change in the opposite direction: for each year we have reduced precipitation by 50 % in the 6 warmest months (May-October) and redistributed those amounts over the other 6 months. The computed precipitation series has a more uniform distribution over the year (Fig. VII).

[Figure]

*Figure VII: precipitation coefficients (monthly means) after a 50 % reduction in May-October precipitation values, re-distributed over the rest of each year. As in the main model run, the 12 monthly values of each year add up to 1.*

The resulting changes in firn temperatures show a marked dependence on the location (Fig. VIII), and are strongly anti-correlated with the mean annual accumulation at the three sites (correlation coefficient -0.99). This suggests that a more realistic accumulation seasonality in the model – with a further reduction in winter and an increase in summer – would increase firn temperatures proportionally to the mean accumulation rates (except for the mentioned albedo issues). Since model bias is already positively correlated with the accumulation amounts, this correction could amplify the spatial pattern of the firn temperature biases. In the revised manuscript we mention precipitation seasonality as a systematic source of heat under-estimation in the model.

[Figure]

*Figure VIII: mean monthly deviation of the simulated firn temperatures (compared to the baseline) after changing the precipitation seasonality.*

*8) Calculation of refrozen ice fraction in the unifr-2019 core.*
*The authors compare their value of 31 cm of ice layers to melt amounts. Do they account for the fact that 31 cm of ice layers is not equivalent to 31 cm ice equivalent of refreezing? Meltwater refreezes in firn that has a density >0 kg m -3.*
*As such, multiplying the ice layer thickness by the ice density does not give the amount of refrozen water. Also, I note that there is no data between 3.7 and 3.9 metres depth, which seems to be an ice rich section of the core. I believe that the authors should mention this in their discussion in section 5.3.*

We fully agree, we are now making the comparison more informative by computing an adjusted estimate (14 cm w.e.) for the amount of refrozen ice in the core. We compute the correction from the mean density of the ice-free core sections. We acknowledge that such a computation involves a fair deal of uncertainty, due to the significant density variability in the ice-free sections of the profile, and also to the presence of "icy firn" which did not form well-defined ice layers. In the revised manuscript we update the discussion in both Sect. 5.3 and the Appendix. We also mention the possibility of some ice lost in the missing core section at 3.8 m depth.

***Technical Comments***

We wish to thank the reviewer for the careful examination of the technical aspects of our manuscript. We are implementing all recommendations from this section, except as discussed below.

*line 32*
*Why "naturally"?*

Our reasoning here is that cold firn is colder at higher elevations – hence more resilient to a warming climate.

*line 42*
*Provide date range of the Pleistocene.*

We think that the full range (starting 2,580,000 years B.P.) may not be very relevant in this context. We would instead provide the estimated age of the oldest ice at CG (19 kyr B.P.).

*line 50*
*Because the study of Haeberli and Funk (1991) is quite old, specify the date range over which the steady state conditions were observed.*

We are not sure that we fully understand this comment. As we have mentioned in the text, steady state conditions were observed by Haeberli and Funk (1991) in a 1983 borehole profile (a single profile: we have now corrected the text from "borehole profiles"). Later, the first indications of non-steady state conditions were found by Lüthi and Funk (2001) in a 1995 profile. We believe that these dates correspond to the date range mentioned by the reviewer.

*Table 1*
*Change "CM" to "Capanna Margherita".*

The abbreviation is already introduced at the end of the *Introduction* and used in the text before Table 1: thus, we think that here the manuscript composition guidelines would prescribe the use of the abbreviation.

*Table 2*
*Is it possible to add a column ΔT CM (difference in annual mean temperature with respect to CM)?*

We have carefully considered this option, but (1) the stations cover different date ranges, thus a difference in annual mean temperature would have to be computed over inconsistent periods (sometimes very short ones) for the various stations; (2) when needed, the suggested difference can be estimated from the elevation difference which is reported in the table.

*Figure 2*
*I suggest showing the daily mean values rather than the hourly values. I think it would give a better picture of the short-term variability, but I leave this choice to the authors.*

At present the figure shows 140256 hourly values for each variable; daily aggregation results in 5844 values, a number which is still well beyond the resolution of the figure: we have verified that the visual change is almost undetectable.

*line 148*
*I believe it is more relevant to refer the reader to van Pelt et al. (2012), which includes the model equations.*

The study of van Pelt et al. (2012) is cited 8 lines before, at the very beginning of the section describing the model. Here, we make the point that the model version which we use is the more recent one, introduced by van Pelt *et al.* (2019).

*line 172*
*Make sure to use T only for a single variable (it is used for air temperature in the rest of the manuscript, and not snow temperature).*

Good catch. We are renaming snow surface temperature to $T_s$.

*Table 4*
*The notation b is used for two different parameters. And what is $Q_{ground}$ used for in the model?*

We are changing notation *b* for LW radiation into *c*. For $Q_{ground}$, as mentioned at line 144: "surface temperature and melt amounts [...], together with the lower boundary condition of geothermal heat flux, drive the sub-surface evolution."

*line 180*
*Is vapor pressure the same as relative humidity? If so, I recommend sticking to the same wording as in the rest of the manuscript (e.g. Table 2 and Figure 2).*

We are not sure that we understand this comment. In Eq. 9, *VP* is vapor pressure measured in Pa, as in the original formulation of Konzelmann *et al.* (1994). While a related concept, relative humidity refers to a fraction of the saturation vapor pressure at a given temperature.

*lines 217-218*
*"Accumulation measured at each stake over a single year was re-scaled to a mean annual estimate by using the overlaps with firn cores and GPR points." This is not clear to me.*

To extend coverage of the long-term mean accumulation measurements, we included some measurements taken at accumulation stakes (stake network of Suter and Hoelzle, 2002). These stake measurements span a single year of accumulation, therefore (due to inter-annual variability) they are in principle not representative of the long-term means. Thus, out of all the single-year stakes we took the ones which were more or less at the same location as a firn core or GPR point, and compared the accumulation between the single year (from the stake) and the long-term mean (from the core/GPR point). Then we used the mean ratio of these accumulation pairs to re-scale all the single year stake readings to the corresponding long-term mean. This provided some point estimates of long-term mean accumulation at locations where ice cores and GPR profiles were not available

(Fig. 3a). This method relies on the approximation (already used in the accumulation model) that the relative spatial patterns of accumulation are constant in time. The revised text includes a better explanation of the method.

*line 246*
*I am not sure that the use of the word "conspicuous" is appropriate here.*

We agree, we are replacing it with "extensive".

*line 264*
*"Across the CG saddle": does that mean over the entire domain (see Minor Comment 1)?*

This refers to the CG saddle, that is, excluding Seserjoch and the Grenzgletscher slope where the thermal regime is different. We are adding the mention "across the saddle" to the caption of Fig. 1 in order to clarify the naming.

*lines 273-274*
*But the NE domain region is also where the melt amounts are lowest (see Fig. 7). Are the authors certain that it is this region where melt corresponds to the highest fraction of net accumulation?*

Yes, we have verified this by direct computation. This does not concern the very edge on the N side of the domain (where melt basically vanishes on the steep NE-facing slope), but rather the region to the east of the SK-saddle point line.

*lines 281-283*
*In my opinion, it would be interesting to investigate whether the importance of micro-melt events in the total melt amount tends to increase/decrease over time. For example, the authors could provide the trend in the ratio Total melt from melt events below 4 mm w.e. per day divided by Total melt. This is only a suggestion.*

This would certainly be an interesting analysis but we believe that it could be premature at this time: as we write in the conclusion, for the moment "more field observations are needed to verify the occurrence and improve the understanding of such events". We are currently working on a setup for continuous melt monitoring at Colle Gnifetti, whose results could prove helpful for such a trend analysis of micro-melt events.

*line 289*
*Why "long-term"?*

We agree that it is an unnecessary specification, we are removing it.

*Figure 8*
*The term "GHF" is not defined. And I believe that the surface fluxes at CG could also be shown.*

In the revised manuscript we are renaming all energy fluxes in the figure to be consistent with Eq. 1. We feel that adding a third panel to show fluxes at the saddle point would make this one-column

figure too busy (too small panels, or too tall a figure), with little benefit as stated in the figure caption: the flux distribution at the saddle point is intermediate between the two (already very similar) distributions at the two other points.

*lines 339-340*
*I do not agree that the increase in percolation depth through the melting season is necessarily "obvious". For example, ice lenses could form and hinder future percolation of surface meltwater.*

We agree, we are changing "obvious" into "often observed".

*line 381*
*Is it necessary to provide yet other location names (see Minor Comment 1)?*

We understand that the many location names can prove confusing. However, *Sattelkern* and *Zumsteinkern* are the names of the ice cores from which the given refreezing estimates were derived (by Lier, 2018): we feel that mentioning them is necessary for the reader to understand the origin of these estimates, and this could simplify future comparisons of melt/refreeze amounts in the area. We are making the sentence clearer by replacing "saddle point" and "south facing slope" with the location names already used throughout the manuscript (SP and ZS).

*lines 399-400*
*If the authors discuss the applicability of the EBFM to different scenarios, they must mention the limitations of the meltwater percolation scheme and of the highly site-specific calibration procedure.*

In the revised manuscript, we are adding a brief discussion of the need to overcome these limitations. We would place this later in the conclusion, where we discuss about "attempt[ing] model deployment at other cold firn/ice sites".

**References**

Bohleber P., Erhardt T., Spaulding N., Hoffmann H., Fischer H., and Mayewski P. (2018). Temperature and mineral dust variability recorded in two low-accumulation Alpine ice cores over the last millennium, *Climate of the Past* 14: 21–37. https://doi.org/10.5194/cp-14-21-2018.

Brock B., Willis I., and Sharp M. (2006). Measurement and parameterization of aerodynamic roughness length variations at Haut Glacier d'Arolla, Switzerland. *Journal of Glaciology, 52*(177), 281-297. doi:10.3189/172756506781828746

Essery R., and Etchevers P. (2004). Parameter sensitivity in simulations of snowmelt. *Journal of Geophysical Research* 109(D20).

Feld S. I., Cristea N. C., and Lundquist J. D. (2013). Representing atmospheric moisture content along mountain slopes: Examination using distributed sensors in the Sierra Nevada, California: Representing Atmospheric Moisture Content in Mountains. *Water Resources Research* 49(7): 4424–4441.

Gilbert A., Vincent C., Six D., Wagnon P., Piard L., and Ginot P. (2014). Modeling near-surface firn temperature in a cold accumulation zone (Col du Dôme, French Alps): from a physical to a semi-parameterized approach, *The Cryosphere* 8: 689–703. https://doi.org/10.5194/tc-8-689-2014.

Greuell W., Knap W. H., and Smeets P. C. (1997). Elevational changes in meteorological variables along a midlatitude glacier during summer. *Journal of Geophysical Research: Atmospheres* 102(D22): 25941–25954.

Haeberli W. and Funk M. (1991). Borehole temperatures at the Colle Gnifetti core-drilling site (Monte Rosa, Swiss Alps), *Journal of Glaciology* 37: 37–46. https://doi.org/10.3189/S0022143000042775.

Konzelmann T., Vandewal R., Greuell W., Bintanja R., Henneken E., and Abeouchi A. (1994). Parameterization of global and longwave incoming radiation for the Greenland Ice Sheet. *Global and Planetary Change* 9: 143–164. https://doi.org/10.1016/0921-8181(94)90013-2.

Lier J. (2018). *Estimating the amount of latent heat released by refreezing surface melt water for the high-Alpine glacier saddle Colle Gnifetti, Swiss/Italian Alps*. Master's thesis, University of Heidelberg.

Lüthi M. P. and Funk M. (2001). Modelling heat flow in a cold, high-altitude glacier: interpretation of measurements from Colle Gnifetti, Swiss Alps. *Journal of Glaciology* 47: 314–324. https://doi.org/10.3189/172756501781832223.

Mattea E. (2020). *Measuring and modelling changes in the firn at Colle Gnifetti, 4400 m a.s.l., Swiss Alps*. Master's thesis, University of Fribourg. Available online at https://bigweb.unifr.ch/Science/Geosciences/GeographyTechnical/Secretary/Pub/Publications/

Geography/SelectedBachelorMasterThesis/2020/
Mattea_E._(2020)_M_Measuring_modelling_changes_Colle_Gnifetti.pdf.

Silva A. A., and Souza-Echer M. P. (2016). Ground-based observations of clouds through both an automatic imager and human observation. *Meteorological Applications* 23(1): 150–157.

Suter S., and Hoelzle M. (2002). Cold firn in the Mont Blanc and Monte Rosa areas, European Alps: spatial distribution and statistical models. *Annals of Glaciology* 35: 9–18.

Suter S., Hoelzle M., and Ohmura A. (2004). Energy balance at a cold Alpine firn saddle, Seserjoch, Monte Rosa. *International Journal of Climatology* 24(11): 1423–1442.

van Pelt W. J. J., Oerlemans J., Reijmer C. H., Pohjola V. A., Pettersson R., and van Angelen J. H. (2012). Simulating melt, runoff and refreezing on Nordenskiöldbreen, Svalbard, using a coupled snow and energy balance model. *The Cryosphere* 6(3): 641–659.

van Pelt W. J., and Kohler J. (2015). Modelling the long-term mass balance and firn evolution of glaciers around Kongsfjorden, Svalbard. *Journal of Glaciology* 61(228): 731–744.

van Pelt W., Pohjola V., Pettersson R., Marchenko S., Kohler J., Luks B., Hagen J. O., Schuler T. V., Dunse T., Noël B., and Reijmer C. (2019). A long-term dataset of climatic mass balance, snow conditions, and runoff in Svalbard (1957–2018), *The Cryosphere* 13: 2259–2280, https://doi.org/10.5194/tc-13-2259-2019.

[Figure]

*Updated Figure 1*

[Figure]

*Updated Figure 8*

[Figure]

*Updated Figure 9*

---

## Author Comment (AC2)

**Firn changes at Colle Gnifetti revealed with a high-resolution process-based physical model approach**
Enrico Mattea *et al.*

**Response to reviewer 2 (Adrien Gilbert)**

*This paper is a modeling study of near-surface firn temperature evolution at Colle Gnifetti (Swiss Alps) between 2003 and 2018. The study uses a collection of unique meteorological dataset from high elevation to force a distributed surface energy balance coupled with a firn-pack model (Van Pelt et al., 2012). This study has the potential for an excellent scientific contribution regarding the quantification of thermal changes happening in cold accumulation area in response to atmospheric warming. I really appreciate the effort put by the authors in building the meteorological dataset based on an impressive amount of data to force their model. The use of a full surface energy balance associated with a representation of melt water percolation and refreezing allows to capture the firn temperature spatial pattern observed in the unique collection of temperature measurement realized at Colle Gnifetti over the last (almost) 20 years. The paper is also pretty well written and structured.*

*However, the manuscript suffers at this stage of incomplete or inadequate referencing to previous studies in the introduction and along the text and more importantly of the absence of sensitivity test regarding the subsurface model parameters. The consequence is that the discussion concerning model bias is not convincing and poorly supported. The parameters value of the sub-surface model as well as its mathematical description are absent which is critical for a paper focusing on firn temperature.*

*The manuscript therefore clearly needs major revision before publication. I hope to help to improve its weaknesses by highlighting the major points to revise in my general comments bellow and by providing a list of specific comments embedded in the attached PDF.*

We wish to thank the referee for the thorough and constructive review, which is significantly helping to improve the quality of our manuscript. Below, we provide point-by-point answers to the referee's comments. The review text is reported in *black italic*, while our responses are in blue. Figures in this response are labeled with Roman numerals to distinguish them from figures in the manuscript.

***General Comments***

*In the general introduction paragraph, the referencing to relevant studies in really poorly done (line 16 to 38). The same reference about the use of ice core archive is used multiple times when there is lot of other more relevant and specific studies. Please do a proper research in the literature. Also you cite Master degree thesis (inaccessible and not reviewed) when relevant published work exists. See my specific comments in the attached PDF.*

In the revised manuscript we substantially improve the referencing, by citing specific, peer-reviewed studies relevant to the mentioned subjects. Later in this document we provide an extract of the introduction with updated references. We have opted to maintain some general references (e.g., Haeberli and Beniston, 1998, and Wagenbach *et al.*, 2012) as we think they provide a general context relevant as background for our study. In particular, the Introduction section of Wagenbach *et al.* (2012) presents a detailed comparison of the characteristics of alpine ice cores in relationship to polar ones, as well as a comprehensive description of ice coring projects at both Colle Gnifetti and Col du Dôme. We will keep references to Master's degree theses only in relation to relevant work carried out at Colle Gnifetti within these theses, and not published elsewhere (specifically, the estimation of refreezing amounts done by Lier, 2018; a mention of the energy balance and firn model of Buri, 2013; and the reference to Mattea, 2020, for detailed information on the weather stations around Colle Gnifetti). The URLs to the full text of Buri (2013) and Mattea (2020), which are openly accessible, will be added to the References.

*Also concerning past studies, you mostly ignored other studies done on the same topic and for very similar setup. I am probably oversensitive to it since it concerns my work but lot of the work done in Gilbert et al. 2014a and Gilbert et al. 2014b should be discussed and compared to your results. You will see many reference to it in my specific comments.*

In the revised Discussion section we compare and discuss results from Gilbert *et al.* (2014a, 2014b).

*The description of the sub-surface model should be included in the paper. There is no reason to describe the surface energy balance and not the energy transfer within the firn-pack. This is the essential part of the modeling and parameters are not even listen nor their value given.*

Our motivation to describe the full surface model but not go into the sub-surface details was that we introduced significant changes to the surface model, compared to the version of van Pelt *et al.* (2019): we re-calibrated several parameters and we changed the accumulation model and the turbulent fluxes formulation. By contrast, we kept all sub-surface routines and parameters exactly as described in van Pelt *et al.* (2012) and Marchenko *et al.* (2017), except as detailed in Sect. 3.4. We understand that our choice could prove confusing to a reader not familiar with the model, and in the revised manuscript we are adding a description of the main sub-surface routines, as well as improved referencing to the studies which first introduced the sub-surface EBFM parametrizations. Specifically, we will present the following equations governing the evolution of layer temperature and density:

$$\rho_f \, c_p(T_f) \frac{\partial T_f}{\partial t} = \frac{\partial}{\partial z}\left(\kappa(\rho_f) \frac{\partial T_f}{\partial z}\right) + \frac{F \, L_m}{\Delta z} \qquad (1)$$

$$\frac{\partial \rho_f}{\partial t} = K_g(\rho_f, T_f) + \frac{F}{\Delta z} \qquad (2)$$

$$K_g(\rho_f, T) = b_{acc} \; g \, (\rho_{ice} - \rho_f) \exp\left(\frac{-E_c}{R \, T_f} + \frac{E_g}{R \, T_{avg}}\right) \; C_{Lig}(b_{acc}) \qquad (3)$$

where $\rho_f$ and $T_f$ are layer density and temperature, $c_p$ firn heat capacity, $z$ depth, $\kappa$ effective conductivity, $F$ refreezing rate, $L_M$ latent heat of melting, $\Delta z$ layer thickness, $K_g$ gravitational densification as in Arthern et al. (2010), $b_{acc}$ accumulation rate in mm a$^{-1}$, $\rho_{ice}$ ice density, $R$ universal gas constant, $E_c$ (60 kJ mol$^{-1}$) and $E_g$ (42.4 kJ mol$^{-1}$) activation energies of creep by respectively lattice diffusion and grain growth, $T_{avg}$ year-averaged firn temperature and $C_{Lig}$ a correction based on the accumulation rate, accounting for different densification regimes above and below the critical density of 550 kg m$^{-3}$ (Ligtenberg *et al.*, 2011; van Pelt *et al.*, 2012). We are also adding mention of the parametrizations for specific heat (Yen, 1981) and thermal conductivity (Sturm *et al.*, 1997). Furthermore, we show the explicit formulation for the irreducible water content (Schneider and Jansson, 2004):

$$\theta_{mi} \; = \; 0.0143 \exp(3.3 \, n_p) \qquad (4)$$

where $n_p$ is porosity, and we add the formula of the preferential percolation routine (Marchenko *et al.*, 2017):

$$PDF(z, z_{\lim}) \; = \; 2 \frac{\exp\left(\dfrac{-z^2}{2\sigma^2}\right)}{\sigma \sqrt{2\pi}} \qquad (5)$$

where $z$ is depth, and $\sigma$ standard deviation computed as $z_{lim} / 3$, such that 99.7 % of the input water is distributed above $z_{lim}$.

*From my understanding, you do not take vertical advection into account, the vertical advective heat transport can be significant in cold accumulation zone and should be taken into account. Also what are you doing with precipitation? It is not explained, maybe the vertical advective transport is actually taking into account? Since the subsurface model is not described, it is not clear. The only thing that makes me thinking you actually do, is the thickness of your active layer reaching 20m-depth which is possible only with advection. You need to clarify this in the manuscript.*

We acknowledge that this was not clearly specified in the manuscript. Vertical advection is taken into account by the Lagrangian grid discretization (line 242): in the model, layers are free to move on the vertical axis (to prevent numerical diffusion), thus they carry their temperature signal to

depth as they are buried by progressive accumulation. This also explains one purpose of precipitation/accumulation in the EBFM: it adds snow at the top of the grid, creating new layers and pushing the others down. Accumulation also appears in the densification formula (Eq. 3). In the revised manuscript we describe the sub-surface model, including an explanation of the moving layers mechanism and an explicit mention of advection.

*The bigger weakness of the manuscript is the absence of sensitivity test concerning the sub-surface model parameters and their influence on the modeled firn temperature. You cannot discuss the model bias without it. For instance, discussing short wave radiation redistribution due to reflection in order to explain your bias is not convincing at all when many parameters modification could explain the biases. From my experience, cold biases in firn temperature model often arise from neglecting short wave radiation penetration. Gilbert et al. (2014a) show that a characteristic penetration length of 2.5 cm is able to significantly change the modeled firn temperature and explain the cold bias observed in their study site at 4250 m a.s.l. As you mention, warm bias could be explained by not accurate representation of water percolation and refreezing. I agree, but to be convincing, you have to perform sensitivity tests on the water percolation parameters and explicitly show the result of these tests. We don't even know what the real meaning of the percolation depth parameter is, since the model is not described. Also the residual saturation parameter due to capillarity force is a critical parameter which is not well constrained. I suggest to test its influence on your results, you could be able to correct your warm bias.*

In the revised manuscript, we add an Appendix section presenting sensitivity of modeled firn temperatures to several surface and sub-surface parameters, based on reduced model runs (for performance reasons) including the three points of Fig. 1a. We also improve the discussion of the model temperature biases by referring to these computed sensitivities. Moreover, we are adding a full description of the percolation routine, explaining the meaning of the percolation depth parameter $z_{lim}$ (see Eq. 5 above).
Firn temperatures are very sensitive to the value of the percolation depth parameter: a decrease from 4 to 2 m (Fig. Ia) produces a cooling by 1.5-4 °C, with a clear dependence on melt amounts. The largest change is at the high-accumulation, high-melt ZS location. A deeper value of $z_{lim}$ = 6 m (Fig.

[Figure]

**Figure I:** *firn temperature change by month, depth and location, resulting from a change of the percolation depth parameter $z_{lim}$ from 4 m to **(a)** 2 m, **(b)** 6 m.*

Ib) increases firn temperatures by 1-3 °C. Due to this high sensitivity, especially at locations with high accumulation and melt rates, it would be an interesting topic for a future study to test and compare different water percolation routines at our site, such as gravity flow theory (Colbeck and Davidson, 1973) which has been successfully applied on cold firn by Gilbert *et al.* (2014b).

By contrast, sensitivity to the residual saturation (Fig. II) is almost negligible at the two low-melt locations of SP (saddle point, renamed from CG) and SK: this is likely due to the small meltwater amounts being distributed by the percolation routine over a vertical extent of 4 meters, such that all water can refreeze immediately and residual saturation does not play an important role. At the high-melt ZS location, residual saturation begins to show some small effects: firn temperatures increase by about 0.25 °C by halving the residual saturation compared to the baseline (Fig. IIa), and decrease by 0.1 °C with the opposite change (Fig. IIb). Thus, we would conclude that residual saturation is not the most critical parameter for calibration in the present Colle Gnifetti setup; still, this parameter would probably be more relevant when modeling scenarios of future evolution, which are expected to include more meltwater production.

[Figure]

***Figure II:*** *firn temperature change by month, depth and location, resulting from a change of residual saturation by a factor **(a)** 0.5, **(b)** 2, from the formula of Schneider and Jansson (2004).*

We have now quantified the effect of reflected radiation redistribution between Sun-exposed and shaded cells. For this, we have used a simple model of Lambert (isotropic) reflectance (e.g. Koppal, 2014), applied to our simulated series of 2003-2018 hourly reflected SW radiation over the model grid. Our estimations (Table I) for the three grid cells of Fig. 1a indicate that the magnitude of SW radiation redistribution is indeed negligible compared to the other energy fluxes. We are updating the text to reflect these findings.

***Table I:*** *mean (2003-2018) simulated energy fluxes arising from SW radiation redistribution. The net difference between received radiation and intercepted outgoing radiation is the relevant metric towards the spatial distribution of temperature biases, because absolute received radiation alone could be easily compensated by albedo calibration.*

| | ZS | SP | SK |
|---|---|---|---|
| **Mean SW radiation received by a cell from the other grid cells [W m$^{-2}$]** | 2.19 | 0.62 | 0.68 |
| **Mean SW radiation outgoing from a cell and intercepted by other grid cells [W m$^{-2}$]** | 2.83 | 0.65 | 0.72 |
| **Net difference of the previous two [W m$^{-2}$]** | -0.64 | -0.03 | -0.04 |
| **Net difference reduced by 80 % (mean albedo) [W m$^{-2}$]** | -0.14 | -0.01 | -0.01 |

Incorporating SW radiation penetration in the EBFM (in an energy-conserving manner) is unfortunately not straightforward. This is because melt amounts would no longer be computed simply from the surface energy balance and surface temperatures. Penetration of SW radiation into the sub-surface implies that melt can happen inside the snow pack instead of originating entirely at the surface. In fact we would expect that a significant fraction of modeled melt would happen in the (shallow) sub-surface: the reason is that (with SW penetration) the energy balance of an infinitesimally thin surface layer will have in principle no incoming SW component (e.g., Kuipers Munneke *et al.*, 2009), thus the SEB would rarely reach melting conditions. Simulation of this process would require a major restructuring of the model architecture, going beyond the scope of our study. Due to the relatively shallow penetration depths (e.g., Warren, 1982; Fukami *et al.*, 1985), we anticipate that including the penetration of SW radiation would require significantly thinner near-surface layers in the model compared to our 5-10 cm layers. A realistic simulation of radiation penetration should also include the non-exponential decay of incoming flux close to the surface, due to non-uniform spectral extinction (Warren, 1982; Beaglehole *et al.*, 1998). In the revised manuscript, we add a paragraph discussing the issue of radiation penetration, mentioning the process as a potential contributor to our aspect-dependent temperature bias.

*What about the firn thermal conductivity? Recent work of Calonne et al. (2019) should be used. The author corrected a significant bias on the commonly used conductivity/density relationship.*

In the revised manuscript we provide information on which parametrization of thermal conductivity is used in the EBFM (Sturm *et al.*, 1997). We agree that the formula of Calonne *et al.* (2019) should become the norm in firn modeling. Due to the very high computational cost of performing a new full EBFM run (spin-up and actual simulation), we have opted to test the Calonne *et al.* (2019) parametrization within a reduced model run, consisting of the three model cells highlighted in Fig. 1a. These locations are representative of the varying conditions of accumulation and melt found across the CG saddle. In the reduced model run (still at 20 m resolution and 1 h time-step) we have applied the full-grid topographic shading routine to ensure consistency with the original "baseline"

model result. We will provide the result in the new Appendix B, in terms of the sensitivity of firn temperatures to the change of parametrization. In the firn density range of interest at CG, the Calonne *et al.* (2019) formula increases conductivity by about 20-50 % compared to Sturm *et al.* (1997). As a result, deep firn temperatures decrease by 1-2.5 °C, with some more differences in the seasonal cycle at shallower depths (Fig. III). We interpret this cooling as the result of two factors: (1) melt amounts decrease (about 10 %) because the higher conductivity delays the onset of melt (larger heat loss towards the glacier when the SEB approaches melting conditions); (2) modeled near-surface temperatures are on average colder (by 3-4 °C) than deep temperatures, thus a higher conductivity shifts the deep equilibrium temperature towards colder values. In the revised manuscript we present this discussion of thermal conductivity; the Calonne *et al.* (2019) parametrization will also be included by default in an upcoming release of the EBFM.

[Figure]

***Figure III:*** *firn temperature change by month, depth and location, resulting from a change of the thermal conductivity parametrization from Sturm* et al. *(1997) to Calonne* et al. *(2019).*

*My final general comment is about the presentation of the results. You have a really nice distributed model but you do not really use it to show the spatial heterogeneity of the firn warming which would be a valuable result. I suggest to add a map of current firn 20m-depth temperature and a map of the associated warming rate. You will see it in my specific comments in the attached pdf.*

In the revised manuscript, we provide maps of current firn temperatures and 2003-2018 trends; they are also shown below (Fig. IV). The relative distribution of firn temperatures is consistent with the result of Suter and Hoelzle (2002), confirming the observed strong spatial gradient towards the western region of the domain. Warming rates have a relatively complex distribution, likely affected by the relative importance of incoming solar radiation and air temperature on the present-day firn temperatures. The slower warming rate in the near-temperate region matches the observations of Hoelzle *et al.* (2011).

[Figure]

*Figure IV: (a)* modeled 20 m firn temperatures at Colle Gnifetti on 31 December 2018; *(b)* modeled 20 m temperature trends over 2003-2018.

*Specific Comments*

*You will find a list of specific comments embedded in the attached pdf. They are sometimes redundant with my general comments but will help to clarify them.*

In the revised manuscript we are implementing all the recommendations from the specific comments, except as noted below or (for repeated subjects) in the corresponding general comments.

*lines 33-34*
*transition from cold to temperate do not necessarily mean mass loss, not very relevant I would just keep the degradation of the climatic archive.*

This observation would appear to contradict the role of cold firn as a buffer against mass losses through refreezing. As stated in Vandecrux *et al*. (2020), *"The meltwater retention capacity of the firn depends on three physical characteristics: (i) the availability of pore space to host the meltwater, (ii) the availability of cold content to refreeze the meltwater and (iii) the possibility for meltwater to percolate in deeper firn where conditions (i) and (ii) are met"*. Transition from cold to temperate corresponds to the disappearance of characteristic (ii). For the CG setting, Hoelzle *et al*. (2011) state that *"As soon as all these areas become temperate, meltwater will be released in large quantities into the water cycle [...]"*.
Thus, we would keep the mention of mass loss, while improving the referencing within this section. We would also add mention of a third consequence of the cold-temperate transition, namely the possible destabilization of cold-based hanging glaciers (Gilbert *et al*., 2015).

*line 63*
*What do you call "cold content" ? Surface accumulation control the vertical advection of the heat which influence the thickness of the active layer and the efficiency of the heat transfert toward the glacier base.*

Here we used "cold content" as the amount of energy required to bring the snow cover temperature up to freezing (e.g., Vandecrux *et al*., 2020). As mentioned by Kuipers Munneke *et al*. (2014), *"the total refreezing capacity of the firn is ultimately determined by the total cold content provided by snowfall. This cold content is linearly proportional to the accumulation rate"*. We are adding the Kuipers Munneke *et al*. (2014) reference to the statement, together with an explicit mention of heat advection.

*lines 65-66*
*It does not tell what complexity Suter is missing ? Mean snow accumulation is a good proxy of surface vertical velocity for steady state topography which roughly the case at Colle Gnifetti.*

Good catch, our description of the Suter *et al*. (2001) model was wrong. As stated in that paper, *"The following assumptions and simplifications are made:*

- *heat transfer is reduced to vertical heat conduction (vertical heat advection by surface accumulation, and corresponding downward motion of the snow and firn are neglected)*
- *latent heat (refreezing meltwater); convective heat transport by the air and liquid water; sensible heat; radiation; frictional heat by ice deformation; lateral firn and ice advection; and the ground heat flux are neglected*
- *the firn density is assumed to be constant with depth*
- *the air temperature at the surface equals the snow surface temperature (being the result of the surface energy balance) and follows a sine curve with a period of 1 year*
- *the monthly mean air temperatures are assigned to the 15th of each month."*

In the revised manuscript we provide the correct mention of missing heat advection.

*lines 315-317*
*Not really convincing argumentation to explain the cold bias. For instance, neglecting short wave penetration through the snowpack (even of a few centimeter) strongly impact the firn temperature. A sensitivity study of the firn pack model parameter is really missing in the paper.*

We agree that the argumentation here was mostly speculative and we are removing it from the revised manuscript. As mentioned above, we are also adding the new sensitivity results to the sub-surface parameters. In the discussion of the cold bias, we are adding a mention that radiation penetration could explain the bias, as reported by Gilbert *et al.* (2014b). We are also formulating another hypothesis for a process which could contribute to the cold bias, arising from the observations of sensitivity to thermal conductivity (Fig. III). Specifically, in the Discussion we had mentioned that repeated melt/refreeze cycles of the same surface could contribute to the discrepancy between modeled and observed melt amounts (Sect. 5.3). After each melt event, the percolation routine always distributes meltwater over the first 4 meters, even if melt amounts are small (the frequent micro-events of Fig. 10). As such, repeated melting is not accounted for (a new snow surface is melted each time in the model), and refreezing over the first 4 meters keeps increasing the firn density (see Eq. 2 above), especially at low-accumulation locations where the addition of new low-density layers is slow. This could result in a positive bias in modeled density, which would correspond to a positive bias in thermal conductivity: in turn, this would induce a cold bias, as shown above in the discussion about the Calonne *et al*. (2019) thermal conductivity. This explanation appears to be consistent with the observations of (1) cold bias mostly affecting the regions of very low accumulation/advection, and (2) reduced bias when accumulation (hence advection of low-density layers) is artificially increased (line 310). Indeed, even though our paper focuses on melt amounts and firn temperatures, a simple visual inspection of modeled firn densities revealed a positive bias at locations of low accumulation (Fig. Va), mostly disappearing at locations of high accumulation (Fig. Vb).

[Figure]

***Figure V:*** *measured and modeled densities for **(a)** core KCS, **(b)** core Zumsteinkern. Deep core KCS is located close to the saddle point (4450 m a.s.l.) and has a mean annual accumulation of 0.51 m w.e. (Licciulli et al., 2020). Shallow core Zumsteinkern is located at the south-facing ZS location (4470 m a.s.l.) and has a mean annual accumulation of 0.87 m w.e. (Lier, 2018).*

*line 393*
*This is not a strong constrain validating your results...*

We fully agree that this is not a validation of our results. If anything, our results are an example which corroborates the theoretical estimations of the cited paper. We feel that this reference is worth mentioning in a discussion about the conditions for melt initiation and melt occurrence at negative temperatures, which have not been examined very often in the literature. In the revised manuscript we reword the sentence to make it more neutral.

**Updated Introduction (first section)**

[revised manuscript text omitted]

Colbeck, S. C. and Davidson, G.: Water percolation through homogenous snow, The Role of Snow and Ice in Hydrology, Proc. Banff. Symp., 1, 242–257, 1973.

Fukami H., Kojima K., and Aburakawa H. (1985). The Extinction and Absorption of Solar Radiation Within a Snow Cover. *Annals of Glaciology* 6: 118–122.

Gabrielli P., Carturan L., Gabrieli J., Dinale R., Krainer K., Hausmann H., Davis M., Zagorodnov V., Seppi R., Barbante C., Dalla Fontana G., and Thompson L. G. (2010). Atmospheric warming threatens the untapped glacial archive of Ortles mountain, South Tyrol. *Journal of Glaciology* 56(199): 843–853.

Gilbert A., and Vincent C. (2013). Atmospheric temperature changes over the 20 [th] century at very high elevations in the European Alps from englacial temperatures. *Geophysical Research Letters* 40(10): 2102–2108.

Gilbert A., Gagliardini O., Vincent C., and Wagnon P. (2014a). A 3-D thermal regime model suitable for cold accumulation zones of polythermal mountain glaciers: A 3-D thermal regime model for glaciers. *Journal of Geophysical Research: Earth Surface* 119(9): 1876–1893.

Gilbert A., Vincent C., Six D., Wagnon P., Piard L., and Ginot P. (2014b). Modeling near-surface firn temperature in a cold accumulation zone (Col du Dôme, French Alps): from a physical to a semi-parameterized approach. *The Cryosphere* 8(2): 689–703.

Gilbert A., Vincent C., Gagliardini O., Krug J., and Berthier E. (2015). Assessment of thermal change in cold avalanching glaciers in relation to climate warming. *Geophysical Research Letters* 42(15): 6382–6390.

Gilbert A., Wagnon P., Vincent C., Ginot P., and Funk M. (2010). Atmospheric warming at a high-elevation tropical site revealed by englacial temperatures at Illimani, Bolivia (6340 m above sea level, 16°S, 67°W). *Journal of Geophysical Research* 115(D10): D10109.

Haeberli W., and Beniston M. (1998). Climate Change and Its Impacts on Glaciers and Permafrost in the Alps. *Ambio* 27(4,): 258–265.

Harper J., Humphrey N., Pfeffer W. T., Brown J., and Fettweis X. (2012). Greenland ice-sheet contribution to sea-level rise buffered by meltwater storage in firn. *Nature* 491(7423): 240–243.

Hoelzle M., Darms G., Lüthi M. P., and Suter S. (2011). Evidence of accelerated englacial warming in the Monte Rosa area, Switzerland/Italy. *The Cryosphere* 5(1): 231–243.

Jenk T. M., Szidat S., Schwikowski M., Gäggeler H. W., Brütsch S., Wacker L., Synal H.-A., and Saurer M. (2006). Radiocarbon analysis in an Alpine ice core: record of anthropogenic and biogenic contributions to carbonaceous aerosols in the past (1650–1940). *Atmospheric Chemistry and Physics* 6(12): 5381–5390.

Konrad H., Bohleber P., Wagenbach D., Vincent C., and Eisen O. (2013). Determining the age distribution of Colle Gnifetti, Monte Rosa, Swiss Alps, by combining ice cores, ground-penetrating radar and a simple flow model. *Journal of Glaciology* 59(213): 179–189.

Koppal S.J. (2014). Lambertian Reflectance. In: Ikeuchi K. (eds) *Computer Vision*. Springer, Boston, MA. https://doi.org/10.1007/978-0-387-31439-6_534

Kuipers Munneke P., van den Broeke M. R., Reijmer C. H., Helsen M. M., Boot W., Schneebeli M., and Steffen K. (2009). The role of radiation penetration in the energy budget of the snowpack at Summit, Greenland. *The Cryosphere* 3(2): 155–165.

[revised manuscript text omitted]

Warren S. G. (1982). Optical properties of snow. *Reviews of Geophysics* 20(1): 67.

Wolff E. W., Barbante C., Becagli S., Bigler M., Boutron C. F., Castellano E., de Angelis M., Federer U., Fischer H., Fundel F., Hansson M., Hutterli M., Jonsell U., Karlin T., Kaufmann P., Lambert F., Littot G. C., Mulvaney R., Röthlisberger R., Ruth U., Severi M., Siggaard-Andersen M. L., Sime L. C., Steffensen J. P., Stocker T. F., Traversi R., Twarloh B., Udisti R., Wagenbach D., and Wegner A. (2010). Changes in environment over the last 800,000 years from chemical analysis of the EPICA Dome C ice core. *Quaternary Science Reviews* 29(1–2): 285–295.

Yen Y.-C. (1981). *Review of thermal properties of snow, ice and sea ice*. no. 81–10. DTIC.

---

## Author Comment (AC3)

**Firn changes at Colle Gnifetti revealed with a high-resolution process-based physical model approach**
Enrico Mattea *et al.*

**Response to reviewer 3 (Anonymous)**

*In this study, the authors present a coupled energy balance and firn model and compare the model's output to a large dataset of firn temperature records, as well as one firn core record of refrozen melt, at Colle Gnifetti. The authors quantified the increase in firn temperature as well as surface melt totals in this location over the period of 2003-2018. Improving surface energy and firn models is an important pursuit, especially under a warming climate scenario and the uncertainties in firn meltwater retention capabilities.*
*This is a nicely organized and clearly written manuscript. Additionally, the figures are logically organized and easy to interpret (with a few minor suggestions for improvement below). Please find my general and line-specific comments below.*

We wish to thank the referee for the constructive and positive review. In the revised manuscript we are adopting all the suggestions, as detailed below. In this document, the review text is reported in *black italic*, while our responses are in blue. Updated figures are shown at the end of the document.

*General comments:*

*Throughout the Discussion section, there were many mentions of imprecise comparisons and statements of significance without any quantification. The Discussion would be improved by the incorporation of values that justify statements of significant changes, variability, and appearances of correlation.*

In the revised manuscript, we are adding quantification throughout the Discussion section. Specifically, we are adding numerical values to our statements of lines 307, 321 and 390, as well as to the caption of Fig. 6. Additional quantification will be provided in the Results section.

*Further explanation of how the authors calculated the amount of refrozen melt was present in the unifr-2019 firn core would be helpful. How were the 31 cm of refrozen layers in the core, which certainly contained a mixture of ice and firn layers, converted to m w.e.?*

The original manuscript reported the observed total ice content of the core (31 cm), without an estimation the corresponding refrozen amount. We are now making this section more informative by computing an adjusted estimate (14 cm w.e.) for the amount of refrozen ice in the core. We have computed the correction from the mean density of the ice-free core sections. We acknowledge that this computation involves a fair deal of uncertainty, due to the high variability of the density profile even in the ice-free core sections, and also due to the presence of "icy firn" which did not form well-defined ice layers. In the revised manuscript we update the discussion in both Sect. 5.3 and the Appendix.

*The study highlights that there are still many unknowns with respect to predicting the depth of refreezing meltwater in a firn column. The authors mention that the percolation routine for the EBFM needs to account for the firn density and stratigraphy in order to improve the estimates of z_lim. Additionally, the microstructure of firn layers as well as the permeability of both undisturbed firn layers and those containing refrozen meltwater will be important for accurately estimating these depths of percolation.*

We fully agree with this statement. In the revised manuscript we are adding a sensitivity study of the percolation parameter $z_{lim}$, showing a strong dependence of firn temperatures on the choice of its value. Thus, refining the percolation routine can be a future improvement for the EBFM simulation of cold firn.

*Figure 1:*
*- The '(a)' and '(b)' labels in the two panels of the figure are hard to notice. Consider enlarging the labels or bolding the font.*

Done.

*- The legend symbol for the model cells is confusing because it's the same color as only one of the sites (SK). Perhaps make the legend symbol a neutral color to make it clearer that you're referring to all of the square boxes in the figure.*

Done.

*-  It's not clear how the areas depicted in the two panels overlap. Consider adding a marker in panel (b) to designate where the CG study site is.*

Done.

*Figure 3:*
*- Which of the firn core sites in panel (a) is the KCC core site? Indicating this information would give more context to the results shown in panel (b) as well as areas of the text where KCC is mentioned.*

The KCC core is now highlighted in Fig. 3.

*Figure 6:*
*- The '(a)' and '(b)' labels in the two panels of the figure are hard to notice. Consider enlarging the labels or bolding the font.*

Done.

*Figure 8:*
*- It would be helpful to remind the reader of what the energy balance component acronyms stand for in the caption, especially 'SHF', 'LHF', and 'GHF' which are note explicitly defined in the text before this figure.*

We agree, to solve this we are renaming all the energy balance components in the figure to match the notation of Eq. 1.

*Line Specific Comments:*
*Line 27: 'Besides' instead of 'Beside'*

Done.

*Line 35: somewhat awkward transition here. What 'Then' is referring to is vague?*

We agree, we are replacing it with "Thus".

*Lines 48-49: is this the current range of firn temperatures, or the range measured in the 1976 campaign?*

It is the range measured by Suter and Hoelzle (2002): their measurements are the most recent published results which include the south-facing slope in the North of the domain. We are rewording the sentence to make clear the source of the measurements.

*Line 68: change 'on' to 'of' at the end of this line*

Done.

*Line 121: 'Besides' instead of 'Beside'*

Done.

*Lines 234-235: It's not immediately clear why the depth of 4 m in this study matches the firn temperatures of the CG saddle point at a depth of 20 m. What data was compared to determine the 4 m depth for z_lim?*

In the revised manuscript we rewrite this section entirely, describing in detail the sub-surface model and the water percolation routine. We also clarify that the 20 m firn temperatures at the saddle point were measured by Haeberli and Funk (1991).

*Line 314: remove 'in' after 'As such,'*

Done. We are also substantially improving this section by removing the speculative interpretation of the dense, thick refrozen layers with limited refreezing capacity: a more convincing and verifiable explanation is based on thermal conductivity, which has a positive bias due to a bias in density induced by the fixed-depth percolation routine. In Appendix we are adding a sensitivity analysis showing the cooling effect of a positive thermal conductivity bias.

*Line 322: should these units be °C yr-1?*

Good catch, we are correcting the units.

**References**

Haeberli W. and Funk M. (1991). Borehole temperatures at the Colle Gnifetti core-drilling site (Monte Rosa, Swiss Alps), *Journal of Glaciology* 37: 37–46.

Suter S., and Hoelzle M. (2002). Cold firn in the Mont Blanc and Monte Rosa areas, European Alps: spatial distribution and statistical models. *Annals of Glaciology* 35: 9–18.

[Figure]

*Updated Figure 1*

[Figure]

*Updated Figure 3*

[Figure]

*Updated Figure 6*

[Figure]

*Updated Figure 8*

---

## Editor Decision (ED1)

Dear Enrico Mattea and colleagues,

Your manuscript received positive comments, and after the new round of revisions the reviewer who had the most substantial comments (Adrien Gilbert) indicated that "The authors did a very good job in addressing my comments and concerns, I don't have anything to add.".

I went through the last version of the manuscript, and found the story to be very clear and well presented. The figures nicely support your findings – thank you also for incorporating the initial suggestions that I made, e.g. on Figure 2 – and I appreciate the 'honest' way in which your results are presented. The model capabilities are highlighted, but you are not 'hiding' what does not work and give an elaborate explanation concerning the possible discrepancies. I am convinced that some of your main findings (e.g. the role of micro-melt events) will be of large relevance to scientists working on firn modelling and those interested in deriving palaeoclimatic information from ice cores on alpine glaciers.

At this stage, I have formulated a list of final remarks and suggestions that I would like you to address when uploading your (most likely final) version of your manuscript. These are mainly easy-to-incorporate changes, although some may require a small amount of work:

- l.2: "...need to improve understanding and further develop" → "...need to improve our understanding and to further develop"
- l.37: order the references in chronological order
- l.102: "...the coupled model in..." → "...the coupled model is described in..."
- l.114-116: here you describe how other station data is used to perform quality checks, to fill gaps in time series and to determine parameters that were not measured at CM. This seems like quite a lot of work, which must not have been trivial. A few questions here:
  - Do I understand it correctly that you have performed this work yourself? If so, would be good to state this more explicitly
  - Can you say something about the "quality checks"? Potentially give some numbers
  - The new dataset you have created through your approach seems to be quite unique – it is definitely than what is directly available from the data providers. As this dataset would be quite valuable for other researchers, could you make it directly available? So far you mention "the meteorological time series...should be requested from the respective providers" in the "code and data availability section": but what you provide here goes "beyond" this.
- Table 1: what is CAE? Please define
- l.129: "...within the model input", maybe rephrase to "...as model input"?
- l.149: "...18 boreholes, some locations having been measured..." → "18 boreholes, where some locations have been measured..."
- l.184-186: "...corrected for...calibration parameters.". I got a bit lost in this sentence and could not understand 'what is what'. Could you rephrase this, potentially by splitting the explanation in two different sentences?
- Table 4: very nice and useful for the reader! In the caption, you mention "Additional parameters not listed here were kept at the default value". It would probably be useful for the reader to also have this information directly at hand, without having to dive into the three studies that you mention (and having to look for 'what is where'). Could you add this information in a separate table (e.g. in suppl. mat.)?
- l.313: you initialize by running the model over the time period 2004-2011 for 8 times. Two questions here:
  - Why did you decide to go for 8 times, and for instance not 4 times, or 10 times? Would be good if you could give a hint.
  - How sensitive are you results to running this time period 8 times? Could you include a short analysis on this, for instance in Appendix B?
- l.315: you mention '20 m / 1h' and '100 m / 3h' as spatio-temporal resolution. I found this quite confusing at first, as this made me think that you change two things at once: the spatial resolution and the temporal resolution: how can you then discern the effect of both on your results? Subsequently, I kind of understood that you probably change the time resolution to ensure numerical stability (correct?). My question here: would the results differ if you would run with '100 m / 1h' vs. '100 m / 3h' (where the former is obviously computationally more expensive). I would appreciate it if

you could provide some information on this, and potentially reconsider reformulating this to 20 m vs. 100 m, and only mention the time resolution separately, indicating that this is changed to ensure numerical stability (if this would the case of course, I'd gladly be corrected here, but I am trying to take away any possible source of confusion ☺ ).

- l.355-356: "in every month sublimation is a more effective energy sink than melt": ok, nice. Probably not that surprising of a finding for specialists in the field, but was for me at when reading the sentence at first. Maybe also consider mentioning this explicitly in your conclusion?
- l.360: "with only minor amounts...": could you quantify this statement? Would be useful, as is difficult to visually derive from figure 9
- l.364: "... it becomes statistically significant over the rest of the period": maybe good to be more specific here directly – to have info without having to refer to figure explicitly. i.e. "... it becomes statistically significant over the rest of the period ($p<0.05$ for 2004-2018)"
- l.370: "The majority of melt happens under clear sky conditions...". This contrasts with the previous sentence, from which I had derived that the cloudiness / sky conditions do not play a big role ("...unlike cloud cover which appears to have almost no effect"). Maybe consider slightly rewording? Could potentially remove the clear sky info and just focus on the "slightly positive temperatures"?
- l. 375: "...mean melt rates, slightly decreasing the likelihood of melt under high winds": could you add a sentence on why the likelihood is decreasing in this case?
- l.392-400: you explain how the density does not increase as long as melt-refreezing occurs at same location, and how your model is not accounting for this. Could you provide a hint somewhere about how this could be solved? Not suggesting that this needs to be changed, but would be good to provide a possible solution (like you do for some of the other limitations that you nicely put forward!)
- l.415: maybe reword to: "...be affected by the lack of SW radiation reflected from the surrounding terrain in the modeled SEB"
- l.442-443: "temperatures were initialized with repeated model runs over 2004-2011": see related question earlier. If you would have repeated this more (or less) than 8 times, how would this have affected your firn warming?
- "Conclusions and outlook": nice overview of your study! It would maybe be useful, for someone who has a quick look at the paper and directly looks at your conclusions, to state here which data is used for calibration/tuning, and which one for evaluation: e.g. for external reader it is not clear whether the fact that the firn temperatures are reproduced reasonably well is a result from the fact that you tune to this (i.e. calibration) or that this is just an outcome of your model without specific tuning to this (i.e. evaluation). Would be useful to shortly say something about this.
- l.555: 'code and data availability': would really be nice if you could also provide the processed series for GC directly, where the data gaps are filled and non-measured variables are derived from other station data (see comment on l.114-116)
- Please acknowledge the three reviewers (Adrien Gilbert, Vincent Verjans and one anonymous) in the 'Acknowledgments' section

Thank you for going through this final series of minor comments. Once these are addressed, we should be able to proceed to the acceptance of your manuscript.

Best regards,
Harry

---

## Author Response (AR2)

**Firn changes at Colle Gnifetti revealed with a high-resolution process-based physical model approach**
Enrico Mattea *et al.*

**Response to final remarks from the editor**

*I went through the last version of the manuscript, and found the story to be very clear and well presented. The figures nicely support your findings – thank you also for incorporating the initial suggestions that I made, e.g. on Figure 2 – and I appreciate the 'honest' way in which your results are presented. The model capabilities are highlighted, but you are not 'hiding' what does not work and give an elaborate explanation concerning the possible discrepancies. I am convinced that some of your main findings (e.g. the role of micro-melt events) will be of large relevance to scientists working on firn modelling and those interested in deriving palaeoclimatic information from ice cores on alpine glaciers.*

We wish to sincerely thank the editor and the referee for the positive comments on our manuscript. We have incorporated most of the suggested changes; exceptions are discussed below. Editor comments are reported in *black italic,* while our responses are in blue. Moreover, in the manuscript we have now updated the DOI link to the GitHub repository with the model code: we have cleaned up the code in terms of commented lines and indentation to make it easier to read (but without any functional change).

*At this stage, I have formulated a list of final remarks and suggestions that I would like you to address when uploading your (most likely final) version of your manuscript. These are mainly easy-to-incorporate changes, although some may require a small amount of work:*

- *l.2: "...need to improve understanding and further develop" → "...need to improve our understanding and to further develop"*
  Done.

- *l.37: order the references in chronological order*
  Done.

- *l.102: "...the coupled model in…" → "...the coupled model is described in…"*
  Done.

- *l.114-116: here you describe how other station data is used to perform quality checks, to fill gaps in time series and to determine parameters that were not measured at CM. This seems like quite a lot of work, which must not have been trivial. A few questions here:*
  ○ *Do I understand it correctly that you have performed this work yourself? If so, would be good to state this more explicitly*
    Indeed, we have performed this (quite time-consuming) work ourselves. We have updated the text to state this more clearly.

  ○ *Can you say something about the "quality checks"? Potentially give some numbers*
    In the manuscript, we provided a simple overview of our workflow for quality checks: "automated pre-filtering routine […] based on objective criteria (absolute values, rates of change, comparison with reconstructed series and reanalysis). […] potential outliers […] were then manually checked. The CM AWS was always processed last […]". We have now added a new table (Table 3) with the numbers and fractions of missing values/gaps

for each measured meteorological parameter, before and after the quality check of the CM series. For a more detailed (technical) description, the interested reader is directed (l.138-139) to the full description in Mattea (2020).

- *The new dataset you have created through your approach seems to be quite unique–it is definitely than what is directly available from the data providers. As this dataset would be quite valuable for other researchers, could you make it directly available? So far you mention "the meteorological time series...should be requested from the respective providers"in the "code and data availability section": but what you provide here goes "beyond"this.*
  We have contacted all data providers, asking about this possibility. Unfortunately, MeteoSwiss policy explicitly prohibits users from providing a public download link to anything which is (partly) based on their paid data/services (in our case, the hourly series of Gornergrat and Monte Rosa-Plattje, which we used to validate and complete the CM AWS data). Thus, we are allowed to provide our processed series only to a third-party who has already obtained access to the original datasets by the respective providers.

- *Table 1: what is CAE? Please define*
  CAE is the name of the company which has supplied the CM AWS sensors. Apparently it is not an acronym (https://www.cae.it/eng/), so at present we do not have additional information which could be added to the table. An internet search for e.g. "CAE" "VV20" immediately provides the sensor data-sheet, but we welcome any suggestion as to possible rewording within the table.

- *l.129: "...within the model input", maybe rephrase to "...as model input"?*
  Done.

- *l.149: "...18 boreholes, some locations having been measured…" → "18 boreholes, where some locations have been measured…"*
  Done.

- *l.184-186: "...corrected for...calibration parameters.". I got a bit lost in this sentence and could not understand 'what is what'. Could you rephrase this, potentially by splitting the explanation in two different sentences?*
  Done.

- *Table 4: very nice and useful for the reader! In the caption, you mention "Additional parameters not listed here were kept at the default value". It would probably be useful for the reader to also have this information directly at hand, without having to dive into the three studies that you mention (and having to look for 'what is where'). Could you add this information in a separate table (e.g. in suppl. Mat.)?*
  We have added a table with the additional parameters, their value, unit and explanation in the supplementary material.

- *l.313: you initialize by running the model over the time period 2004-2011 for 8 times. Two questions here:*
  - *Why did you decide to go for 8 times, and for instance not 4 times, or 10 times? Would be good if you could give a hint.*
    We opted for 8 times because the resulting 64-year spin-up series is just long enough (with a small margin) for the whole grid to reach temperature and density equilibrium with the surface forcing, up to a depth of 20 m over the entire domain. Specifically, after the 64-year spin-up the entire grid down to a depth of at least 20 m consists of snow and

firn layers which have been added by snowfall during the spin-up period (moving layers scheme: Sect. 3.4). We also experimented with shorter spin-up runs, where we observed a temperature and density transient at depth, most notable on the grid cells with lowest accumulation (Fig. 3a). An even longer spin-up series does not provide any advantage (within the first 20 m of the domain) and would be very time consuming, since model spin-up is needed after each parameter change (thus also for sensitivity experiments), to avoid transient adjustments. We have added a brief mention of the role of spin-up duration to the "Model initialization" section.

- ○ *How sensitive are you results to running this time period 8 times? Could you include a short analysis on this, for instance in Appendix B?*
  We have tested sensitivity with two model runs, initialized using 4x8 and 12x8 spin-up loops instead of the 8x8 used for the main run. With the longer spin-up, the model output is basically the same (due to periodic forcing at the surface, which produces periodic conditions within the firn: the temperature change is within 0.02 °C everywhere and typically one order of magnitude smaller). The shorter spin-up run results in a firn temperature and density transient at the beginning of the actual model run, within the depth region which has not yet been reached by the newly accumulated snow layers. Such a transient is strongly affected by the choice of artificial starting conditions imposed at the beginning of the spin-up run. By contrast, these artificial conditions have no effect on the actual initial conditions of the main simulation (thanks to the complete 8x8 spin-up run, which forces a complete replacement of firn layers within the first 20 m): thus, we think it would not be very informative to quantify (one arbitrary example of) this transient adjustment. Section "3.5 Model initialization" now explains that the used spin-up allows complete adjustment of the grid to the surface forcing.

- *l.315: you mention '20 m / 1h' and '100 m / 3h' as spatio-temporal resolution. I found this quite confusing at first, as this made me think that you change two things at once: the spatial resolution and the temporal resolution: how can you then discern the effect of both on your results? Subsequently, I kind of understood that you probably change the time resolution to ensure numerical stability (correct?). My question here: would the results differ if you would run with '100 m / 1h' vs. '100 m / 3h' (where the former is obviously computationally more expensive). I would appreciate it if you could provide some information on this, and potentially reconsider reformulating this to 20 m vs. 100 m, and only mention the time resolution separately, indicating that this is changed to ensure numerical stability (if this would the case of course, I'd gladly be corrected here, but I am trying to take away any possible source of confusion ☺ ).*
  During exploratory work we performed some tests at 3 h / 20 m, and we did not find a significant difference compared to the higher time resolution (1 h / 20 m). For our analyses we used the 20 m / 1 h results as often as possible (since some of the shorter and less intense micro-melt events could in principle be lost within a 3 h run; also, surface slope is more accurate on a 20 m grid). For the examination of the full sub-surface grids, the data volume of the model output was simply too large to process and analyze (our 55x70x250 grid on 140256 hourly time-steps produces about 135 billion values for each sub-surface variable), so for example we show the 100 m / 3 h result in Fig. 6.

- *l.355-356: "in every month sublimation is a more effective energy sink than melt": ok, nice. Probably not that surprising of a finding for specialists in the field, but was for me at when reading the sentence at first. Maybe also consider mentioning this explicitly in your conclusion?*
  Done.

- *l.360: "with only minor amounts...": could you quantify this statement? Would be useful, as is difficult to visually derive from figure 9*
  Done. The "minor amounts" of April and October together reach up to 1 % of the annual melt totals at ZS (less at lower-melt locations, where melt is even less frequent in these two months).

- *l.364: "... it becomes statistically significant over the rest of the period": maybe good to be more specific here directly–to have info without having to refer to figure explicitly. i.e. "... it becomes statistically significant over the rest of the period (p < 0.05 for 2004-2018)"*
  Done.

- *l.370: "The majority of melt happens under clear sky conditions...". This contrasts with the previous sentence, from which I had derived that the cloudiness / sky conditions do not play a big role ("...unlike cloud cover which appears to have almost no effect"). Maybe consider slightly rewording? Could potentially remove the clear sky info and just focus on the "slightly positive temperatures"?*
  Cloud cover appears to have almost no effect on melt rates (i.e. the melt intensity, in mm w.e. h$^{-1}$), but the total amounts also depend on the frequency of occurrence of each set of meteorological conditions, and especially on the correlation between variables, such as warm temperatures and clear skies. We believe that it is important to convey both findings: (1) that cloud cover does not strongly affect the intensity of a melt event (thus for example melt duration might be investigated as a proxy for total melt amount within a specific melt event, even with limited or absent cloud cover information); and (2) that it is under clear skies and slightly positive air temperatures that the majority of meltwater and refrozen ice are produced at our site (and not, for example, only during extreme heat waves with air temperatures above 5 °C). We have reworded the sentence, adding a mention of the warm air/clear skies correlation, to clarify the different behavior of melt rates and total melt amounts.

- *l. 375: "...mean melt rates, slightly decreasing the likelihood of melt under high winds": could you add a sentence on why the likelihood is decreasing in this case?*
  Done. The effect appears to happen only at air temperatures between -5 and 0 °C, thus it is likely related to turbulent losses.

- *l.392-400: you explain how the density does not increase as long as melt-refreezing occurs at same location, and how your model is not accounting for this. Could you provide a hint somewhere about how this could be solved? Not suggesting that this needs to be changed, but would be good to provide a possible solution (like you do for some of the other limitations that you nicely put forward!)*
  The issue could be addressed with a dependence of percolation depths on melt rates/amounts, possibly in the form of a threshold for the occurrence of deep preferential percolation. With this refinement, the very small meltwater amounts which in reality undergo repeated melt-freeze cycles would no longer be allowed to percolate to 4 m in the model: they would remain close to the surface where they could melt again within a later melt event, avoiding unrealistic density increases. We are adding a summary of these considerations to the section.

- *l.415: maybe reword to: "...be affected by the lack of SW radiation reflected from the surrounding terrain in the modeled SEB"*
  Done.

- *l.442-443: "temperatures were initialized with repeated model runs over 2004-2011": see related question earlier. If you would have repeated this more (or less) than 8 times, how would this have affected your firn warming?*
  See answer to earlier question.

- *"Conclusions and outlook": nice overview of your study! It would maybe be useful, for someone who has a quick look at the paper and directly looks at your conclusions, to state here which data is used for calibration/tuning, and which one for evaluation: e.g. for external reader it is not clear whether the fact that the firn temperatures are reproduced reasonably well is a result from the fact that you tune to this (i.e. calibration) or that this is just an outcome of your model without specific tuning to this (i.e. evaluation). Would be useful to shortly say something about this.*
  We have now mentioned in the conclusions that we tune the model to a single deep value at the saddle point, and evaluate the model over 25 profiles distributed over the domain.

- *l.555: 'code and data availability': would really be nice if you could also provide the processed series for GC directly, where the data gaps are filled and non-measured variables are derived from other station data (see comment on l.114-116)*
  See answer to comment on l.114-116.

- *Please acknowledge the three reviewers (Adrien Gilbert, Vincent Verjans and one anonymous) in the 'Acknowledgments' section*
  Done.